# FEATURE COLLAPSE

**Thomas Laurent[1], James H. von Brecht, Xavier Bresson[2]**
[1] Loyola Marymount University, `tlaurent@lmu.edu`
[2] National University of Singapore, `xaviercs@nus.edu.sg`

## ABSTRACT

We formalize and study a phenomenon called *feature collapse* that makes precise the intuitive idea that entities playing a similar role in a learning task receive similar representations. As feature collapse requires a notion of task, we leverage a synthetic task in which a learner must classify 'sentences' constituted of $L$ tokens. We start by showing experimentally that feature collapse goes hand in hand with generalization. We then prove that, in the large sample limit, distinct tokens that play identical roles in the task receive identical local feature in the first layer of the network. This analysis shows that a neural network trained on this task provably learns interpretable and meaningful representations in its first layer.

## 1 INTRODUCTION

Many machine learning practices implicitly rely on the belief that good generalization and transfer learning requires good features. Despite this, the notion of 'good features' remains vague and carries many potential meanings. One definition is that features/representations should only encode the information necessary to do the task at hand, and discard any unnecessary information as noise. For example, two distinct patches of grass should map to essentially identical representations even if these patches differ in pixel space. Intuitively, a network that gives the same representation to many distinct patches of grass has learned the 'grass' concept. We call this phenomenon feature collapse, meaning that a learner gives the same feature to entities that play similar roles for the task at hand. This phenomenon captures the common intuition, often confirmed by practice, that the ability to learn good representations in early layers is essential for the empirical success of neural networks.

Broadly speaking, we conduct a theoretical investigation into how such representations are learned in the early layers of a neural network. To make progress we adopt a common approach in theoretical deep-learning community. One starts with a synthetic data model exhibiting a clear latent structure, and then prove that a specific neural network architecture successfully uncovers this latent structure during training. For example, recent work in representation learning (Damian et al., 2022; Mousavi-Hosseini et al., 2022) leverages a data model (the multiple-index model) with latent structure defined by a low dimensional subspace. In this specific setting, the first layer of a fully connected neural network provably learns this low-dimensional subspace.

Our work follows along the same lines. Specifically, we start with a data model that asks a learner to classify 'sentences' comprised of $L$ tokens. Some of these tokens play identical roles in the sense that replacing one with another does not change the label of the sentence, and this equivalence of tokens defines the latent structure in the data. We then consider a neural network containing a shared embedding module, optionally followed by a LayerNorm module, and then a final linear classification head. We show that this network, when equipped with the LayerNorm module, provably learn the equivalence of these tokens in its first layer. To do so, we consider the large sample limit and derive analytical formulas for the weights of the network trained on the data model. Under some symmetry assumptions on the task, these analytical formulas reveal:

(i) If the network includes LayerNorm then feature collapse takes place. The neural network learns to give the same embedding to tokens that play identical roles.

(ii) If the network does not include LayerNorm then feature collapse fails. The network does not give the same embedding to tokens that play identical roles. Moreover, this failure stems from the fact that common tokens receive large embeddings and rare tokens receive small embeddings.

Finally, we conduct experiments that show feature collapse and generalization go hand in hand. These experiments demonstrate that for the network to generalize well it is essential for tokens playing identical roles to receive the same embeddings.

In summary, our main contributions are as follow:

- We study how a network learns representations in its first layer. To do so, we make the notion of 'good features' mathematically rigorous via a synthetic data model with latent structure.

- We derive analytical formulas for the weights of a two-layer network trained on this data model. These analytical formulas show that, when equipped with a LayerNorm module, the network provably learns interpretable and meaningful representations in its first layer.

The remainder of the paper proceeds as follows: In subsection 1.1 we discuss related works; In section 2 we describe the data model and present a set of visual experiments to illustrate the main ideas used in the paper; In section 3 we present the three theorems that constitute the main results of the paper; Finally, section 4 outlines a set of additional experiments performed in the appendix.

## 1.1 RELATED WORKS

Our work most closely resembles recent work on theoretical representation learning and, to some extent, the recent literature on neural collapse. The works by Damian et al. (2022) and Mousavi-Hosseini et al. (2022) consider a synthetic data model, the multiple-index model, to investigate representation learning. In this model, the learner must solve a regression task with normally distributed inputs $\mathbf{x} \in \mathbb{R}^d$ and targets $y = g(\langle \mathbf{u}_1, \mathbf{x} \rangle, \ldots, \langle \mathbf{u}_r, \mathbf{x} \rangle)$ for some function $g : \mathbb{R}^r \to \mathbb{R}$, some vectors $\mathbf{u}_1, \ldots, \mathbf{u}_r \in \mathbb{R}^d$, and with $r \ll d$. The target $y$ therefore solely depends on the projection of $\mathbf{x}$ on the low-dimensional subspace spanned by $\mathbf{u}_1, \ldots, \mathbf{u}_r$. These works prove that a fully connected, two-layer neural network learns, in its first layer, the low dimensional subspace. The behavior of fully connected networks trained on the multiple-index model, or on related data models, has also been studied in various other works, including Ba et al. (2022), Bietti et al. (2022), Abbe et al. (2022) and Parkinson et al. (2023). Both this line of investigation and our work prove, in the appropriate sense, that a network uncovers latent structure. Nevertheless, our data model differs quite substantially from the multi-index model. Moreover, we do not study a fully connected network. Our network has shared embedding module, and this allows us to tie good features to a notion of semantic equivalence.

The phenomenon we study in this work, *feature collapse*, has superficial similarity to *neural collapse* but in detail is quite different. In a pioneering work, Papyan et al. (2020) conducted a series of experiments that revealed that a well-trained network gives identical representations, in its last layer, to training points that belong to the same class. In a $K$-class classification task we therefore see the emergence, in the last layer, of $K$ vectors coding for the $K$ classes. Additionally, these $K$ vectors 'point' in 'maximally opposed' directions. This phenomenon, coined *neural collapse*, has been studied extensively since its discovery (e.g. Mixon et al. (2020); Lu & Steinerberger (2020); Wojtowytsch et al. (2020); Fang et al. (2021); Zhu et al. (2021); Ji et al. (2021); Tirer & Bruna (2022); Zhou et al. (2022)). To emphasize the difference with feature collapse, note that neural collapse refers to a phenomenon where all training points from the same class receive the same representation at the end of the network. Unlike feature collapse, this does not provide any indication that the network has learned good representations in its early layers, or that the neural network has uncovered some latent structure beyond the one encoded in the training labels.

## 2 A TALE OF FEATURE COLLAPSE

We begin by more fully telling the empirical tale that motivates our theoretical investigation. To make progress we adopt a common approach in theoretical deep-learning and leverage a synthetic data model exhibiting a clear latent structure. The model generates sequences of length $L$ from some underlying set of latent variables that encode the $K$ classes of a classification task. To make it concrete we describe this generative process using NLP terminology like 'sentences' and 'words', but of course do not intend it as a realistic language model. Figure 1 illustrates the basic idea. The left side of the figure depicts a vocabulary of $n_w = 12$ word tokens and $n_c = 3$ concept tokens

$$\mathcal{V} = \{\text{potato, cheese, carrots, chicken}, \ldots\} \qquad \text{and} \qquad \mathcal{C} = \{\text{vegetable, dairy, meat}\}$$

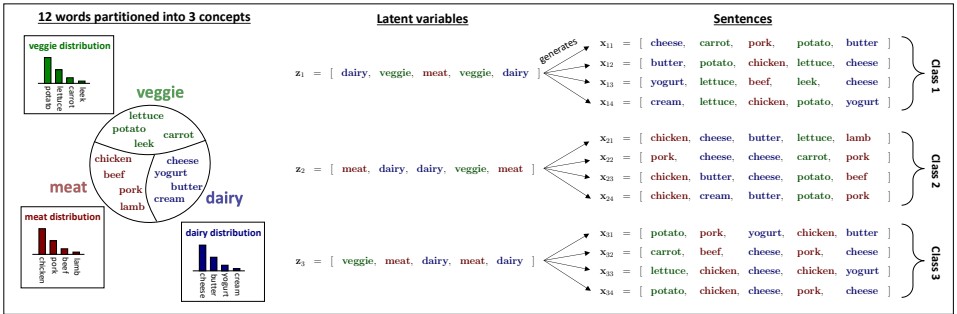

Figure 1: Data model with parameters set to $L = 5$, $n_w = 12$, $n_c = 3$, $K = 3$.

with the 12 words partitioned into the 3 equally sized concepts. A sentence $\mathbf{x} \in \mathcal{V}^L$ is a sequence of $L$ words ($L = 5$ on the figure), and a latent variable $\mathbf{z} \in \mathcal{C}^L$ is a sequence of $L$ concepts. The latent variables generate sentences. For example

$$\mathbf{z} = \begin{bmatrix} \text{dairy, veggie, meat, veggie, dairy} \end{bmatrix} \quad \xrightarrow{\text{generates}} \quad \mathbf{x} = \begin{bmatrix} \text{cheese, carrot, pork, potato, butter} \end{bmatrix}$$

with the sentence on the right obtained by sampling each word at random from the corresponding concept. The first word represents a random sample from the dairy concept (*butter, cheese, cream, yogurt*) according to the dairy distribution (square box at left), the second word represents a random sample from the vegetable concept (*potato, carrot, leek, lettuce*) according to the vegetable distribution, and so forth. At right, figure 1 depicts a classification task with $K = 3$ categories prescribed by the three latent variables $\mathbf{z}_1, \mathbf{z}_2, \mathbf{z}_3 \in \mathcal{C}^L$. Sentences generated by the latent variable $\mathbf{z}_k$ share the same label $k$, yielding a classification problem that requires a learner to classify sentences among $K$ categories.

We use two similar networks to empirically study if and when the feature collapse phenomenon occurs. The first network $\mathbf{x} \mapsto h_{W,U}(\mathbf{x})$, depicted on the top panel of figure 2, starts by embedding each word in a sentence by applying a $d \times n_w$ matrix $W$ to the one-hot representation of each word. It then concatenates these $d$-dimensional embeddings of each word into a single vector. Finally, it applies a linear transformation $U$ to produce a $K$-dimensional score vector $\mathbf{y} = h_{W,U}(\mathbf{x})$ with one entry for each of the $K$ classes. The $d \times n_w$ embedding matrix $W$ and the $K \times Ld$ matrix $U$ of linear weights are the only learnable parameters, and the network has no nonlinearities. The second network $\mathbf{x} \mapsto h^*_{W,U}(\mathbf{x})$, depicted at bottom, differs only by the application of a LayerNorm module to the word embeddings prior to the concatenation. For simplicity we use a LayerNorm module which does not contain any learnable parameters; the module simply removes the mean and divides by the standard deviation of its input vector. As for the first network, the only learnable weights are $W$ and $U$.

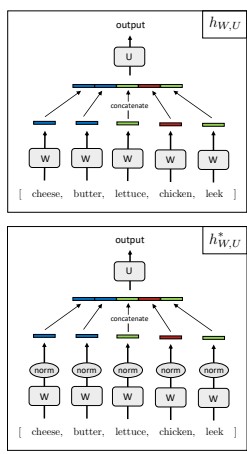

Figure 2: Networks

The task depicted on figure 1, and the networks depicted on figure 2, provide a clear way of studying how interpretable and meaningful representations are learned in the first layer of a network. For example, the four words *butter*, *cheese*, *cream* and *yogurt* clearly play identical role for the task at hand (replacing one with another does not change the label of the sentence). As a consequence we would expect the embedding layer of the network to map them to the same representation. Similarly, the words belonging to the *vegetable* concepts should receive same representation, and the words belonging to the *meat* concept should receive same representation. If this takes place, we say that feature collapse has occurred.

If feature collapse occurs, this will also be reflected in the second layer of the network. To see this, partition the linear transformation

$$U = \begin{bmatrix} -\mathbf{u}_{1,1}- & -\mathbf{u}_{1,2}- & \cdots & -\mathbf{u}_{1,L}- \\ -\mathbf{u}_{2,1}- & -\mathbf{u}_{2,2}- & \cdots & -\mathbf{u}_{2,L}- \\ \vdots & \vdots & & \vdots \\ -\mathbf{u}_{K,1}- & -\mathbf{u}_{K,2}- & \cdots & -\mathbf{u}_{K,L}- \end{bmatrix} \tag{1}$$

into its components $\mathbf{u}_{k,\ell} \in \mathbb{R}^d$. Suppose for example that $z_{k,\ell} = veggie$, meaning that the latent variable $\mathbf{z}_k$ contains the *veggie* concept in the $\ell$th position. If $W$ properly encodes concepts then we expect the vector $\mathbf{u}_{k,\ell}$ to give a strong response when presented with the embedding of a word that belongs to the *veggie* concept. So we would expect $\mathbf{u}_{k,\ell}$ to align with the embeddings of the words that belong to the *veggie* concept, and so feature collapse would occur in this manner as well.

In the remainder of this section we conduct experiments that visually illustrate the feature collapse phenomenon and the formation of interpretable representations in networks $h$ and $h^*$. These experiments also show how layer normalization plays a key role in the feature collapse phenomenon. In our experiments we use the standard cross entropy loss $\ell(\mathbf{y}, k)$, with $\mathbf{y} \in \mathbb{R}^K$ and $1 \leq k \leq K$, and then minimize the corresponding regularized empirical risks

$$\mathcal{R}_{\text{emp}}(W, U) = \frac{1}{K} \frac{1}{n_{\text{spl}}} \sum_{k=1}^{K} \sum_{i=1}^{n_{\text{spl}}} \ell\big( h_{W,U}\left(\mathbf{x}_{k,i}\right) \ , \ k \ \big) + \frac{\lambda}{2} \|U\|_F^2 + \frac{\lambda}{2} \|W\|_F^2 \tag{2}$$

$$\mathcal{R}^*_{\text{emp}}(W, U) = \frac{1}{K} \frac{1}{n_{\text{spl}}} \sum_{k=1}^{K} \sum_{i=1}^{n_{\text{spl}}} \ell\big( h^*_{W,U}\left(\mathbf{x}_{k,i}\right) \ , \ k \ \big) + \frac{\lambda}{2} \|U\|_F^2 \tag{3}$$

of each network via stochastic gradient descent. The $\mathbf{x}_{k,i}$ denote the $i$-th sentence of the $k$-th category in the training set, and so each of the $K$ categories has $n_{\text{spl}}$ representatives. For the parameters of the architecture, loss, and training procedure, we use an embedding dimension of $d = 100$, a weight decay of $\lambda = 0.001$, a mini-batch size of 100 and a constant learning rate 0.1, respectively, for all experiments. The LayerNorm module implicitly regularizes the matrix $W$ so we do not penalize it in equation (3). The codes for our experiments are available at `https://github.com/xbresson/feature_collapse`.

**Remark:** Without weight decay (i.e. $\lambda = 0$), the above objectives typically do not have global minima. We therefore focus our theoretical investigation on the case $\lambda > 0$ which is analytically more tractable. In appendix A.1 we provide an empirical investigation of the case without weight decay to show that both cases (i.e. $\lambda > 0$ and $\lambda = 0$) exhibit the same behavior in practice.

### 2.1 THE UNIFORM CASE

We start with an instance of the task from figure 1 with parameters $n_c = 3$, $n_w = 1200$, $L = 15$, $K = 1000$, and with uniform word distributions. So each of the 3 concepts (say *veggie*, *dairy*, and *meat*) contain 400 words and the corresponding distributions (the veggie distribution, the dairy distribution, and the meat distribution) are uniform. We form $K = 1000$ latent variables $\mathbf{z}_1, \ldots, \mathbf{z}_{1000}$ by selecting them uniformly at random from the set $\mathcal{C}^L$, which simply means that any concept sequence $\mathbf{z} = [z_1, \ldots, z_L]$ has an equal probability of occurrence. We then construct a training set by generating $n_{\text{spl}} = 5$ data points from each latent variable. We then train both networks $h$, $h^*$ and evaluate their generalization performance; both achieve $100\%$ accuracy on test points.

Since both networks generalize perfectly, we expect them to have learned good representations. To confirm this, we start by visualizing in figure 3 the learnable parameters $W, U$ of the network $h_{W,U}$ after training. The embedding matrix $W$ contains $n_w = 1200$ columns. Each column is a vector in $\mathbb{R}^{100}$ and corresponds to a word embedding. The top panel of figure 3 depicts these 1200 word embeddings after dimensionality reduction via PCA. The top singular values $\sigma_1 = 34.9$, $\sigma_2 = 34.7$ and $\sigma_3 = 0.001$ associated with the PCA indicate that the word embeddings essentially live in a 2 dimensional subspace of $\mathbb{R}^{100}$, and so the PCA paints an accurate picture of the distribution of word embeddings. We then color code each word embedding accorded to its concept, so that all embeddings of words within a concept receive the same color (say all *veggie* words in green, all *dairy* words in blue, and so forth). As the figure illustrates, words from the same concept receive nearly identical embeddings, and

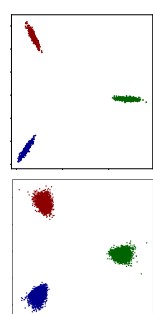

Figure 3: $W$ & $U$

these embeddings form an equilateral triangle or two-dimensional simplex. We therefore observe collapse of features into a set of $n_c = 3$ *equi-angular vectors* at the level of word embeddings. The bottom panel of figure 3 illustrates collapse for the parameters $U$ of the linear layer. We partition the matrix $U$ into vectors $\mathbf{u}_{k,\ell} \in \mathbb{R}^{100}$ via (1) and visualize them once again with PCA. As for the word embeddings, the singular values of the PCA ($\sigma_1 = 34.9$, $\sigma_2 = 34.6$ and $\sigma_3 = 0.0003$) reveal

that the vectors $\mathbf{u}_{k,\ell}$ essentially live in a two dimensional subspace of $\mathbb{R}^{100}$. We color code each $\mathbf{u}_{k,\ell}$ according to the concepts contained in the corresponding latent variable (say $\mathbf{u}_{k,\ell}$ is green if $z_{k,\ell} = veggie$, and so forth). The figure indicates that vectors $\mathbf{u}_{k,\ell}$ that correspond to a same concept collapse around a single vector. A similar analysis applied to the weights of the network $h^*_{W,U}$ tells the same story, provided we examine the actual word features (i.e. the embeddings *after* the LayerNorm) rather than the weights $W$ themselves.

In theorem 1 and 3 (see section 3) we prove the correctness of this empirical picture. We show that the weights of $h$ and $h^*$ collapse into the configurations illustrated on figure 3 in the large sample limit. Moreover, this limit captures the empirical solution very well. For example, the word embeddings in figure 3 have a norm equal to $1.41 \pm 0.13$, while we predict a norm of $1.42214$ theoretically.

## 2.2 THE LONG-TAILED CASE

At a superficial glance it appears as if the LayerNorm module plays no essential role, as both networks $h$ and $h^*$, in the previous experiment, exhibit feature collapse and generalize perfectly. To probe this issue further, we continue our investigation by conducting a similar experiment (keeping $n_c = 3$, $n_w = 1200$, $L = 15$, and $K = 1000$) but with non-uniform, long-tailed word distributions within each of the $n_c = 3$ concepts. For concreteness, say the *veggie* concept contains the 400 words

$$potato, \quad lettuce, \quad \ldots\ldots, \quad arugula, \quad parsnip, \ldots\ldots, \quad achojcha$$

where *achojcha* is a rare vegetable that grows in the Andes mountains. We form the *veggie* distribution by sampling *potato* with probability $C/1$, sampling *lettuce* with probability $C/2$, and so forth down to *achojcha* that has probability $C/400$ of being sampled ($C$ is chosen so that all the probabilities sum to 1). This "$1/i$" power law distribution has a long-tail, meaning that relatively infrequent words such as *arugula* or *parsnip* collectively capture a significant portion of the mass. Natural data in the form of text or images typically exhibit long-tailed distributions (Salakhutdinov et al., 2011; Zhu et al., 2014; Liu et al., 2019; Feldman, 2020; Feldman & Zhang, 2020). For instance, the frequencies of words in natural text approximately conform to the "$1/i$" power law distribution, also known as Zipf's law (Zipf, 1935), which motivates the specific choice made in this experiment. Many datasets of interest display some form of long-tail behavior, whether at the level of object occurrences in computer vision or the frequency of words or topics in NLP, and effectively addressing these long-tail behaviors is frequently a challenge for the learner.

To investigate the impact of a long-tailed word distributions, we first randomly select the latent variables $\mathbf{z}_1, \ldots, \mathbf{z}_{1000}$ uniformly at random as before. We then use them to build two distinct training sets. We build a large training set by generating $n_{\text{spl}} = 500$ training points per latent variable and a small training set by generating $n_{\text{spl}} = 5$ training points per latent variable. We use the "$1/i$" power law distribution when sampling words from concepts in both cases. We then train $h$ and $h^*$ on both training sets and evaluate their generalization performance. When trained on the large training set, both are $100\%$ accurate at test time (as they should be — the large training set has $500,000$ total samples). A significant difference emerges between $h$ and $h^*$ when trained on the small training set. The network $h$ achieves a test accuracy of $45\%$ while $h^*$ remains $100\%$ accurate.

We once again visualize the weights of each network to study the relationship between generalization and collapse. Figure 4(a) depicts the weights of $h_{W,U}$ (via dimensionality reduction and color coding) after training on the large training set. The word embeddings are on the left sub-panel and the linear weights $\mathbf{u}_{k,\ell}$ on the right sub-panel. Words that belong to the same concept still receive

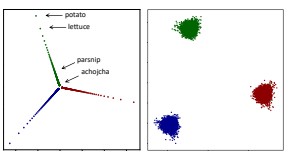 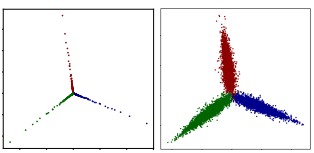 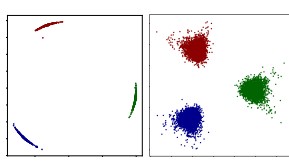

(a) $h$ trained on the **large** training set. Test acc. $= 100\%$

(b) $h$ trained on the **small** training set. Test acc. $= 45\%$

(c) $h^*$ trained on the **small** training set. Test acc. $= 100\%$

Figure 4: Visualization of matrices $W$ (left in each subfigure) and $U$ (right in each subfigure)

embeddings that are aligned, however, the magnitude of these embeddings depends upon word frequency. The most frequent words in a concept (e.g. *potato*) have the largest embeddings while the least frequent words (e.g. *achojcha*) have the smallest embeddings. In other words, we observe 'directional collapse' of the embeddings, but the magnitudes do not collapse. In contrast, the linear weights $\mathbf{u}_{k,\ell}$ mostly concentrate around three well-defined, equi-angular locations; they collapse in both direction and magnitude. A major contribution of our work (c.f. theorem 2 in the next section) is a theoretical insight that explains the configurations observed in figure 4(a), and in particular, explains why the magnitudes of word embeddings depend on their frequencies.

Figure 4(b) illustrates the weights of $h_{W,U}$ after training on the small training set. While the word embeddings exhibit a similar pattern as in figure 4(a), the linear weights $\mathbf{u}_{k,\ell}$ remain dispersed and fail to collapse. This leads to poor generalization performance ($45\%$ accuracy at test time).

To summarize, when the training set is large, the linear weights $\mathbf{u}_{k,\ell}$ collapse correctly and the network $h_{W,U}$ generalizes well. When the training set is small the linear weights fail to collapse, and the network fails to generalize. This phenomenon can be attributed to the long-tailed nature of the word distribution. To see this, say that $\mathbf{z}_k = [\,\text{veggie, dairy, veggie}, \ldots, \text{meat, dairy}\,]$ represents the $k^{\text{th}}$ latent variable for the sake of concreteness. With only $n_{\text{spl}} = 5$ samples for this latent variable, we might end up in a situation where the 5 words selected to represent the first occurrence of the *veggie* concept have very different frequencies than the five words selected to represent the third occurrence of the *veggie* concept. Since word embeddings have magnitudes that depend on their frequencies, this will result in a serious imbalance between the vectors $\mathbf{u}_{k,1}$ and $\mathbf{u}_{k,3}$ that code for the first and third occurrence of the *veggie* concept. This leads to two vectors $\mathbf{u}_{k,1}, \mathbf{u}_{k,3}$ that code for the same concept but have different magnitudes (as seen on figure 4(b)), so features do not properly collapse. This imbalance results from the 'noise' introduced by sampling only 5 training points per latent variable. Indeed, if $n_{\text{spl}} = 500$ then each occurrence of the veggie concept will exhibit a similar mix of frequent and rare words, $\mathbf{u}_{k,1}$ and $\mathbf{u}_{k,3}$ will have roughly same magnitude, and full collapse will take place (c.f. figure 4(a)). Finally, the poor generalization ability of $h_{W,U}$ when the training set is small really stems from the long-tailed nature of the word distribution. The failure mechanism occurs due to the relatively balanced mix of rare and frequent words that occurs with long-tailed data. If the data were dominated by a few very frequent words, then all rare words combined would just contribute small perturbations and would not adversely affect performance.

We conclude this section by examining the weights of the network $h^*_{W,U}$ after training on the small training set. The left panel of figure 4(c) provides a visualization of the word embeddings *after* the LayerNorm module. These word representations collapse both in *direction* and *magnitude*; they do not depend on word frequency since the LayerNorm forces vectors to have identical magnitude. The right panel of figure 4(c) depicts the linear weights $\mathbf{u}_{k,\ell}$ and shows that they properly collapse. As a consequence, $h^*_{W,U}$ generalizes perfectly (100% accurate) even with only $n_{\text{spl}} = 5$ sample per class. Normalization plays a crucial role by ensuring that word representations do not depend upon word frequency. In turn, this prevents the undesired mechanism that causes $h_{W,U}$ to have uncollapsed linear weights $\mathbf{u}_{k,\ell}$ when trained on the small training set. Theorem 3 in the next section proves the correctness of this picture. The weights of the network $h^*$ collapse to the 'frequency independent' configuration of figure 4(c) in the large sample limit.

## 3 Theory

Our main contributions consist in three theorems. In theorem 1 we prove that the weights of the network $h_{W,U}$ collapse into the configurations depicted on figure 3 when words have identical frequencies. In theorem 2 we provide theoretical justification of the fact that, when words have distinct frequencies, the word embeddings of $h_{W,U}$ must depend on frequency in the manner that figure 4(a) illustrates. Finally, in theorem 3 we show that the weights of the network $h^*_{W,U}$ exhibit full collapse even when words have distinct frequencies. Each of these theorems hold in the large $n_{\text{spl}}$ limit and under some symmetry assumptions on the latent variables. All proofs are in the appendix.

**Notation.** The set of concepts, which up to now was $\mathcal{C} = \{\text{veggie, dairy, meat}\}$, will be represented in this section by the more abstract $\mathcal{C} = \{1, \ldots, n_c\}$. We let $s_c := n_w/n_c$ denote the number of words per concept, and represent the vocabulary by

$$\mathcal{V} = \left\{ (\alpha, \beta) \in \mathbb{N}^2 \,:\, 1 \leq \alpha \leq n_c \text{ and } 1 \leq \beta \leq s_c \right\}$$

So elements of $\mathcal{V}$ are tuples of the form $(\alpha, \beta)$ with $1 \leq \alpha \leq n_c$ and $1 \leq \beta \leq s_c$, and we think of the tuple $(\alpha, \beta)$ as representing the $\beta^{\text{th}}$ word of the $\alpha^{\text{th}}$ concept. Each concept $\alpha \in \mathcal{C}$ comes equipped with a probability distribution $p_\alpha : \{1, \ldots, s_c\} \to [0, 1]$ over the words within it, so that $p_\alpha(\beta)$ is the probability of selecting the $\beta^{\text{th}}$ word when sampling out of the $\alpha^{\text{th}}$ concept. For simplicity we assume that the word distributions within each concept follow identical laws, so that

$$p_\alpha(\beta) = \mu_\beta \quad \text{for all } (\alpha, \beta) \in \mathcal{V}$$

for some positive scalars $\mu_\beta > 0$ that sum to 1. We think of $\mu_\beta$ as being the 'frequency' of word $(\alpha, \beta)$ in the vocabulary. For example, choosing $\mu_\beta = 1/s_c$ gives uniform word distributions while $\mu_\beta \propto 1/\beta$ corresponds to Zipf's law. We use $\mathcal{X} := \mathcal{V}^L$ to denote the data space and $\mathcal{Z} := \mathcal{C}^L$ to denote the latent space. The elements of the data space $\mathcal{X}$ correspond to sequences $\mathbf{x} = [(\alpha_1, \beta_1), \ldots, (\alpha_L, \beta_L)]$ of $L$ words, while elements of the latent space $\mathcal{Z}$ correspond to sequences $\mathbf{z} = [\alpha_1, \ldots, \alpha_L]$ of $L$ concepts. For a given latent variable $\mathbf{z}$ we write $\mathbf{x} \sim \mathcal{D}_\mathbf{z}$ to indicate that the data point $\mathbf{x}$ was generated by $\mathbf{z}$ (formally $\mathcal{D}_\mathbf{z} : \mathcal{X} \to [0, 1]$ is a probability distribution).

**Word embeddings, LayerNorm, and word representations.** We use $\mathbf{w}_{(\alpha, \beta)} \in \mathbb{R}^d$ to denote the *embedding* of word $(\alpha, \beta) \in \mathcal{V}$. The collection of all $\mathbf{w}_{(\alpha, \beta)}$ determines the columns of the matrix $W \in \mathbb{R}^{d \times n_w}$. These embeddings feed into a LayerNorm module *without learnable parameters*:

$$\varphi(\mathbf{v}) = \frac{\mathbf{v} - \text{mean}(\mathbf{v})\mathbf{1}_d}{\sigma(\mathbf{v})} \quad \text{where} \quad \text{mean}(\mathbf{v}) = \frac{1}{d}\sum_{i=1}^d v_i \quad \text{and} \quad \sigma^2(\mathbf{v}) = \frac{1}{d}\sum_{i=1}^d \left(v_i - \text{mean}(\mathbf{v})\right)^2.$$

So the LayerNorm module converts a word embedding $\mathbf{w}_{(\alpha, \beta)} \in \mathbb{R}^d$ into a vector $\varphi(\mathbf{w}_{(\alpha, \beta)}) \in \mathbb{R}^d$, and we call this vector a *word representation*.

**Equiangular vectors.** We call a collection of $n_c$ vectors $\mathfrak{f}_1, \ldots, \mathfrak{f}_{n_c} \in \mathbb{R}^d$ *equiangular* if

$$\sum_{\alpha=1}^{n_c} \mathfrak{f}_\alpha = 0 \qquad \text{and} \qquad \langle \mathfrak{f}_\alpha, \mathfrak{f}_{\alpha'} \rangle = \begin{cases} 1 & \text{if } \alpha = \alpha' \\ -1/(n_c - 1) & \text{otherwise} \end{cases} \qquad (4)$$

hold for all possible pairs $\alpha, \alpha' \in [n_c]$ of concepts. For example, three vectors $\mathfrak{f}_1, \mathfrak{f}_2, \mathfrak{f}_3 \in \mathbb{R}^{100}$ are equiangular exactly when they have unit norms, live in a two dimensional subspace of $\mathbb{R}^{100}$, and form the vertices of an equilateral triangle in this subspace. This example exactly corresponds to the configurations in figure 3 and 4 (up to a scaling factor). Similarly, four vectors $\mathfrak{f}_1, \mathfrak{f}_2, \mathfrak{f}_3, \mathfrak{f}_4 \in \mathbb{R}^{100}$ are equiangular when they have unit norms and form the vertices of a regular tetrahedron. We will sometimes require $\mathfrak{f}_1, \ldots, \mathfrak{f}_{n_c} \in \mathbb{R}^d$ to also satisfy $\langle \mathfrak{f}_\alpha, \mathbf{1}_d \rangle = 0$ for all $\alpha \in [n_c]$, in which case we say $\mathfrak{f}_1, \ldots, \mathfrak{f}_{n_c} \in \mathbb{R}^d$ form a collection of *mean-zero equiangular vectors*.

**Collapse configurations.** Our empirical investigations reveal two distinct candidate solutions for the weights $(W, U)$ of the network $h_{W, U}$ and $h_{W, U}^*$. We therefore isolate each of these possible candidates as a definition before turning to the statements of our main theorems. We begin by defining the type of collapse observed when training the network $h_{W, U}$ with uniform word distributions (see figure 3 for a visual illustration of this type of collapse).

**Definition 1** (Type-I Collapse). *The weights $(W, U)$ of the network $h_{W, U}$ form a type-I collapse configuration if and only if the conditions*

    i) *There exists $c \geq 0$ so that $\mathbf{w}_{(\alpha, \beta)} = c\,\mathfrak{f}_\alpha$ for all $(\alpha, \beta) \in \mathcal{V}$.*

    ii) *There exists $c' \geq 0$ so that $\mathbf{u}_{k, \ell} = c'\,\mathfrak{f}_\alpha$ for all $(k, \ell)$ satisfying $z_{k, \ell} = \alpha$ and all $\alpha \in \mathcal{C}$.*

*hold for some collection $\mathfrak{f}_1, \ldots, \mathfrak{f}_{n_c} \in \mathbb{R}^d$ of equiangular vectors.*

Recall that the network $h_{W, U}^*$ exhibits collapse as well, up to the fact that the word representations $\varphi(\mathbf{w}_{\alpha, \beta})$ collapse rather than the word embeddings themselves. Additionally, the LayerNorm also fixes the magnitude of the word representations. We isolate these differences in the next definition.

**Definition 2** (Type-II Collapse). *The weights $(W, U)$ of the network $h_{W, U}^*$ form a type-II collapse configuration if and only if the conditions*

    i) *$\varphi(\mathbf{w}_{(\alpha, \beta)}) = \sqrt{d}\,\mathfrak{f}_\alpha$ for all $(\alpha, \beta) \in \mathcal{V}$.*

    ii) *There exists $c \geq 0$ so that $\mathbf{u}_{k, \ell} = c\,\mathfrak{f}_\alpha$ for all $(k, \ell)$ satisfying $z_{k, \ell} = \alpha$ and all $\alpha \in \mathcal{C}$.*

*hold for some collection $\mathfrak{f}_1, \ldots, \mathfrak{f}_{n_c} \in \mathbb{R}^d$ of mean-zero equiangular vectors.*

Finally, when training the network $h_{W,U}$ with non-uniform word distributions (c.f. figure 4(a)) we observe collapse in the direction of the word embeddings $\mathbf{w}_{(\alpha,\beta)}$ but their magnitudes depend upon word frequency. We therefore isolate this final observation as

**Definition 3** (Type-III Collapse). *The weights $(W, U)$ of the network $h_{W,U}$ form a type-III collapse configuration if and only if*

   i) *There exists positive scalars $r_\beta \geq 0$ so that $\mathbf{w}_{(\alpha,\beta)} = r_\beta \mathfrak{f}_\alpha$  for all $(\alpha, \beta) \in \mathcal{V}$.*

   ii) *There exists $c \geq 0$ so that $\mathbf{u}_{k,\ell} = c \mathfrak{f}_\alpha$  for all $(k,\ell)$ satisfying $z_{k,\ell} = \alpha$ and all $\alpha \in \mathcal{C}$.*

*hold for some collection $\mathfrak{f}_1, \ldots, \mathfrak{f}_{n_c} \in \mathbb{R}^d$ of equiangular vectors.*

In a type-III collapse we allow the word embedding $\mathbf{w}_{(\alpha,\beta)}$ to have a frequency-dependent magnitude $r_\beta$ while in type-I collapse we force all embeddings to have the same magnitude; this makes type-I collapse a special case of type-III collapse, but not vice-versa.

### 3.1 PROVING COLLAPSE

Our first result proves that the words embeddings $\mathbf{w}_{(\alpha,\beta)}$ and linear weights $\mathbf{u}_{k,\ell}$ exhibit type-I collapse in an appropriate large-sample limit. When turning from experiment (c.f. figure 3) to theory we study the true risk

$$\mathcal{R}(W, U) = \frac{1}{K} \sum_{k=1}^{K} \mathbb{E}_{\mathbf{x} \sim \mathcal{D}_{\mathbf{z}_k}} \left[ \ell(h_{W,U}(\mathbf{x}), k) \right] + \frac{\lambda}{2} \left( \|W\|_F^2 + \|U\|_F^2 \right) \tag{5}$$

rather than the empirical risk $\mathcal{R}_{\mathrm{emp}}(W, U)$ and place a symmetry assumption on the latent variables.

**Assumption 1** (Latent Symmetry). *For every $k \in [K]$, $r \in [L]$, $\ell \in [L]$, and $\alpha \in [n_c]$ the identities*

$$\left| \left\{ k' \in [K] : \mathrm{dist}(\mathbf{z}_k, \mathbf{z}_{k'}) = r \text{ and } z_{k',\ell} = \alpha \right\} \right| = \begin{cases} \frac{K}{|\mathcal{Z}|} \binom{L-1}{r} (n_c - 1)^r & \text{if } z_{k,\ell} = \alpha \\ \frac{K}{|\mathcal{Z}|} \binom{L-1}{r-1} (n_c - 1)^{r-1} & \text{if } z_{k,\ell} \neq \alpha \end{cases} \tag{6}$$

*hold, with $\mathrm{dist}(\mathbf{z}_k, \mathbf{z}_{k'})$ denoting the Hamming distance between a pair $(\mathbf{z}_k, \mathbf{z}_{k'})$ of latent variables.*

With this assumption in hand we may state our first main result

**Theorem 1** (Full Collapse of $h$). *Assume uniform sampling $\mu_\beta = 1/s_c$ for each word distribution. Let $\tau \geq 0$ denote the unique minimizer of the strictly convex function $H(t) :=$ $\log\left( 1 - \frac{K}{n_c^L} + \frac{K}{n_c^L} \left( 1 + (n_c - 1)e^{-\eta t} \right)^L \right) + \lambda t$ where $\eta = \frac{n_c}{n_c - 1} \frac{1}{\sqrt{n_w K L}}$. Assume $\mathbf{z}_1, \ldots, \mathbf{z}_K$ are mutually distinct and satisfy the symmetry assumption 1. Then any $(W, U)$ in a type-I collapse configuration with constants $c = \sqrt{\tau/n_w}$ and $c' = \sqrt{\tau/(KL)}$ is a global minimizer of (5).*

We also prove two strengthenings of this theorem in the appendix. First, under an additional technical assumption on the latent variables $\mathbf{z}_1, \ldots, \mathbf{z}_K$ we prove its converse; any $(W, U)$ that minimizes (5) must be in a type-I collapse configuration (with the same constants $c, c'$). This additional assumption is mild but technical, so we state it in appendix C. We also prove that if $d > n_w$ then $\mathcal{R}(W, U)$ does not have spurious local minimizers; all local minimizers are global (see appendix H).

The symmetry assumption, while odd at a first glance, is both needed and natural. Indeed, a type-I collapse configuration is highly symmetric and perfectly homogeneous. We therefore expect that such configurations could only solve an analogously 'symmetric' and 'homogeneous' optimization problem. In our case this means using the true risk (5) rather than the empirical risk (2), and imposing that the latent variables satisfy the symmetry assumption. This assumption means that all latent variables play interchangeable roles, or at an intuitive level, that there is no 'preferred' latent variable. To understand this better, consider the extreme case $K = n_c^L$ and $\{\mathbf{z}_1, \ldots, \mathbf{z}_K\} = \mathcal{Z}$, meaning that all latent variables in $\mathcal{Z}$ are involved in the task. The identity (6) then holds by simple combinatorics. We may therefore think of (6) as an equality that holds in the large $K$ limit, so it is neither impossible nor unnatural. We refer to appendix C for more discussion about assumption 1.

While theorem 1 proves global optimality of type-I collapse configurations in the limit of large $n_{\mathrm{spl}}$ and large $K$, these solutions still provide valuable predictions when $K$ and $n_{\mathrm{spl}}$ have small to

moderate values. For example, in the setting of figure 3 ($n_{\text{spl}} = 5$ and $K = 1000$) the theorem predicts that word embeddings should have a norm $c = \sqrt{\tau/n_w} = 1.42214$ (with $\tau$ obtained by minimizing $H(t)$ numerically). By experiment we find that, on average, word embeddings have norm 1.41 with standard deviation 0.13. To take another example, when $K = 50$ and $n_{\text{spl}} = 100$ (and keeping $n_c = 3$, $n_w = 1200$, $L = 15$) the theorem predicts that words embeddings should have norm 0.61602. This compares well against the values $0.61 \pm 0.06$ observed in experiments. The idealized solutions of the theorem capture their empirical counterparts very well.

For non-uniform $\mu_\beta$ we expect $h_{W,U}$ to exhibit type-III collapse rather than type-I collapse. Additionally, in our long-tail experiments, we observe that frequent words (i.e. large $\mu_\beta$) receive large embeddings. We now prove that this is the case in our next theorem. To state it, consider the following system of $s_c + 1$ equations

$$\frac{\lambda}{L} \frac{r_\beta}{c} \left( n_c - 1 + \exp\left( \frac{n_c}{n_c-1} c \, r_\beta \right) \right) = \mu_\beta \qquad \text{for all } 1 \leq \beta \leq s_c \tag{7}$$

$$\sum_{\beta=1}^{s_c} \left( \frac{r_\beta}{c} \right)^2 = L n_c^{L-1} \tag{8}$$

for the unknowns $(c, r_1, \ldots, r_{s_c})$. If the regularization parameter $\lambda$ is small enough, namely $\lambda^2 < \frac{L}{n_c^{L+1}} \sum_{\beta=1}^{s_c} \mu_\beta^2$, then (7)–(8) has a unique solution. This solution defines the magnitudes of the word embeddings. The left hand side of (7) is an increasing function of $r_\beta$, so $\mu_\beta < \mu_{\beta'}$ implies $r_\beta < r_{\beta'}$ and more frequent words receive larger embeddings.

**Theorem 2** (Directional Collapse of $h$). *Assume $\lambda^2 < (L/n_c^{L+1}) \sum_{\beta=1}^{s_c} \mu_\beta^2$, $K = n_c^L$ and $\{\mathbf{z}_1, \ldots, \mathbf{z}_K\} = \mathcal{Z}$. Suppose $(W, U)$ is in a type-III collapse configuration for some constants $(c, r_1, \ldots, r_{s_c})$. Then $(W, U)$ is a critical point of the true risk (5) if and only if $(c, r_1, \ldots, r_{s_c})$ solve the system (7)–(8).*

Essentially this theorem shows that word embeddings *must* depend on word frequency and so feature collapse fails. Even in the fully-sampled case $K = n_c^L$ and $\{\mathbf{z}_1, \ldots, \mathbf{z}_K\} = \mathcal{Z}$ a network exhibiting type-I collapse is never critical if the word distributions are non-uniform. While we conjecture global optimality of the solutions in theorem 2 under appropriate symmetry assumptions, we have no proof of this yet. The bound on $\lambda$ is the natural one for theorem 2, for if $\lambda$ is too large the trivial solution $(W, U) = (0, 0)$ is the only one. In our experiments, $\lambda$ satisfies this bound.

Our final theorem completes the picture; it shows that normalization restores global optimality of fully-collapsed configurations. For the network $h_{W,U}^*$ with LayerNorm, we use the appropriate limit

$$\mathcal{R}^*(W, U) = \frac{1}{K} \sum_{k=1}^{K} \mathbb{E}_{\mathbf{x} \sim \mathcal{D}_{\mathbf{z}_k}} \left[ \ell(h_{W,U}^*(\mathbf{x}), k) \right] + \frac{\lambda}{2} \|U\|_F^2 \tag{9}$$

of the associated empirical risk and place no assumptions on the sampling distribution.

**Theorem 3** (Full Collapse of $h^*$). *Assume the non-degenerate condition $\mu_\beta > 0$ holds. Let $\tau \geq 0$ denote the unique minimizer of the strictly convex function $H^*(t) = \log\left( 1 - \frac{K}{n_c^L} + \frac{K}{n_c^L}\left( 1 + (n_c - 1)e^{-\eta^* t} \right)^L \right) + \frac{\lambda}{2} t^2$ where $\eta^* = \frac{n_c}{n_c-1} \frac{1}{\sqrt{KL/d}}$. Assume $\mathbf{z}_1, \ldots, \mathbf{z}_K$ are mutually distinct and satisfy assumption 1. Then any $(W, U)$ in a type-II collapse configuration with constant $c = \tau/\sqrt{KL}$ is a global minimizer of (9).*

As for theorem 1, we prove the converse under an additional technical assumption on the latent variables. Any $(W, U)$ that minimizes (9) must be in a type-II collapse configuration with $c = \tau/\sqrt{KL}$. The proof and exact statement can be found in section F of the appendix.

## 4 ADDITIONAL EXPERIMENTS

Our theoretical investigation of feature collapse uses a simple synthetic data model and a basic network. These simplifications allow us to rigorously prove that feature collapse occurs in this setting. The first section of the appendix provides preliminary evidence that the feature collapse phenomenon also occurs in more complex settings, which are beyond the reach of our current analytical tools. In particular, we experimentally observe feature collapse in more complex data models that involve a deeper hierarchy of latent structures. We also investigate the feature collapse phenomenon in transformer architectures in both a classification setup and the usual next-word-prediction setup.

ACKNOWLEDGMENT

Xavier Bresson is supported by NUS Grant ID R-252-000-B97-133. The authors would like to express their gratitude to the reviewers for their feedback, which has improved the clarity and contribution of the paper.

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

# Appendix

In section A we conduct an empirical investigation of the feature collapse phenomenon in settings beyond the reach of our current analytical tools. The remaining sections are all devoted to the proofs of the three theorems that constitute the main results of the paper.

Section B provides formulas for the networks $h_{W,U}$ and $h_{W,U}^*$ depicted on figure 2 of the main paper, and formula for the distribution $\mathcal{D}_{\mathbf{z}_k} : \mathcal{X} \to [0,1]$ underlying the data model depicted on figure 1 of the main paper. We also use this section to introduce various notations that our proofs will rely on.

Section C is devoted to the symmetry assumptions that we impose on the latent variables. We start with an in depth discussion of assumption 1. from the main paper. This assumption is required for theorem 1 and 3 to hold. We then present and discuss an additional technical assumption on the latent variables (c.f. assumption B) that we will use to prove the *converse* of theorems 1 and 3.

Whereas sections B and C are essentially devoted to notations and discussions, most of the analysis occurs in section D, E, F and G. We start by deriving a sharp lower bound for the unregularized risk in section D. Theorem 1 from the main paper, as well as its converse, are proven in section E. Theorem 3 and its converse are proven in section F. Finally we prove theorem 2 in section G.

We conclude this appendix by proving in section H that if $d > \min(n_w, KL)$, then the risk associated to the network $h_{W,U}$ does not have spurious local minimizers; all local minimizers are global. This proof follows the same strategy that was used in Zhu et al. (2021).

## A  FURTHER EMPIRICAL INVESTIGATIONS

In this section, show empirically that the feature collapse phenomenon is not limited to the simple controlled setting where we were able to prove it. In particular, we show that feature collapse occurs in the absence of weight decay and when the LayerNorm has learnable parameters. We also show that feature collapse occurs in transformer architectures, and also when the classification task is replaced with a language modeling task. Finally, we show that feature collapse occurs in data models involving a deeper hierarchy of latent structures, such as a Context Free Grammar (CFG).

### A.1  EXPERIMENTS WITHOUT WEIGHT DECAY

We start by reproducing the experiments depicted in Figure 3 and 4 but without weight decay. Specifically, we set $\lambda = 0$ in equations (2) and (3). All other parameters defining the networks and data model remain the same. To train the networks, we perform 5 million iterations of stochastic gradient descent with a batch size of 100 and a learning rate of 0.1. After training, the empirical losses for all networks are below $10^{-4}$. The outcomes of these experiments are depicted in Figures 5 and 6. These figures are virtually identical to Figures 3 and 4 in the main paper. In other words, the absence or presence of weight decay does not affect our main qualitative findings.

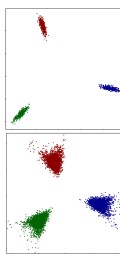

Figure 5

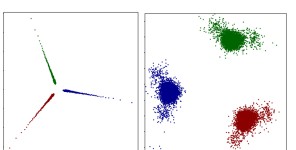

(a) $h$ trained on the **large** training set. Test acc. $= 100\%$

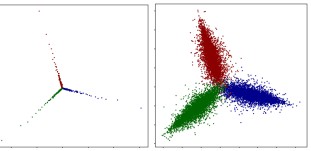

(b) $h$ trained on the **small** training set. Test acc. $= 66\%$

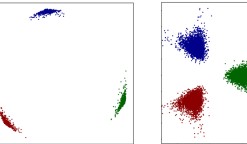

(c) $h^*$ trained on the **small** training set. Test acc. $= 100\%$

Figure 6: Same experiments as in Figure 4 but with no weight decay. Note that the result are qualitatively similar.

## A.2 LAYERNORM WITH LEARNABLE PARAMETERS

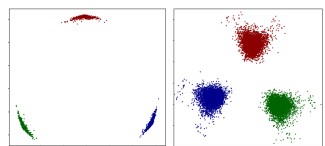

When the LayerNorm module has no learnable weights, the word embeddings must lie on a sphere of constant radius. This constraint aids collapse. A natural question is whether collapse still occurs when the LayerNorm has learnable parameters. To answer this question, we reproduce the experiment that corresponds to Figure 4(c) but allow the LayerNorm module to have learnable weights. The result of this experiment is depicted on Figure 7, and one can clearly observe that feature collapse does occur in this setting as well.

Figure 7: Same experiment as in Figure 4(c) but with learnable weight in the LayerNorm

## A.3 CLASSIFICATION EXPERIMENTS WITH TRANSFORMERS

In this set of experiments, we train a transformer on the classification task depicted on figure 1. The transformer has 2 layers, 8 heads, 512 dimensions, and we use absolute positional embeddings (as in GPT-2). A classification token is appended to each input sentence, and this classification token is used in the last layer to predict the category. The network is trained with AdamW (constant learning rate of $10^{-4}$, weight decay of 0.1, $\beta_1 = 0.9$ and $\beta_2 = 0.95$) during 3 epochs on a training sets containing 0.5 million sentences. For the data model, we use $n_c = 3$, $n_w = 1200$, $L = 15$, $K = 1000$ as in the main paper. In figure 8(a) and 8 (b) we display the word embeddings via dimensionality reduction and color coding. These are the word embeddings obtained before addition of the positional embeddings, and before going through the first transformer layer. Figure 8(a) corresponds to the case in which the words are uniformly distributed, and Figure 8(b) corresponds to the long-tail case. We observe that the word embeddings, in both the uniform and long-tail case, are properly collapsed.

## A.4 LANGUAGE MODELING EXPERIMENTS WITH TRANSFORMERS

In this set of experiments, we train a transformer to predict the next token on sentences generated by the data model depicted on Figure 1. We use the GPT-2 architecture (Radford et al., 2019) with 2 layers, 8 heads, and 512 dimensions. The training set contains 1 million sentences generated by our data model with parameters $n_c = 3$, $n_w = 1200$, $L = 15$, $K = 1000$, and with uniform word distributions. We perform a single epoch through the training set and use AdamW with same parameters as above. On figure 8 (c) we display the word embeddings via dimensionality reduction and color coding, and we observe that they are they are properly collapsed.

## A.5 EXPERIMENTS WITH CONTEXT FREE GRAMMAR

The data model presented in the paper extends to one with a deeper hierarchy of latent structures. Recall that words are partitioned into concepts and that the latent variables are sequences of concepts. We can further partitioned the latent variables into 'meta-concepts' and create deeper latent variables that are sequences of 'meta-concepts'. We can iterate this process to obtain a hierarchy of any depth. Such a data model is a particular instance of a Context Free Grammar (Chomsky, 1956), which generates sentences with probabilistic trees and are widely used to understand natural

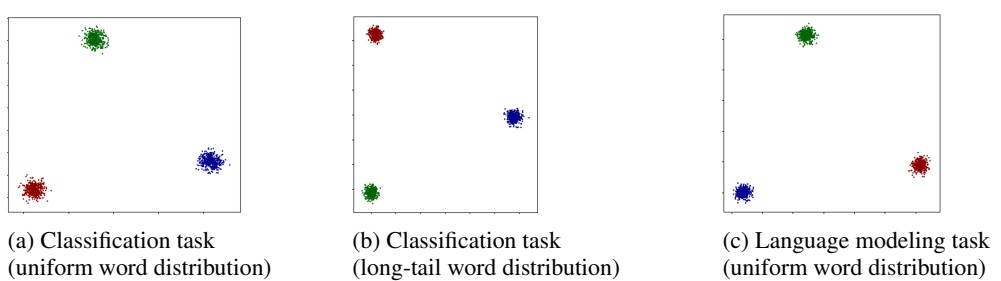

(a) Classification task
(uniform word distribution)

(b) Classification task
(long-tail word distribution)

(c) Language modeling task
(uniform word distribution)

Figure 8: Experiments with transformers.

language models (e.g. Kim et al. (2019); Allen-Zhu & Li (2023); Liu et al. (2023)). In Figure 9 we provide an illustration of a simple depth 3 context free grammar.

We ran experiments with a context free grammar of depth 4, meaning that we have words, concepts, meta-concepts and meta-meta-concepts. We used a deep neural network with ReLU nonlinearities and LayerNorm module at each layer. The architecture of the neural network was chosen to match that of the context free grammar, see Figure 10. In Figure 11 we plot the activations after each of the three hidden layers and readily observe the expected feature collapse phenomenon. All segments of the input sentence that correspond to same concept, meta-concept, or meta-meta-concept receive the same representations in the appropriate layer of the network (layer 1 for concepts, layer 2 for meta-concepts, and layer 3 for meta-meta-concepts). This shows that the feature collapse phenomenon is a general one.

**Details of the Context Free Grammar:** We used the following context free grammar for our experiment. We choose $K = 100$ categories. Each category generates a sequence of 'meta-meta-concepts' of length 8 by choosing uniformly at random among 5 possible sequences of meta-meta-concepts. Each meta-meta-concept then generates a sequence of meta-concepts of length 8, again by choosing uniformly at random among 5 possible sequences of meta-concepts. Each meta-concept then generates a sequence of concepts of length 8 by choosing among 5 possible sequences of concepts. Finally, each concept generates a sequence of words of length 8 by choosing among 5 possible sequences of words. At level 0 the sequences of words have an overall length of $8^4 = 4096$.

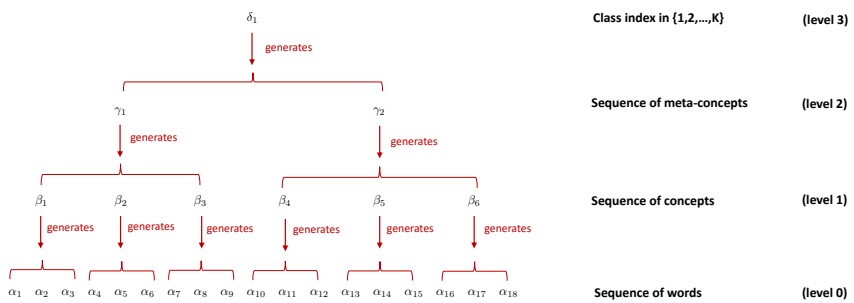

Figure 9: Probabilistic Context Free Grammar of depth 3. The class index generates a sequence of meta-concepts. Each meta-concept further generates a sequence of concepts. Finally each concept generates a sequence of words. The process by which a token from one level generates a sequence of tokens in the level below is random. For example, the meta-concept $\gamma_2 = 5$ might generate the sequence of concepts $[\beta_4, \beta_5, \beta_6] = [4, 1, 3]$ with probability $1/3$, the sequence $[\beta_4, \beta_5, \beta_6] = [2, 5, 3]$ with probability $1/3$, and the sequence $[\beta_4, \beta_5, \beta_6] = [3, 5, 5]$ with probability $1/3$.

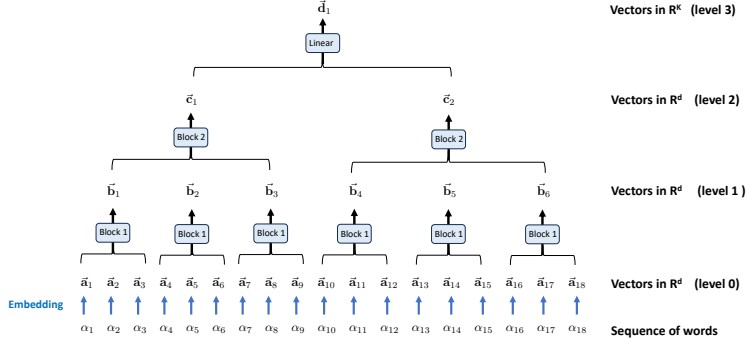

Figure 10: Neural network architecture matching the context free grammar from Figure 9. Each block consists in a MLP followed by layer normalization. The vectors are concatenated before being fed to a block. If feature collapse occurs, then two sequences of words $[\alpha_1, \alpha_2, \alpha_3]$ generated by the same concept $\beta_1$ should have almost identical representation $\vec{\mathbf{b}}_1$ in level 1 of the network. Similarly, two sequences of words $[\alpha_1, \alpha_2, \ldots, \alpha_9]$ generated by the same meta-concept $\gamma_1$ should have almost identical representation $\vec{\mathbf{c}}_1$ in level 2 of the network.

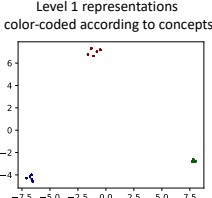 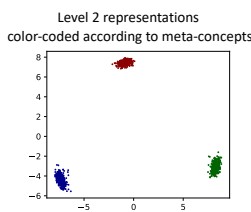 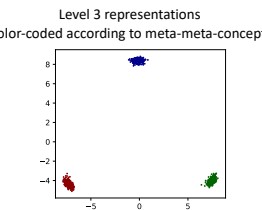

Figure 11: Results of an experiment ran on a context free grammar of depth 4 (so deeper than the one depicted on Figure 1). At every level, each token generates in the level below a sequence of length 8 chosen uniformly at random among 5 possible sequences. After training the network, we generate 1000 test sequences, feed them to the network, and visualize via PCA the representations obtained at each layer (i.e. we plot the vectors $\vec{\mathbf{b}}_i$, $\vec{\mathbf{c}}_i$ and $\vec{\mathbf{d}}_i$). The vectors are color coded according to the concept, meta-concept, and meta-meta-concept that generated them, and we keep only representations corresponding to the first 3 concepts, meta-concepts, and meta-meta-concepts. We clearly observe the feature collapse phenomenon. Note that since we are dealing with a network of depth 4, the vectors $\vec{\mathbf{d}}_i$ are not the output, but the last hidden representations.

# B  PRELIMINARIES AND NOTATIONS

## B.1  FORMULA FOR THE NEURAL NETWORKS

Recall that the vocabulary is the set

$$\mathcal{V} = \{(\alpha, \beta) \in \mathbb{N}^2 : 1 \le \alpha \le n_c \text{ and } 1 \le \beta \le s_c\},$$

and that we think of the tuple $(\alpha, \beta) \in \mathcal{V}$ as representing the $\beta^{th}$ word of the $\alpha^{th}$ concept. The data space is $\mathcal{X} = \mathcal{V}^L$, and a sentence $\mathbf{x} \in \mathcal{X}$ is a sequence of $L$ words:

$$\mathbf{x} = [(\alpha_1, \beta_1), \dots, (\alpha_L, \beta_L)] \qquad 1 \le \alpha_\ell \le n_c \text{ and } 1 \le \beta_\ell \le s_c.$$

The two neural networks $h, h^*$ studied in this work process such a sentence $\mathbf{x} \in \mathcal{X}$ in multiple steps:

1. Each word $(\alpha_\ell, \beta_\ell)$ of the sentence is encoded into a one-hot vector.

2. These one-hot vectors are multiplied by a matrix $W$ to produce word embeddings that live in a $d$-dimensional space.

3. Optionally (i.e. in the case of the network $h^*$), these word embeddings go through a LayerNorm module without learnable parameters.

4. The word embeddings are concatenated and then goes through a linear transformation $U$.

We now formalize these 4 steps, and in the process, we set the notations on which we will rely in all our proofs.

**Step 1: One-hot encoding.**  Without loss of generality, we choose the following one-hot encoding scheme: word $(\alpha, \beta) \in \mathcal{V}$ receives the one-hot vector which has a 1 in entry $(\alpha - 1)s_c + \beta$ and 0 everywhere else. To formalize this, we define the one-hot encoding function

$$\zeta(\alpha, \beta) = \mathbf{e}_{(\alpha-1)s_c+\beta} \tag{10}$$

where $\mathbf{e}_i$ denotes the $i^{th}$ basis vector of $\mathbb{R}^{n_w}$. The one-hot encoding function $\zeta$ can also be applied to a sequence of words. Given a sentence $\mathbf{x} = [(\alpha_1, \beta_1), \dots, (\alpha_L, \beta_L)] \in \mathcal{X}$ we let

$$\zeta(\mathbf{x}) := \begin{bmatrix} | & | & & | \\ \zeta(\alpha_1, \beta_1) & \zeta(\alpha_2, \beta_2) & \dots & \zeta(\alpha_L, \beta_L) \\ | & | & & | \end{bmatrix} \in \mathbb{R}^{n_w \times L} \tag{11}$$

and so $\zeta$ maps sentences to $n_w \times L$ matrices.

**Step 2: Embedding.** The embedding matrix $W$ has $n_w$ columns and each of these columns belongs to $\mathbb{R}^d$. Since $\zeta(\alpha, \beta)$ denote the one-hot vector associated to word $(\alpha, \beta) \in \mathcal{V}$, we define the *embedding* of word $(\alpha, \beta)$ by

$$\mathbf{w}_{(\alpha,\beta)} := W\,\zeta(\alpha, \beta) \ \in \mathbb{R}^d. \tag{12}$$

Due to (10), this means that $\mathbf{w}_{(\alpha,\beta)}$ is the $j^{th}$ column of $W$, where $j = (\alpha - 1)s_c + \beta$. The embedding matrix $W$ can therefore be visualized as follow (for concreteness we choose $n_c = 3$ and $n_w = 12$ as in figure 1 of the main paper):

$$W = \begin{bmatrix} | & | & | & | & | & | & | & | & | & | & | & | \\ \mathbf{w}_{(1,1)} & \mathbf{w}_{(1,2)} & \mathbf{w}_{(1,3)} & \mathbf{w}_{(1,4)} & \mathbf{w}_{(2,1)} & \mathbf{w}_{(2,2)} & \mathbf{w}_{(2,3)} & \mathbf{w}_{(2,4)} & \mathbf{w}_{(3,1)} & \mathbf{w}_{(3,2)} & \mathbf{w}_{(3,3)} & \mathbf{w}_{(3,4)} \\ | & | & | & | & | & | & | & | & | & | & | & | \end{bmatrix}$$

$$\underbrace{\hphantom{xxxxxxxxxxxx}}_{\text{Embeddings of the words in the } 1^{st} \text{ concept.}} \quad \underbrace{\hphantom{xxxxxxxxxxxx}}_{\text{Embeddings of the words in the } 2^{nd} \text{ concept.}} \quad \underbrace{\hphantom{xxxxxxxxxxxx}}_{\text{Embeddings of the words in the } 3^{rd} \text{ concept.}}$$

Given a sentence $\mathbf{x} = [(\alpha_1, \beta_1), \ldots, (\alpha_L, \beta_L)] \in \mathcal{X}$, appealing to (11) and (12), we find that

$$W\zeta(\mathbf{x}) = \begin{bmatrix} | & | & & | \\ \mathbf{w}_{(\alpha_1,\beta_1)} & \mathbf{w}_{(\alpha_2,\beta_2)} & \cdots & \mathbf{w}_{(\alpha_L,\beta_L)} \\ | & | & & | \end{bmatrix} \in \mathbb{R}^{d \times L} \tag{13}$$

and therefore $W\zeta(\mathbf{x})$ is the matrix that contains the $d$-dimensional embeddings of the words that constitute the sentence $\mathbf{x} \in \mathcal{X}$.

**Step 3: LayerNorm.** Recall from the main paper that the LayerNorm function $\varphi : \mathbb{R}^d \to \mathbb{R}^d$ is defined by

$$\varphi(\mathbf{v}) = \frac{\mathbf{v} - \text{mean}(\mathbf{v})\mathbf{1}_d}{\sigma(\mathbf{v})} \quad \text{where} \quad \text{mean}(\mathbf{v}) = \frac{1}{d}\sum_{i=1}^d v_i \quad \text{and} \quad \sigma^2(\mathbf{v}) = \frac{1}{d}\sum_{i=1}^d \left(v_i - \text{mean}(\mathbf{v})\right)^2,$$

We will often apply this function column-wise to a matrix. For example if $V$ is the $d \times m$ matrix

$$V = \begin{bmatrix} | & | & & | \\ \mathbf{v}_1 & \mathbf{v}_2 & \cdots & \mathbf{v}_m \\ | & | & & | \end{bmatrix}, \quad \text{then} \quad \varphi(V) = \begin{bmatrix} | & | & & | \\ \varphi(\mathbf{v}_1) & \varphi(\mathbf{v}_2) & \cdots & \varphi(\mathbf{v}_m) \\ | & | & & | \end{bmatrix}$$

Applying $\varphi$ to (13) gives

$$\varphi\left(W\zeta(\mathbf{x})\right) = \begin{bmatrix} | & | & & | \\ \varphi\left(\mathbf{w}_{(\alpha_1,\beta_1)}\right) & \varphi\left(\mathbf{w}_{(\alpha_2,\beta_2)}\right) & \cdots & \varphi\left(\mathbf{w}_{(\alpha_L,\beta_L)}\right) \\ | & | & & | \end{bmatrix} \in \mathbb{R}^{d \times L} \tag{14}$$

and so $\varphi\left(W\zeta(\mathbf{x})\right)$ contains the *word representations* of the words from the input sentence (recall that by word representations we mean the word embeddings *after* the LayerNorm).

**Step 4: Linear Transformation.** Recall from the main paper that

$$U = \begin{bmatrix} -\mathbf{u}_{1,1}- & -\mathbf{u}_{1,2}- & \cdots & -\mathbf{u}_{1,L}- \\ -\mathbf{u}_{2,1}- & -\mathbf{u}_{2,2}- & \cdots & -\mathbf{u}_{2,L}- \\ \vdots & \vdots & & \vdots \\ -\mathbf{u}_{K,1}- & -\mathbf{u}_{K,2}- & \cdots & -\mathbf{u}_{K,L}- \end{bmatrix} \in \mathbb{R}^{K \times Ld} \tag{15}$$

where each vector $\mathbf{u}_{k,\ell}$ belongs to $\mathbb{R}^d$. The neural networks $h_{W,U}$ and $h^*_{W,U}$ are then given by the formula

$$h_{W,U}(\mathbf{x}) = U\,\text{Vec}\left[W\zeta(\mathbf{x})\right] \tag{16}$$

$$h^*_{W,U}(\mathbf{x}) = U\,\text{Vec}\left[\varphi\left(W\zeta(\mathbf{x})\right)\right] \tag{17}$$

where $\text{Vec} : \mathbb{R}^{d \times L} \to \mathbb{R}^{dL}$ is the function that takes as input a $d \times L$ matrix and flatten it out into a vector with $dL$ entries (with the first column filling the first $d$ entries of the vector, the second column filling the next $d$ entries, and so forth). It will prove convenient to gather the $L$ vectors $\mathbf{u}_{k,\ell}$ that constitute the $k^{th}$ row of $U$ into the matrix

$$\hat{U}_k = \begin{bmatrix} | & | & & | \\ \mathbf{u}_{k,1} & \mathbf{u}_{k,2} & \cdots & \mathbf{u}_{k,L} \\ | & | & & | \end{bmatrix} \in \mathbb{R}^{d \times L} \tag{18}$$

With this notation, we have the following alternative expressions for the networks $h_{W,U}$ and $h_{W,U}^*$

$$h_{W,U}(\mathbf{x}) = \begin{bmatrix} \left\langle \hat{U}_1 , W\zeta(\mathbf{x}) \right\rangle_F \\ \left\langle \hat{U}_2 , W\zeta(\mathbf{x}) \right\rangle_F \\ \vdots \\ \left\langle \hat{U}_K , W\zeta(\mathbf{x}) \right\rangle_F \end{bmatrix} \quad \text{and} \quad h_{W,U}^*(\mathbf{x}) = \begin{bmatrix} \left\langle \hat{U}_1 , \varphi\left(W\zeta(\mathbf{x})\right) \right\rangle_F \\ \left\langle \hat{U}_2 , \varphi\left(W\zeta(\mathbf{x})\right) \right\rangle_F \\ \vdots \\ \left\langle \hat{U}_K , \varphi\left(W\zeta(\mathbf{x})\right) \right\rangle_F \end{bmatrix} \tag{19}$$

where $\langle \cdot, \cdot \rangle_F$ denote the Frobenius inner product between matrices (see next subsection for a definition).

Finally, we use $\hat{U}$ to denote the matrix obtained by concatenating the matrices $\hat{U}_1, \ldots, \hat{U}_K$, that is

$$\hat{U} := \begin{bmatrix} \hat{U}_1 & \hat{U}_2 & \cdots & \hat{U}_K \end{bmatrix} \in \mathbb{R}^{d \times KL} \tag{20}$$

The matrix $\hat{U}$, which is nothing but a reshaped version of the original weight matrix $U \in \mathbb{R}^{K \times Ld}$, will play a crucial role in our analysis.

## B.2 BASIC PROPERTIES OF THE FROBENIUS INNER PRODUCT

We recall that the Frobenius inner product between two matrices $A, B \in \mathbb{R}^{m \times n}$ is defined by

$$\langle A, B \rangle_F = \sum_{i=1}^{m} \sum_{j=1}^{n} A_{ij} B_{ij}$$

and that the Frobenius norm of a matrix $A \in \mathbb{R}^{m \times n}$ is given by $\|A\|_F = \sqrt{\langle A, A \rangle_F}$. In the course of our proofs, we will constantly appeal to the following property of the Frobenius inner product, so we state it in a lemma once and for all.

**Lemma A.** *Suppose $A \in \mathbb{R}^{m \times n}$, $B \in \mathbb{R}^{m \times r}$ and $C \in \mathbb{R}^{r \times n}$. Then*

$$\langle A, BC \rangle_F = \left\langle B^T A, C \right\rangle_F \quad \text{and} \quad \langle A, BC \rangle_F = \left\langle AC^T, B \right\rangle_F$$

*Proof.* The Frobenius inner product can be expressed as $\langle A, B \rangle_F = \text{Tr}(A^T B)$, and so we have

$$\langle A, BC \rangle_F = \text{Tr}(A^T BC) = \text{Tr}\left(\left(B^T A\right)^T C\right) = \left\langle B^T A, C \right\rangle_F.$$

Using the cyclic property of the trace, we also get

$$\langle A, BC \rangle_F = \text{Tr}(A^T BC) = \text{Tr}(CA^T B) = \text{Tr}\left(\left(AC^T\right)^T B\right) = \left\langle AC^T, B \right\rangle_F$$

$\square$

## B.3 THE TASK, THE DATA MODEL, AND THE DISTRIBUTION $\mathcal{D}_{\mathbf{z}_k}$

Recall that $\mathcal{C} = \{1, \ldots, n_c\}$ represents the set of concepts, and that $\mathcal{Z} = \mathcal{C}^L$ is the latent space. We aim to study a classification task in which the $K$ classes are defined by $K$ latent variables

$$\mathbf{z}_1, \ldots, \mathbf{z}_k \in \mathcal{Z}$$

We write $\mathbf{x} \sim \mathcal{D}_{\mathbf{z}_k}$ to indicate that the sentence $\mathbf{x} \in \mathcal{X}$ is generated by the latent variable $\mathbf{z}_k \in \mathcal{Z}$ (see figure 1 of the main paper for a visual illustration). Formally, $\mathcal{D}_{\mathbf{z}_k}$ is a probability distribution on the data space $\mathcal{X}$, and we now give the formula for its p.d.f. First, recall that $\mu_\beta > 0$ stands for the probability of sampling the $\beta^{th}$ word of the $\alpha^{th}$ concept. Let us denote the $k^{th}$ latent variable by

$$\mathbf{z}_k = [\, z_{k,1}\, ,\, z_{k,2}\, ,\, \dots\, ,\, z_{k,L}\,] \in \mathcal{Z}$$

where $1 \leq z_{k,\ell} \leq n_c$. The probability of sampling the sentence

$$\mathbf{x} = [\,(\alpha_1, \beta_1)\, ,\, (\alpha_2, \beta_2)\, \dots\, ,\, (\alpha_L, \beta_L)\,] \in \mathcal{X}$$

according to $\mathcal{D}_{\mathbf{z}_k}$ is then given by the formula

$$\mathcal{D}_{\mathbf{z}_k}(\{\mathbf{x}\}) = \prod_{\ell=1}^{L} \mathbf{1}_{\{\alpha_\ell = z_{k,\ell}\}}\, \mu_{\beta_\ell}$$

Note that $\mathcal{D}_{\mathbf{z}_k}(\{\mathbf{x}\}) > 0$ if and only if $[z_{k,1}, \dots, z_{k,L}] = [\alpha_1, \dots, \alpha_L]$. So a sentence $\mathbf{x}$ has a non-zero probability of being generated by the latent variable $\mathbf{z}_k$ only if its words match the concepts in $\mathbf{z}_k$. If this is the case, then the probability of sampling $\mathbf{x}$ according to $\mathcal{D}_{\mathbf{z}_k}$ is simply given by the product of the frequencies of the words contained in $\mathbf{x}$.

We use $\mathcal{X}_k$ to denote the support of the distribution $\mathcal{D}_{\mathbf{z}_k}$, that is

$$\mathcal{X}_k := \{\mathbf{x} \in \mathcal{X} : \mathcal{D}_{\mathbf{z}_k}(\mathbf{x}) > 0\}$$

and we note that if the latent variables $\mathbf{z}_1, \dots, \mathbf{z}_K$ are mutually distinct, then $\mathcal{X}_j \cap \mathcal{X}_k = \emptyset$ for all $j \neq k$. Since the $K$ latent variables define the $K$ classes of our classification problem, we may alternatively define $\mathcal{X}_k$ by

$$\mathcal{X}_k = \{\mathbf{x} \in \mathcal{X} : \mathbf{x} \text{ belongs to the } k^{th} \text{ category}\}$$

To each latent variable $\mathbf{z}_k = [\, z_{k,1}\, ,\, z_{k,2}\, ,\, \dots\, ,\, z_{k,L}\,]$ we associate a matrix

$$Z_k = \begin{bmatrix} | & | & & | \\ \mathbf{e}_{z_{k,1}} & \mathbf{e}_{z_{k,2}} & \cdots & \mathbf{e}_{z_{k,L}} \\ | & | & & | \end{bmatrix} \in \mathbb{R}^{n_c \times L} \tag{21}$$

In other words, the matrix $Z_k$ provides a one-hot representation of the concepts contained in the latent variable $\mathbf{z}_k$. Concatenating the matrices $Z_1, \dots, Z_K$ gives the matrix

$$Z = \begin{bmatrix} Z_1 & Z_2 & \cdots & Z_K \end{bmatrix} \in \mathbb{R}^{n_c \times KL} \tag{22}$$

which is reminiscent of the matrix $\hat{U}$ defined by (20).

We encode the way words are partitioned into concepts into a 'partition matrix' $P \in \mathbb{R}^{n_c \times n_w}$. For example, if we have 12 words and 3 concepts, then the partition matrix is

$$P = \begin{bmatrix} 1 & 1 & 1 & 1 & 0 & 0 & 0 & 0 & 0 & 0 & 0 & 0 \\ 0 & 0 & 0 & 0 & 1 & 1 & 1 & 1 & 0 & 0 & 0 & 0 \\ 0 & 0 & 0 & 0 & 0 & 0 & 0 & 0 & 1 & 1 & 1 & 1 \end{bmatrix} \in \mathbb{R}^{n_c \times n_w}, \tag{23}$$

indicating that the first 4 words belong to concept 1, the next 4 words belongs to concept 2, and so forth. Formally, recalling that $\zeta(\alpha, \beta)$ is the the one-hot encoding of word $(\alpha, \beta) \in \mathcal{V}$, the matrix $P$ is defined the relationship

$$P\, \zeta(\alpha, \beta) = \mathbf{e}_\alpha \qquad \text{for all } (\alpha, \beta) \in \mathcal{V}. \tag{24}$$

Importantly, note that the matrix $P$ maps datapoints to their associated latent variables. Indeed, if $\mathbf{x} = [(\alpha_1, \beta_1), \dots, (\alpha_L, \beta_L)]$ is generated by the latent variable $\mathbf{z}_k$ (meaning that $\mathbf{x} \in \mathcal{X}_k$), then we have that

$$P\, \zeta(\mathbf{x}) = P \begin{bmatrix} | & | & & | \\ \zeta(\alpha_1, \beta_1) & \zeta(\alpha_2, \beta_2) & \cdots & \zeta(\alpha_L, \beta_L) \\ | & | & & | \end{bmatrix} = \begin{bmatrix} | & | & & | \\ \mathbf{e}_{\alpha_1} & \mathbf{e}_{\alpha_2} & \cdots & \mathbf{e}_{\alpha_L} \\ | & | & & | \end{bmatrix} = Z_k \tag{25}$$

where the last equality is due to definition (21) of the matrix $Z_k$.

Another important matrix for our analysis will be the matrix $Q \in \mathbb{R}^{n_c \times n_w}$. In the concrete case where we have 12 words and 3 concepts, this matrix takes the form

$$Q = \begin{bmatrix} \mu_1 & \mu_2 & \mu_3 & \mu_4 & 0 & 0 & 0 & 0 & 0 & 0 & 0 & 0 \\ 0 & 0 & 0 & 0 & \mu_1 & \mu_2 & \mu_3 & \mu_4 & 0 & 0 & 0 & 0 \\ 0 & 0 & 0 & 0 & 0 & 0 & 0 & 0 & \mu_1 & \mu_2 & \mu_3 & \mu_4 \end{bmatrix} \in \mathbb{R}^{n_c \times n_w} \tag{26}$$

and, in general, it is defined by the relationship

$$Q\, \zeta(\alpha, \beta) = \mu_\beta\, \mathbf{e}_\alpha \qquad \text{for all } (\alpha, \beta) \in \mathcal{V}. \tag{27}$$

## C    SYMMETRY ASSUMPTIONS ON THE LATENT VARIABLES

In subsection C.1 we provide an in depth discussion of the symmetry assumption required for theorems 1 and 3 to hold. In subsection C.2 we present and discuss the assumption that will be needed to prove the *converse* of theorems 1 and 3.

### C.1    SYMMETRY ASSUMPTION NEEDED FOR THEOREM 1 AND 3

To better understand the symmetry assumption 1 from the main paper, let us start by considering the extreme case

$$K = n_c^L \qquad \text{and} \qquad \{\mathbf{z}_1, \mathbf{z}_2, \ldots, \mathbf{z}_K\} = \mathcal{Z}, \tag{28}$$

meaning that $\mathbf{z}_1, \ldots, \mathbf{z}_K$ are mutually distinct and represent all possible latent variables in $\mathcal{Z}$. In this case, we easily obtain the formula

$$\left|\left\{j \in [K] : \text{dist}(\mathbf{z}_j, \mathbf{z}_1) = r \text{ and } z_{j,L} = z_{1,L}\right\}\right| = \binom{L-1}{r}(n_c - 1)^r \tag{29}$$

where $\text{dist}(\mathbf{z}_j, \mathbf{z}_1)$ is the Hamming distance between the latent variables $\mathbf{z}_j$ and $\mathbf{z}_1$. To see this, note that the left side of (29) counts the number of latent variables $\mathbf{z}_j$ that *differs* from $\mathbf{z}_1$ at $r$ locations and *agrees* with $\mathbf{z}_1$ at the last location $\ell = L$. This number is clearly equal to the right side of (29) since we need to choose $r$ positions out of the first $L - 1$ positions, and then, for each chosen position $\ell$, we need to choose a concept out of the $n_c - 1$ concepts that differs from $\mathbf{z}_{1,\ell}$. A similar reasoning shows that, if $z_{1,L} \neq \alpha$, then

$$\left|\left\{j \in [K] : \text{dist}(\mathbf{z}_j, \mathbf{z}_1) = r \text{ and } z_{j,L} = \alpha\right\}\right| = \binom{L-1}{r-1}(n_c - 1)^{r-1} \tag{30}$$

where the term $\binom{L-1}{r-1}$ arises from the fact that we only need to choose $r - 1$ positions, since $\mathbf{z}_1$ and $\mathbf{z}_j$ differ in their last position $\ell = L$. Suppose now that the random variables $\mathbf{z}_1, \ldots, \mathbf{z}_K$ are selected uniformly at random from $\mathcal{Z}$, and say, for the sake of concreteness, that

$$K = \frac{1}{5} n_c^L$$

so that $\mathbf{z}_1, \ldots, \mathbf{z}_K$ represent 20% of all possible latent variables (note that $|\mathcal{Z}| = n_c^L$). Then (29) – (30) should be replaced by

$$\left|\left\{j \in [K] : \text{dist}(\mathbf{z}_j, \mathbf{z}_1) = r \text{ and } z_{j,L} = z_{1,L}\right\}\right| \approx \frac{1}{5}\binom{L-1}{r}(n_c - 1)^r \tag{31}$$

$$\left|\left\{j \in [K] : \text{dist}(\mathbf{z}_j, \mathbf{z}_1) = r \text{ and } z_{j,L} = \alpha\right\}\right| \approx \frac{1}{5}\binom{L-1}{r-1}(n_c - 1)^{r-1} \quad \text{for } \alpha \neq z_{1,L} \tag{32}$$

where the equality only holds approximatively due to the random choice of the latent variables. In the above example, we chose $\mathbf{z}_1$ as our 'reference' latent variables and we 'froze' the concept appearing in position $\ell = L$. These choices were clearly arbitrary. In general, when $K$ is large, we have

$$\left|\left\{j \in [K] : \text{dist}(\mathbf{z}_j, \mathbf{z}_k) = r \text{ and } z_{j,\ell} = \alpha\right\}\right| \approx \begin{cases} \frac{K}{n_c^L}\binom{L-1}{r}(n_c - 1)^r & \text{if } z_{k,\ell} = \alpha \\ \frac{K}{n_c^L}\binom{L-1}{r-1}(n_c - 1)^{r-1} & \text{if } z_{k,\ell} \neq \alpha \end{cases} \tag{33}$$

and this approximate equality hold for most $k \in [K]$, $r \in [L]$, $\ell \in [L]$, and $\alpha \in [n_c]$. The symmetry assumption 1 from the main paper requires (33) to hold not approximatively, but exactly. For convenience we restate below this symmetry assumption:

**Assumption A** (Latent Symmetry). *For every $k \in [K]$, $r \in [L]$, $\ell \in [L]$, and $\alpha \in [n_c]$ the identities*

$$\left|\left\{j \in [K] : \text{dist}(\mathbf{z}_j, \mathbf{z}_k) = r \text{ and } z_{j,\ell} = \alpha\right\}\right| = \begin{cases} \frac{K}{n_c^L}\binom{L-1}{r}(n_c - 1)^r & \text{if } z_{k,\ell} = \alpha \\ \frac{K}{n_c^L}\binom{L-1}{r-1}(n_c - 1)^{r-1} & \text{if } z_{k,\ell} \neq \alpha \end{cases} \tag{34}$$

*hold.*

To be clear, if the latent variables $\mathbf{z}_1, \ldots, \mathbf{z}_K$ are selected uniformly at random from $\mathcal{Z}$, then they will only *approximatively* satisfy assumption A. Our analysis, however, is conducted in the idealized case where the latent variables *exactly* satisfy the symmetry assumption. Specifically, we show that, in the idealized case where assumption A is *exactly* satisfied, then the weights $W$ and $U$ of the network are given by some explicit analytical formula. Importantly, as it is explained in the main paper, our experiments demonstrate that these idealized analytical formula provide very good approximations for the weights observed in experiments when the latent variables are selected uniformly at random.

In the next lemma, we isolate three properties which hold for any latent variables satisfying assumption A. Importantly, when proving collapse, we will only rely on these three properties — we will never explicitly need assumption A. We will see shortly that these three properties, in essence, amount to saying that all position $\ell \in [L]$ and all concepts $\alpha \in [n_c]$ plays interchangeable roles for the latent variables. There are no 'preferred' $\ell$ or $\alpha$, and this is exactly what will allow us to derive symmetric analytical solutions.

Before stating our lemma, let us define the 'sphere' of radius $r$ centered around the $k^{th}$ latent variable

$$S_r(k) := \left\{ j \in [K] : \mathrm{dist}(\mathbf{z}_j, \mathbf{z}_k) = r \right\} \qquad \text{for } r, k \in [L] \tag{35}$$

With this notation in hand we may now state

**Lemma B.** *Suppose the latent variables $\mathbf{z}_1, \ldots, \mathbf{z}_K$ satisfy the symmetry assumption A. Then $\mathbf{z}_1, \ldots, \mathbf{z}_K$ satisfies the following properties:*

*(i)* $|S_r(j)| = |S_r(k)|$ *for all $r \in [L]$ and all $j, k \in [K]$.*

*(ii) The equalities*

$$\sum_{k=1}^{K} Z_k = \frac{K}{n_c} \mathbf{1}_{n_c} \mathbf{1}_L^T \qquad and \qquad ZZ^T = \frac{KL}{n_c} I_{n_c}$$

*hold, with $I_{n_c}$ denoting the $n_c \times n_c$ identity matrix.*

*(iii) There exists $\theta_1, \ldots, \theta_L > 0$ and matrices $A_1, \ldots, A_L \in \mathbb{R}^{n_c \times L}$ such that*

$$Z_k - \frac{1}{|S_r(k)|} \sum_{j \in S_r(k)} Z_j = \theta_r Z_k + A_r$$

*holds for all $r \in [L]$, all $j \in [K]$, and all $k \in [K]$.*

We will prove this lemma shortly, but for now let us start by getting some intuition about properties (i), (ii) and (iii). Property (i) is transparent: it states that all latent variables have the same number of 'distance-$r$ neighbors'. Recalling how matrix $Z_k$ was defined (c.f. (21)), we see that the first identity of (ii) is equivalent to

$$|\{k \in [K] : z_{k,\ell} = \alpha\}| = \frac{K}{n_c} \qquad \text{for all } \ell \in [L] \text{ and all } \alpha \in [n_c]. \tag{36}$$

This means that the number of latent variables that have concept $\alpha$ in position $\ell$ is equal to $K/n_c$. In other words, each concept is equally represented at each position $\ell$. We now turn to the second identity of statement (ii). Recalling the definition (22) of matrix $Z$, we see that $ZZ^T \in \mathbb{R}^{n_c \times n_c}$ is a diagonal matrix since each column of $Z$ contains a single nonzero entry. One can also easily see that the $\alpha^{th}$ entry of the diagonal is

$$\left[ ZZ^T \right]_{\alpha,\alpha} = |\{(k, \ell) \in [K] \times [L] : z_{k,\ell} = \alpha\}|,$$

which is the total number of times concept $\alpha$ appears in the latent variables. Overall, the identity $ZZ^T = \frac{KL}{n_c} I_{n_c}$ is therefore equivalent to the statement

$$|\{(k, \ell) \in [K] \times [L] : z_{k,\ell} = \alpha\}| = \frac{KL}{n_c} \qquad \text{for all } \alpha \in [n_c]$$

and it is therefore a direct consequence of (36).

Property (iii) is harder to interpret. Essentially it is a type of mean value property that states that summing over the latent variables which are at distance $r$ of $\mathbf{z}_k$ gives back $\mathbf{z}_k$. We will see that this mean value property plays a key role in our analysis.

To conclude this subsection, we prove lemma B.

*Proof of lemma B.* We start by proving statement (i). Since $S_r(k) = \{j \in [K] : \text{dist}(\mathbf{z}_j, \mathbf{z}_k) = r\}$, we clearly have that

$$|S_r(k)| = \sum_{\alpha=1}^{n_c} \left| \left\{ j \in [K] : \text{dist}(\mathbf{z}_j, \mathbf{z}_k) = r \text{ and } z_{j,\ell} = \alpha \right\} \right| \tag{37}$$

We then use identity (34) and Pascal's rule to find

$$|S_r(k)| = (n_c - 1) \left( \frac{K}{|\mathcal{Z}|} \binom{L-1}{r-1} (n_c - 1)^{r-1} \right) + \frac{K}{|\mathcal{Z}|} \binom{L-1}{r} (n_c - 1)^r$$

$$= \frac{K}{|\mathcal{Z}|} (n_c - 1)^r \left( \binom{L-1}{r-1} + \binom{L-1}{r} \right)$$

$$= \frac{K}{|\mathcal{Z}|} \binom{L}{r} (n_c - 1)^r \tag{38}$$

which clearly implies that $|S_r(k)| = |S_r(j)|$ for all $j, k \in [K]$ and all $r \in [L]$.

We now turn to the first identity of t (ii). As previously mentioned, this identity is equivalent to (36). Choose $k$ such that $\mathbf{z}_{k,\ell} \neq \alpha$. Then any any latent variable $\mathbf{z}_j$ with $z_{j,\ell} = \alpha$ is least at a distance 1 of $\mathbf{z}_k$ and we may write

$$|\{j \in [K] : z_{j,\ell} = \alpha\}| = \sum_{r=1}^{L} \left| \left\{ j \in [K] : \text{dist}(\mathbf{z}_j, \mathbf{z}_k) = r \text{ and } z_{j,\ell} = \alpha \right\} \right| \tag{39}$$

$$= \sum_{r=1}^{L} \frac{K}{n_c^L} \binom{L-1}{r-1} (n_c - 1)^{r-1} \tag{40}$$

which is equal to $K/n_c$ according to the binomial theorem. The second identity of (ii), as mentioned earlier, is a direct consequence of the first identity.

We finally turn to statement (iii). Appealing to (38), we find that,

$$\frac{\left| \left\{ j \in [K] : \text{dist}(\mathbf{z}_j, \mathbf{z}_k) = r \text{ and } z_{j,\ell} = \alpha \right\} \right|}{|S_r(k)|} = \frac{\frac{K}{|\mathcal{Z}|} \binom{L-1}{r} (n_c - 1)^r}{\frac{K}{|\mathcal{Z}|} \binom{L}{r} (n_c - 1)^r} = \frac{\binom{L-1}{r}}{\binom{L}{r}} = \frac{L-r}{L}$$

if $z_{k,\ell} = \alpha$. On the other hand, if $z_{k,\ell} \neq \alpha$, we obtain

$$\frac{\left| \left\{ j \in [K] : \text{dist}(\mathbf{z}_j, \mathbf{z}_k) = r \text{ and } z_{j,\ell} = \alpha \right\} \right|}{|S_r(k)|} = \frac{\frac{K}{|\mathcal{Z}|} \binom{L-1}{r-1} (n_c - 1)^{r-1}}{\frac{K}{|\mathcal{Z}|} \binom{L}{r} (n_c - 1)^r} = \frac{1}{n_c - 1} \frac{\binom{L-1}{r-1}}{\binom{L}{r}} = \frac{1}{n_c - 1} \frac{r}{L}$$

Fix $\ell \in [L]$ and assume that $z_{k,\ell} = \alpha^\star$. We then have

$$\frac{1}{|S_r(k)|} \sum_{j \in S_r(k)} \mathbf{e}_{z_{j,\ell}} = \frac{1}{|S_r(k)|} \sum_{\alpha=1}^{n_c} \left| \left\{ j \in S_r(k) : \mathbf{z}_{j,\ell} = \mathbf{e}_\alpha \right\} \right| \mathbf{e}_\alpha$$

$$= \sum_{\alpha=1}^{n_c} \frac{\left| \left\{ j \in [K] : \text{dist}(\mathbf{z}_j, \mathbf{z}_k) = r \text{ and } z_{j,\ell} = \alpha \right\} \right|}{|S_r(k)|} \mathbf{e}_\alpha$$

$$= \frac{L-r}{L} \mathbf{e}_{\alpha^\star} + \frac{1}{n_c - 1} \frac{r}{L} \sum_{\alpha \neq \alpha^\star} \mathbf{e}_\alpha$$

$$= \frac{L-r}{L} \mathbf{e}_{\alpha^\star} - \frac{1}{n_c - 1} \frac{r}{L} \mathbf{e}_{\alpha^\star} + \frac{1}{n_c - 1} \frac{r}{L} \sum_{\alpha=1}^{n_c} \mathbf{e}_\alpha$$

$$= \left( 1 - \frac{n_c}{n_c - 1} \frac{r}{L} \right) \mathbf{e}_{\alpha^\star} + \frac{1}{n_c - 1} \frac{r}{L} \mathbf{1}_{n_c}$$

Recalling that $z_{k,\ell} = \alpha^\star$, the above implies that

$$\mathbf{e}_{z_{k,\ell}} - \frac{1}{|S_r(k)|} \sum_{j \in S_r(k)} \mathbf{e}_{z_{j,\ell}} = \frac{n_c}{n_c - 1} \frac{r}{L} \mathbf{e}_{z_{k,\ell}} - \frac{1}{n_c - 1} \frac{r}{L} \mathbf{1}_{n_c} \tag{41}$$

Finally, recalling that

$$Z_k = \begin{bmatrix} | & | & & | \\ \mathbf{e}_{z_{k,1}} & \mathbf{e}_{z_{k,2}} & \cdots & \mathbf{e}_{z_{k,L}} \\ | & | & & | \end{bmatrix} \in \mathbb{R}^{n_c \times L}$$

we see that (41) can be written in matrix format as

$$Z_k - \frac{1}{|S_r(k)|} \sum_{j \in S_r(k)} Z_j = \frac{n_c}{n_c - 1} \frac{r}{L} Z_k - \frac{1}{n_c - 1} \frac{r}{L} \mathbf{1}_{n_c} \mathbf{1}_L^T$$

and therefore the scalars $\theta_r$ and the matrices $A_r$ appearing in statement (iii) are given by the formula $\theta_r = \frac{n_c}{n_c - 1} \frac{r}{L}$ and $A_r = -\frac{1}{n_c - 1} \frac{r}{L} \mathbf{1}_{n_c} \mathbf{1}_L^T$. □

## C.2 SYMMETRY ASSUMPTION NEEDED FOR THE CONVERSE OF THEOREM 1 AND 3

In this subsection we present the symmetry assumption that will be needed to prove the converse of theorem 1 and 3. This assumption, as we will shortly see, is quite mild and is typically satisfied even for small values of $K$.

For each pair of latent variables $(\mathbf{z}_j, \mathbf{z}_k)$ we define the matrix

$$\Gamma^{(j,k)} := Z_j (Z_j - Z_k)^T \in \mathbb{R}^{n_c \times n_c}.$$

We also define

$$\mathcal{A} := \left\{ A \in \mathbb{R}^{n_c \times n_c} : \text{There exists } a, b \in \mathbb{R} \text{ s.t. } A = a I_{n_c} + b \mathbf{1}_{n_c} \mathbf{1}_{n_c}^T \right\} \tag{42}$$

which is the set of matrices whose diagonal entries are equal to some constant and whose off-diagonal entries are equal to some possibly different constant. We may now state our symmetry assumption.

**Assumption B.** *Any positive semi-definite matrix* $A \in \mathbb{R}^{n_c \times n_c}$ *that satisfies*

$$\left\langle A , \Gamma^{(j,k)} - \Gamma^{(j',k')} \right\rangle_F = 0 \qquad \forall j, k, j', k' \in [K] \text{ s.t. } \text{dist}(\mathbf{z}_j, \mathbf{z}_k) = \text{dist}(\mathbf{z}_{j'}, \mathbf{z}_{k'}) \tag{43}$$

*must belongs to* $\mathcal{A}$.

Note that (43) can be viewed as a linear system of equations for the unknown $A \in \mathbb{R}^{n_c \times n_c}$, with one equation for each quadruplet $(j, k, j', k')$ satisfying $\text{dist}(\mathbf{z}_j, \mathbf{z}_k) = \text{dist}(\mathbf{z}_{j'}, \mathbf{z}_{k'})$. To put it differently, each quadruplet $(j, k, j', k')$ satisfying $\text{dist}(\mathbf{z}_j, \mathbf{z}_k) = \text{dist}(\mathbf{z}_{j'}, \mathbf{z}_{k'})$ adds one equation to the system, and our assumption requires that we have enough of these equations so that all positive semi-definite solutions are constrained to live in the set $\mathcal{A}$. Since a symmetric matrix has $(n_c + 1)n_c/2$ distinct entries, we would expect that $(n_c + 1)n_c/2$ quadruplets should be enough to fully determine the matrix. This number of quadruplets is easily achieved even for small values of $K$. So assumption B is quite mild.

The next lemma states that assumption B is satisfied when $K = n_c^L$. In light of the above discussion this is not surprising, since the choice $K = n_c^L$ leads to a system with a number of equations much larger than $(n_c + 1)n_c/2$. The proof, however, is instructive: it simply handpicks $(n_c + 1)n_c/2 - 2$ quadruplets to determine the entries of the matrix $A$. The '$-2$' arises from the fact $\mathcal{A}$ is a 2 dimensional subspace, and therefore $(n_c + 1)n_c/2 - 2$ equations are 'enough' to constrain $A$ to be in $\mathcal{A}$.

**Lemma C.** *Suppose* $K = n_c^L$ *and* $\{\mathbf{z}_1, \ldots, \mathbf{z}_K\} = \mathcal{Z}$. *Then* $\mathbf{z}_1, \ldots, \mathbf{z}_K$ *satisfy the symmetry assumption B.*

*Proof.* Let $A = C^T C$ be a positive semi-definite matrix that solve satisfies (43). We use $\mathbf{c}_\alpha$ to denote the $\alpha^{th}$ column of $C$. Since $\{\mathbf{z}_1, \ldots, \mathbf{z}_K\} = \mathcal{Z}$, we can find $i, j, k \in [K]$ such that

$$\mathbf{z}_i = [2, 1, 1, \ldots, 1] \in \mathcal{Z}$$
$$\mathbf{z}_j = [3, 1, 1, \ldots, 1] \in \mathcal{Z}$$
$$\mathbf{z}_k = [4, 1, 1, \ldots, 1] \in \mathcal{Z}$$

Using lemma A and recalling the definition (21) of the matrix $Z_k$, we get

$$
\begin{aligned}
\left\langle A , \Gamma^{(i,j)} \right\rangle_F &= \langle C^T C, Z_i(Z_i - Z_j)^T \rangle_F \\
&= \langle C(Z_i - Z_j), CZ_i \rangle_F \\
&= \langle CZ_i, CZ_i \rangle_F - \langle CZ_j, CZ_i \rangle_F \\
&= \Big( \langle \mathbf{c}_2, \mathbf{c}_2 \rangle + (L-1)\langle \mathbf{c}_1, \mathbf{c}_1 \rangle \Big) - \Big( \langle \mathbf{c}_2, \mathbf{c}_3 \rangle + (L-1)\langle \mathbf{c}_1, \mathbf{c}_1 \rangle \Big) \\
&= \langle \mathbf{c}_2, \mathbf{c}_2 \rangle - \langle \mathbf{c}_2, \mathbf{c}_3 \rangle
\end{aligned}
$$

Similarly we obtain that

$$
\left\langle A , \Gamma^{(i,k)} \right\rangle_F = \langle \mathbf{c}_2, \mathbf{c}_2 \rangle - \langle \mathbf{c}_2, \mathbf{c}_4 \rangle
$$

Since $\mathrm{dist}(\mathbf{z}_i, \mathbf{z}_j) = \mathrm{dist}(\mathbf{z}_i, \mathbf{z}_k) = 1$, and since $A$ satisfies (43), we must have

$$
\left\langle A , \Gamma^{(i,j)} \right\rangle_F = \left\langle A , \Gamma^{(i,k)} \right\rangle_F
$$

which in turn implies that

$$
A_{2,3} = \langle \mathbf{c}_2, \mathbf{c}_3 \rangle = \langle \mathbf{c}_2, \mathbf{c}_4 \rangle = A_{2,4}
$$

This argument easily generalizes to show that all off-diagonal entries of the matrix $A$ must be equal to some constant $b \in \mathbb{R}$.

We now take care of the diagonal entries. Since $\{\mathbf{z}_1, \ldots, \mathbf{z}_K\} = \mathcal{Z}$, we can find $i', j', k' \in [K]$ such that

$$
\begin{aligned}
\mathbf{z}_{i'} &= [1, 1, \ldots, 1] \in \mathcal{Z} \\
\mathbf{z}_{j'} &= [2, 2, \ldots, 2] \in \mathcal{Z} \\
\mathbf{z}_{k'} &= [3, 3, \ldots, 3] \in \mathcal{Z}
\end{aligned}
$$

As before, we compute

$$
\left\langle A , \Gamma^{(i',j')} \right\rangle_F = \langle CZ_{i'}, CZ_{i'} \rangle_F - \langle CZ_{j'}, CZ_{i'} \rangle_F = L\langle \mathbf{c}_1, \mathbf{c}_1 \rangle - L\langle \mathbf{c}_1, \mathbf{c}_2 \rangle = L\langle \mathbf{c}_1, \mathbf{c}_1 \rangle - Lb
$$

where we have used the fact that the off diagonal entries are all equal to $b$. Similarly we obtain

$$
\left\langle A , \Gamma^{(j',k')} \right\rangle_F = L\langle \mathbf{c}_2, \mathbf{c}_2 \rangle - Lb
$$

Since $\mathrm{dist}(\mathbf{z}_{i'}, \mathbf{z}_{j'}) = \mathrm{dist}(\mathbf{z}'_j, \mathbf{z}_{k'}) = L$, we must have $\left\langle A, \Gamma^{(i',j')} \right\rangle_F = \left\langle A, \Gamma^{(j',k')} \right\rangle_F$ which implies that $A_{1,1} = A_{2,2}$. This argument generalizes to show that all diagonal entries of $A$ are equal. $\qquad\square$

## D  SHARP LOWER BOUND ON THE UNREGULARIZED RISK

In this section we derive a sharp lower bound for the *unregularized* risk associated with the network $h_{W,U}$,

$$
\mathcal{R}_0(W, U) := \frac{1}{K} \sum_{k=1}^K \mathbb{E}_{\mathbf{x} \sim \mathcal{D}_{\mathbf{z}_k}} \Big[ \ell(h_{W,U}(\mathbf{x}), k) \Big], \tag{44}
$$

where $\ell : \mathbb{R}^K \to \mathbb{R}$ is the cross entropy loss

$$
\ell(\mathbf{y}, k) = -\log \left( \frac{\exp(y_k)}{\sum_{j=1}^K \exp(y_j)} \right) \qquad \text{for } \mathbf{y} \in \mathbb{R}^K
$$

The $k^{th}$ entry of the output $\mathbf{y} = h_{W,U}(\mathbf{x})$ of the neural network, according to formula (19), is given by

$$
y_k = \left\langle \hat{U}_k , W \zeta(\mathbf{x}) \right\rangle_F
$$

Recalling that $\mathcal{X}_k$ is the support of the distribution $\mathcal{D}_{\mathbf{z}_k} : \mathcal{X} \to [0, 1]$, we find that the unregularized risk can be expressed as

$$\mathcal{R}_0(W, U) = \frac{1}{K} \sum_{k=1}^{K} \sum_{\mathbf{x} \in \mathcal{X}_k} \ell(h_{W,U}(\mathbf{x}), k) \, \mathcal{D}_{\mathbf{z}_k}(\mathbf{x})$$

$$= \frac{1}{K} \sum_{k=1}^{K} \sum_{\mathbf{x} \in \mathcal{X}_k} -\log \left( \frac{e^{\langle \hat{U}_k, W\zeta(\mathbf{x}) \rangle_F}}{\sum_{j=1}^{K} e^{\langle \hat{U}_j, W\zeta(\mathbf{x}) \rangle_F}} \right) \mathcal{D}_{\mathbf{z}_k}(\mathbf{x})$$

$$= \frac{1}{K} \sum_{k=1}^{K} \sum_{\mathbf{x} \in \mathcal{X}_k} \log \left( 1 + \sum_{j \neq k} e^{-\langle \hat{U}_k - \hat{U}_j, W\zeta(\mathbf{x}) \rangle_F} \right) \mathcal{D}_{\mathbf{z}_k}(\mathbf{x})$$

where we did the slight abuse of notation of writing $\mathcal{D}_{\mathbf{z}_k}(\mathbf{x})$ instead of $\mathcal{D}_{\mathbf{z}_k}(\{\mathbf{x}\})$. Note that a data points $\mathbf{x}$ that belongs to class $k$ is correctly classified by the the network $h_{W,U}$ if and only if

$$\left\langle \hat{U}_k, W\zeta(\mathbf{x}) \right\rangle_F > \left\langle \hat{U}_j, W\zeta(\mathbf{x}) \right\rangle_F \quad \text{for all } j \neq k$$

With this in mind, we introduce the following definition:

**Definition A** (Margin). *Suppose $\mathbf{x} \in \mathcal{X}_k$. Then the margin between data point $\mathbf{x}$ and class $j$ is*

$$\mathfrak{M}_{W,U}(\mathbf{x}, j) := \left\langle \hat{U}_k - \hat{U}_j, W\zeta(\mathbf{x}) \right\rangle_F$$

With this definition in hand, the unregularized risk can conveniently be expressed as

$$\mathcal{R}_0(W, U) = \frac{1}{K} \sum_{k=1}^{K} \sum_{\mathbf{x} \in \mathcal{X}_k} \log \left( 1 + \sum_{j \neq k} e^{-\mathfrak{M}_{W,U}(\mathbf{x}, j)} \right) \mathcal{D}_{\mathbf{z}_k}(\mathbf{x}) \tag{45}$$

and a data point $\mathbf{x} \in \mathcal{X}_k$ is correctly classified by the network if and only if the margins $\mathfrak{M}_{W,U}(\mathbf{x}, j)$ are all strictly positive (for $j \neq k$). We then introduce a definition that will play crucial role in our analysis.

**Definition B** (Equimargin Property). *If*

$$\text{dist}(\mathbf{z}_k, \mathbf{z}_j) = \text{dist}(\mathbf{z}_{k'}, \mathbf{z}_{j'}) \implies \mathfrak{M}_{W,U}(\mathbf{x}, j) = \mathfrak{M}_{W,U}(\mathbf{x}', j') \quad \forall \mathbf{x} \in \mathcal{X}_k \text{ and } \forall \mathbf{x}' \in \mathcal{X}_{k'}$$

*then we say that $(W, U)$ satisfies the equimargin property.*

To put it simply, $(W, U)$ satisfies the equimargin property if the margin between data point $\mathbf{x} \in \mathcal{X}_k$ and class $j$ only depends on $\text{dist}(\mathbf{z}_k, \mathbf{z}_j)$. We denote by $\mathcal{E}$ the set of all the weights that satisfy the equimargin property

$$\mathcal{E} = \{(W, U) : (W, U) \text{ satisfies the equimargin property}\} \tag{46}$$

and by $\mathcal{N}$ the set of weights for which the submatrices $\hat{U}_k$ defined by (18) sum to 0,

$$\mathcal{N} = \left\{ (W, U) : \sum_{k=1}^{K} \hat{U}_k = 0 \right\} \tag{47}$$

We will work under the assumption that the latent variables $\mathbf{z}_1, \ldots, \mathbf{z}_K$ satisfy the symmetry assumption A. According to lemma B, $|S_r(k)|$ then doesn't depend on $k$, and so we will simply use $|S_r|$ to denote the size of the set $S_r(k)$. Lemma B also states that

$$Z_k - \frac{1}{|S_r(k)|} \sum_{j \in S_r(k)} Z_j = \theta_r Z_k + A_r$$

for some matrices $A_1, \ldots, A_L$ and some scalars $\theta_1, \ldots, \theta_L > 0$. We use these scalars to define

$$g(x) := \log \left( 1 + \sum_{r=1}^{L} |S_r| \, e^{\theta_r x / K} \right) \tag{48}$$

and we note that $g : \mathbb{R} \to \mathbb{R}$ is a strictly increasing function. With these definitions in hand we may state the main theorem of this section.

**Theorem D.** *If the latent variables satisfy the symmetry assumption A, then*

$$\mathcal{R}_0(W, U) = g\left(-\left\langle \hat{U}, WQ^T Z\right\rangle_F\right) \qquad \text{for all } (W, U) \in \mathcal{N} \cap \mathcal{E} \tag{49}$$

$$\mathcal{R}_0(W, U) > g\left(-\left\langle \hat{U}, WQ^T Z\right\rangle_F\right) \qquad \text{for all } (W, U) \in \mathcal{N} \cap \mathcal{E}^c \tag{50}$$

We recall that the matrices $\hat{U}$, $Q$, and $Z$ where defined in section B (c.f. (20), (26) and (22)). The remainder of this section is devoted to the proof of the above theorem.

### D.1   PROOF OF THE THEOREM

We will use two lemmas to prove the theorem. The first one (lemma D below) simply leverages the strict convexity of the various components defining the unregularized risk $\mathcal{R}_0$. Recall that if $f : \mathbb{R}^d \to \mathbb{R}$ is strictly convex, and if the strictly positive scalars $p_1, \ldots, p_n > 0$ sum to 1, then

$$f\left(\sum_{i=1}^{n} p_i \mathbf{v}_i\right) \leq \sum_{i=1}^{n} p_i f(\mathbf{v}_i) \tag{51}$$

and that equality holds if and only if $\mathbf{v}_1 = \mathbf{v}_2 = \ldots = \mathbf{v}_n$. For this first lemma, the only property we need on the latent variables is that $|S_r(k)| = |S_r(j)| = |S_r|$ for all $j, k \in [K]$ and all $r \in [L]$.

Define the quantity

$$\mathfrak{N}_{W,U}(r) = \frac{1}{K} \frac{1}{|S_r|} \sum_{k=1}^{K} \sum_{j \in S_r(k)} \sum_{\mathbf{x} \in \mathcal{X}_k} \mathfrak{M}_{W,U}(\mathbf{x}, j) \, \mathcal{D}_{\mathbf{z}_k}(\mathbf{x}) \tag{52}$$

which should be viewed as the averaged margin between data points and classes which are at a distance $r$ of one another. We then have the following lemma:

**Lemma D.** *If $|S_r(k)| = |S_r(j)|$ for all $j, k \in [K]$ and all $r \in [L]$, then*

$$\mathcal{R}_0(W, U) = \log\left(1 + \sum_{r=1}^{L} |S_r| e^{-\mathfrak{N}_{W,U}(r)}\right) \qquad \text{for all } (W, U) \in \mathcal{E} \tag{53}$$

$$\mathcal{R}_0(W, U) > \log\left(1 + \sum_{r=1}^{L} |S_r| e^{-\mathfrak{N}_{W,U}(r)}\right) \qquad \text{for all } (W, U) \notin \mathcal{E} \tag{54}$$

*Proof.* Using the strict convexity of the function $f : \mathbb{R}^{K-1} \to \mathbb{R}$ defined by

$$f(v_1, \ldots, v_{k-1}, v_{k+1}, \ldots, v_K) = \log\left(1 + \sum_{j \neq k} e^{v_j}\right)$$

we obtain

$$\mathcal{R}_0(W, U) = \frac{1}{K} \sum_{k=1}^{K} \sum_{\mathbf{x} \in \mathcal{X}_k} \log\left(1 + \sum_{j \neq k} e^{-\mathfrak{M}(\mathbf{x}, j)}\right) \mathcal{D}_{\mathbf{z}_k}(\mathbf{x})$$

$$\geq \frac{1}{K} \sum_{k=1}^{K} \log\left(1 + \sum_{j \neq k} e^{-\sum_{\mathbf{x} \in \mathcal{X}_k} \mathfrak{M}(\mathbf{x}, j) \mathcal{D}_{\mathbf{z}_k}(\mathbf{x})}\right)$$

and equality holds if and only if, for all $k \in [K]$, we have that

$$\mathfrak{M}(\mathbf{x}, j) = \mathfrak{M}(\mathbf{y}, j) \qquad \text{for all } \mathbf{x}, \mathbf{y} \in \mathcal{X}_k \text{ and all } j \neq k \tag{55}$$

We then let

$$\overline{\mathfrak{M}}(k, j) = \sum_{\mathbf{x} \in \mathcal{X}_k} \mathfrak{M}(\mathbf{x}, j) \mathcal{D}_{\mathbf{z}_k}(\mathbf{x})$$

and use the strict convexity of the exponential function to obtain

$$\frac{1}{K}\sum_{k=1}^{K}\log\left(1+\sum_{j\neq k}e^{-\overline{\mathfrak{M}}(k,j)}\right) = \frac{1}{K}\sum_{k=1}^{K}\log\left(1+\sum_{r=1}^{L}\sum_{j\in S_r(k)}e^{-\overline{\mathfrak{M}}(k,j)}\right)$$

$$= \frac{1}{K}\sum_{k=1}^{K}\log\left(1+\sum_{r=1}^{L}|S_r|\frac{1}{|S_r|}\sum_{j\in S_r(k)}e^{-\overline{\mathfrak{M}}(k,j)}\right)$$

$$\geq \frac{1}{K}\sum_{k=1}^{K}\log\left(1+\sum_{r=1}^{L}|S_r|e^{-\frac{1}{|S_r|}\sum_{j\in S_r(k)}\overline{\mathfrak{M}}(k,j)}\right)$$

Moreover, equality holds if and only if, for all $k \in [K]$ and all $r \in [L]$, we have that

$$\overline{\mathfrak{M}}(k,i) = \overline{\mathfrak{M}}(k,j) \quad \text{for all } i,j \in S_r(k) \tag{56}$$

We finally set

$$\overline{\overline{\mathfrak{M}}}(k,r) = \frac{1}{|S_r|}\sum_{j\in S_r(k)}\overline{\mathfrak{M}}(k,j)$$

and use the strict convexity of the function $f(v_1,\ldots,v_L) = \log\left(1+\sum_{r=1}^{L}|S_r|e^{v_r}\right)$ to get

$$\frac{1}{K}\sum_{k=1}^{K}\log\left(1+\sum_{r=1}^{L}|S_r|e^{-\overline{\overline{\mathfrak{M}}}(k,r)}\right) \geq \log\left(1+\sum_{r=1}^{L}|S_r|e^{-\frac{1}{K}\sum_{k=1}^{K}\overline{\overline{\mathfrak{M}}}(k,r)}\right)$$

Moreover equality holds if and only if, for all $k \in [K]$ and all $r \in [L]$, we have that

$$\overline{\overline{\mathfrak{M}}}(k,r) = \overline{\overline{\mathfrak{M}}}(k',r) \quad \text{for all } k,k' \in [K] \text{ and all } r \in [L] \tag{57}$$

Importantly, note that

$$\frac{1}{K}\sum_{k=1}^{K}\overline{\overline{\mathfrak{M}}}(k,r) = \frac{1}{K}\frac{1}{|S_r|}\sum_{k=1}^{K}\sum_{j\in S_r(k)}\sum_{\mathbf{x}\in\mathcal{X}_k}\mathfrak{M}_{W,U}(\mathbf{x},j)\,\mathcal{D}_{\mathbf{z}_k}(\mathbf{x})$$

which is precisely how $\mathfrak{N}_{W,U}(r)$ was defined (c.f. (52)). To conclude the proof, we remark that conditions (55), (56) and (57) are all satisfied if and only if $(W,U)$ satisfies the equi-margin property.
□

We now show that, if assumption A holds, $\mathfrak{N}_{W,U}(r)$ can be expressed in a simple way.

**Lemma E.** *Assume that the latent variables satisfy the symmetry assumption A. Then*

$$\mathfrak{N}_{W,U}(r) = \frac{\theta_r}{K}\left\langle\hat{U}, WQ^TZ\right\rangle_F \qquad \text{for all } (W,U)\in\mathcal{N} \tag{58}$$

*Proof.* We let

$$\overline{X}_k = \sum_{\mathbf{x}\in\mathcal{X}_k}\zeta(\mathbf{x})\mathcal{D}_{\mathbf{z}_k}(\mathbf{x})$$

and note that the averaged margin can be expressed as

$$\mathfrak{N}_{W,U}(r) = \frac{1}{K}\frac{1}{|S_r|}\sum_{k=1}^{K}\sum_{j\in S_r(k)}\sum_{\mathbf{x}\in\mathcal{X}_k}\mathfrak{M}_{W,U}(\mathbf{x},j)\,\mathcal{D}_{\mathbf{z}_k}(\mathbf{x})$$

$$= \frac{1}{K}\frac{1}{|S_r|}\sum_{k=1}^{K}\sum_{j\in S_r(k)}\sum_{\mathbf{x}\in\mathcal{X}_k}\left\langle\hat{U}_k-\hat{U}_j, W\zeta(\mathbf{x})\right\rangle_F\mathcal{D}_{\mathbf{z}_k}(\mathbf{x})$$

$$= \frac{1}{K}\frac{1}{|S_r|}\sum_{k=1}^{K}\sum_{j\in S_r(k)}\left\langle\hat{U}_k-\hat{U}_j, \overline{X}_k\right\rangle_F$$

$$= \frac{1}{K}\sum_{k=1}^{K}\left\langle\hat{U}_k, W\overline{X}_k\right\rangle_F - \frac{1}{K}\frac{1}{|S_r|}\sum_{k=1}^{K}\sum_{j\in S_r(k)}\left\langle\hat{U}_j, W\overline{X}_k\right\rangle_F \tag{59}$$

Let

$$
a_{k,j}^{(r)} = \begin{cases} 1 & \text{if } \text{dist}(\mathbf{z}_k, \mathbf{z}_j) = r \\ 0 & \text{otherwise} \end{cases}
$$

and rewrite the second term in (59) as

$$
\frac{1}{K}\frac{1}{|S_r|}\sum_{k=1}^{K}\sum_{j\in S_r(k)}\left\langle \hat{U}_j, W\overline{X}_k \right\rangle_F = \frac{1}{K}\frac{1}{|S_r|}\sum_{k=1}^{K}\sum_{j=1}^{K}a_{k,j}^{(r)}\left\langle U_j, W\overline{X}_k \right\rangle_F
$$

$$
= \frac{1}{K}\frac{1}{|S_r|}\sum_{j=1}^{K}\sum_{k=1}^{K}a_{j,k}^{(r)}\left\langle \hat{U}_k, W\overline{X}_j \right\rangle_F
$$

$$
= \frac{1}{K}\frac{1}{|S_r|}\sum_{k=1}^{K}\sum_{j\in S_r(k)}\left\langle \hat{U}_k, W\overline{X}_j \right\rangle_F
$$

$$
= \frac{1}{K}\sum_{k=1}^{K}\left\langle \hat{U}_k, W\frac{1}{|S_r|}\sum_{j\in S_r(k)}\overline{X}_j \right\rangle_F
$$

Combining this with (59) we obtain

$$
\mathfrak{N}_{W,U}(r) = \frac{1}{K}\sum_{k=1}^{K}\left\langle \hat{U}_k, W\left(\overline{X}_k - \frac{1}{|S_r|}\sum_{j\in S_r(k)}\overline{X}_j\right)\right\rangle_F \tag{60}
$$

From formula (26), we see that row $\alpha$ of the matrix $Q$ is given by the formula

$$
Q^T\mathbf{e}_\alpha = \sum_{\beta=1}^{s_c}\zeta(\alpha,\beta)\,\mu_\beta. \tag{61}
$$

We then write $\mathbf{z}_k = [\alpha_1, \ldots, \alpha_L]$ and note that the $\ell^{th}$ column of $\overline{X}_k$ can be expressed as

$$
\left[\overline{X}_k\right]_{:,\ell} = \sum_{\beta=1}^{n_c}\zeta(\alpha_\ell,\beta)\mu_\beta = Q^T\mathbf{e}_{\alpha_\ell}. \tag{62}
$$

From this we obtain that

$$
\overline{X}_k = Q^T Z_k
$$

and therefore (60) becomes

$$
\mathfrak{N}_{W,U}(r) = \frac{1}{K}\sum_{k=1}^{K}\left\langle \hat{U}_k, WQ^T\left(Z_k - \frac{1}{|S_r|}\sum_{j\in S_r(k)}Z_j\right)\right\rangle_F
$$

$$
= \frac{1}{K}\sum_{k=1}^{K}\left\langle \hat{U}_k, WQ^T\left(\theta_r Z_k + A_r\right)\right\rangle_F
$$

where we have used the identity $Z_k - \frac{1}{|S_r|}\sum_{j\in S_r(k)} Z_j = \theta_r Z_k + A_r$ to obtain the second equality. Finally, we use the fact that $\sum_k \hat{U}_k = 0$ to obtain

$$
\mathfrak{N}_{W,U}(r) = \frac{\theta_r}{K}\sum_{k=1}^{K}\left\langle \hat{U}_k, WQ^T Z_k \right\rangle_F = \frac{\theta_r}{K}\left\langle \hat{U}, WQ^T Z \right\rangle_F
$$

$\square$

Combining lemma D and E concludes the proof of theorem D.

## E  PROOF OF THEOREM 1 AND ITS CONVERSE

In this section we prove theorem 1 under assumption A, and its converse under assumptions A and B. We start by recalling the definition of a type-I collapse configuration.

**Definition C** (Type-I Collapse). *The weights $(W, U)$ of the network $h_{W,U}$ form a type-I collapse configuration if and only if the conditions*

i) *There exists $c \geq 0$ so that $\mathbf{w}_{(\alpha, \beta)} = c\mathfrak{f}_\alpha$ for all $(\alpha, \beta) \in \mathcal{V}$.*

ii) *There exists $c' \geq 0$ so that $\mathbf{u}_{k,\ell} = c'\mathfrak{f}_\alpha$ for all $(k, \ell)$ satisfying $z_{k,\ell} = \alpha$ and all $\alpha \in \mathcal{C}$.*

*hold for some collection $\mathfrak{f}_1, \ldots, \mathfrak{f}_{n_c} \in \mathbb{R}^d$ of equiangular vectors.*

It will prove convenient to reformulate this definition using matrix notations. Toward this goal, we define equiangular matrices as follow:

**Definition D.** *(Equiangular Matrices) A matrix $\mathfrak{F} \in \mathbb{R}^{d \times n_c}$ is said to be equiangular if and only if the relations*

$$\mathfrak{F}\mathbf{1}_{n_c} = 0 \qquad and \qquad \mathfrak{F}^T\mathfrak{F} = \frac{n_c}{n_c - 1} I_{n_c} - \frac{1}{n_c - 1} \mathbf{1}_{n_c}\mathbf{1}_{n_c}^T$$

*hold.*

Comparing the above definition with the definition of equiangular vectors provided in the main paper, we easily see that a matrix

$$\mathfrak{F} = \begin{bmatrix} | & | & & | \\ \mathfrak{f}_1 & \mathfrak{f}_2 & \cdots & \mathfrak{f}_{n_c} \\ | & | & & | \end{bmatrix} \in \mathbb{R}^{d \times n_c}$$

is equiangular if and only if its columns $\mathfrak{f}_1, \ldots, \mathfrak{f}_{n_c} \in \mathbb{R}^d$ are equiangular. Relations (i) and (ii) defining a type-I collapse configuration can now be expressed in matrix format as

$$W = c\,\mathfrak{F}\,P \qquad and \qquad \hat{U} = c'\,\mathfrak{F}\,Z \qquad \text{for some equiangular matrix } \mathfrak{F}$$

where the matrices $Z$ and $P$ are given by formula (22) and (23). We then let

$$\Omega_c^I := \Big\{ (W, U) : \text{There exist an equiangular matrix } \mathfrak{F} \text{ such that}$$

$$W = c\,\mathfrak{F}\,P \quad \text{and} \quad \hat{U} = c\,\sqrt{\frac{n_w}{KL}}\,\mathfrak{F}\,Z \Big\} \quad (63)$$

and note that $\Omega_c^I$ is simply the set of weights $(W, U)$ which are in a type-I collapse configuration with constant $c$ and $c' = c\sqrt{n_w/(KL)}$. We now state the main theorem of this section.

**Theorem E.** *Assume uniform sampling $\mu_\beta = 1/s_c$ for each word distribution. Let $\tau \geq 0$ denote the unique minimizer of the strictly convex function*

$$H(t) := \log\left(1 - \frac{K}{n_c^L} + \frac{K}{n_c^L}\left(1 + (n_c - 1)e^{-\eta t}\right)^L\right) + \lambda t \qquad where \quad \eta = \frac{n_c}{n_c - 1}\frac{1}{\sqrt{n_w KL}}$$

*and let $c = \sqrt{\tau/n_w}$. Then we have the following:*

(i) *If the latent variables $\mathbf{z}_1, \ldots, \mathbf{z}_K$ are mutually distinct and satisfy assumption A, then*
$$\Omega_c^I \subset \arg\min \mathcal{R}$$

(ii) *If the latent variables $\mathbf{z}_1, \ldots, \mathbf{z}_K$ are mutually distinct and satisfy assumptions A and B, then*
$$\Omega_c^I = \arg\min \mathcal{R}$$

Note that (i) states that any $(W, U) \in \Omega_c^I$ is a minimizer of the regularized risk — this corresponds to theorem 1 from the main paper. Statement (ii) assert that any minimizer of the regularized risk must belong to $\Omega_c^I$ — this is the converse of theorem 1. The remainder of this section is devoted to the proof of theorem E. We will assume uniform sampling

$$\mu_\beta = 1/s_c \qquad \text{for all } \beta \in [s_c]$$

everywhere in this section — all lemmas and propositions are proven under this assumption, even when not explicitly stated.

### E.1 The Bilinear Optimization Problem

From theorem D, it is clear that the quantity

$$\left\langle \hat{U}, WQ^T Z \right\rangle_F$$

plays an important role in our analysis. In this subsection we consider the bilinear optimization problem

$$\text{maximize} \quad \left\langle \hat{U}, WQ^T Z \right\rangle_F \tag{64}$$

$$\text{subject to} \quad \frac{1}{2}\left( \|W\|_F^2 + \|\hat{U}\|_F^2 \right) = c^2\, n_w \tag{65}$$

where $c \in \mathbb{R}$ is some constant. The following lemma identifies all solutions of this optimization problem.

**Lemma F.** *Assume the latent variables satisfy assumption A. Then $(W, U)$ is a solution of the optimization problem (64) – (65) if and only if it belongs to the set*

$$\mathcal{B}_c^I = \Big\{ (W, U) : \textit{There exist a matrix } F \in \mathbb{R}^{d \times n_c} \textit{ with } \|F\|_F^2 = n_c$$

$$\textit{such that } W = c\, FP \textit{ and } \hat{U} = c\, \sqrt{\frac{n_w}{KL}}\, FZ \Big\} \tag{66}$$

Note that the set $\mathcal{B}_c^I$ is very similar to the set $\Omega_c^I$ that defines type-I collapse configuration (c.f. (92)). In particular, since an equiangular matrix has $n_c$ columns of norm 1, it always satisfies $\|\mathfrak{F}\|_F^2 = n_c$, and therefore we have the inclusion

$$\Omega_c^I \subset \mathcal{B}_c^I. \tag{67}$$

The remainder of this subsection is devoted to the proof of the lemma.

First note that the lemma is trivially true if $c = 0$, so we may assume $c \neq 0$ for the remainder of the proof. Second, we note that since $\mu_\beta = 1/s_c$, then the matrices $P$ and $Q$ defined by (23) and (26) are scalar multiple of one another. We may therefore replace the matrix $Q$ appearing in (64) by $P$, wich leads to

$$\text{maximize} \quad \left\langle \hat{U}, WP^T Z \right\rangle_F \tag{68}$$

$$\text{subject to} \quad \frac{1}{2}\left( \|W\|_F^2 + \|\hat{U}\|_F^2 \right) = c^2\, n_w \tag{69}$$

We now show that any $(W, \hat{U}) \in \mathcal{B}_c^I$ satisfies the constraint (69) and have objective value equal to $s_c\, c^2 \sqrt{KLn_w}$.

**Claim A.** *If $(W, \hat{U}) \in \mathcal{B}_c^I$, then*

$$\frac{1}{2}\left( \|W\|_F^2 + \|\hat{U}\|_F^2 \right) = c^2\, n_w \qquad \textit{and} \qquad \left\langle \hat{U}, WP^T Z \right\rangle_F = c^2\, s_c\, \sqrt{KLn_w}$$

*Proof.* Assume $(W, U) \in \mathcal{B}_c^I$. From definition (23) of the matrix $P$, we have $PP^T = s_c I_{n_c}$, and therefore

$$\|W\|_F^2 = c^2 \|FP\|_F^2 = c^2 \left\langle FP, FP \right\rangle_F = c^2 \left\langle FPP^T, F \right\rangle_F = c^2\, s_c\, \|F\|_F^2 = c^2\, s_c\, n_c = c^2\, n_w$$

where we have used the fact that $s_c = n_w/n_c$. Using $ZZ^T = \frac{KL}{n_c}I$ from lemma B, we obtain

$$\|FZ\|_F^2 = \left\langle FZ, FZ \right\rangle_F = \left\langle FZZ^T, F \right\rangle_F = \left( \frac{KL}{n_c} \right) \|F\|_F^2 = KL$$

As a consequence we have

$$\|\hat{U}\|_F^2 = c^2\, \frac{n_w}{KL}\, \|FZ\|_F^2 = c^2\, n_w$$

and, using $PP^T = s_c I_{n_c}$ one more time,

$$\left\langle \hat{U}, WP^T Z \right\rangle_F = c^2 \sqrt{\frac{n_w}{KL}} \left\langle FZ, FPP^T Z \right\rangle_F = c^2\, s_c \sqrt{\frac{n_w}{KL}} \left\langle FZ, FZ \right\rangle_F = c^2\, s_c\, \sqrt{KLn_w}$$

$\square$

We then prove that $W$ and $\hat{U}$ must have same Frobenius norm if they solve the optimization problem.

**Claim B.** *If $(W, U)$ is a solution of (68) – (69), then*

$$\|W\|_F^2 = \|\hat{U}\|_F^2 = c^2 \, n_w \tag{70}$$

*Proof.* We prove it by contradiction. Suppose $(W, \hat{U})$ is a solution of (64)–(65) with $\|W\|_F^2 \neq \|\hat{U}\|_F^2$. Since the average of $\|W\|_F^2$ and $\|\hat{U}\|_F^2$ is equal to $c^2 n_w > 0$ according to the constraint, there must then exists $\epsilon \neq 0$ such that

$$\|W\|_F^2 = c^2 n_w + \epsilon \qquad \text{and} \qquad \|\hat{U}\|_F^2 = c^2 n_w - \epsilon$$

Let

$$W_0 = \sqrt{\frac{c^2 n_w}{c^2 n_w + \epsilon}} \, W \qquad \text{and} \qquad \hat{U}_0 = \sqrt{\frac{c^2 n_w}{c^2 n_w - \epsilon}} \, \hat{U}$$

and note that

$$\|W_0\|_F^2 = \|\hat{U}_0\|_F^2 = c^2 \, n_w$$

and therefore $(W_0, \hat{U}_0)$ clearly satisfies the constraint. We also have

$$\left\langle \hat{U}_0, W_0 P^T Z \right\rangle_F = \sqrt{\frac{c^4 n_w^2}{c^4 n_w^2 - \epsilon^2}} \left\langle \hat{U}, W P^T Z \right\rangle_F > \left\langle \hat{U}, W P^T Z \right\rangle_F$$

since $\epsilon \neq 0$ and therefore $(W, \hat{U})$ can not be a maximizer, which is a contradiction. $\qquad \square$

As a consequence of the above claim, the optimization problem (68) – (69) is equivalent to

$$\text{maximize} \quad \left\langle \hat{U}, W P^T Z \right\rangle_F \tag{71}$$

$$\text{subject to} \quad \|W\|_F^2 = c^2 \, n_w \quad \text{and} \quad \|\hat{U}\|_F^2 = c^2 \, n_w \tag{72}$$

We then have

**Claim C.** *If $(W, \hat{U})$ is a solution of (71) – (72), then $(W, \hat{U}) \in \mathcal{B}_c^I$.*

Note that according to the first claim, all $(W, \hat{U}) \in \mathcal{B}_c^I$ have same objective value, and therefore, according to the above claim, they must all be maximizer. As a consequence, proving the above claim will conclude the proof of lemma F.

*Proof of the claim.* Maximizing (71) over $\hat{U}$ first gives

$$\hat{U} = c \sqrt{n_w} \frac{W P^T Z}{\|W P^T Z\|_F} \tag{73}$$

and therefore the optimization problem (71) – (72) reduces to

$$\text{maximize} \quad \|W P^T Z\|_F^2$$
$$\text{subject to} \quad \|W\|_F^2 = c^2 \, n_w$$

Using $ZZ^T = \frac{KL}{n_c} I$ from lemma B we then get

$$\|W P^T Z\|_F^2 = \left\langle W P^T Z, W P^T Z \right\rangle_F = \left\langle W P^T Z Z^T, W P^T \right\rangle_F = \frac{KL}{n_c} \|W P^T\|_F^2$$

and therefore the problem further reduces to

$$\text{maximize} \quad \|W P^T\|_F^2$$
$$\text{subject to} \quad \|W\|_F^2 = c^2 \, n_w$$

The KKT conditions for this optimization problem are

$$W P^T P = \nu W \tag{74}$$

$$\|W\|_F^2 = c^2 \, n_w \tag{75}$$

where $\nu \in \mathbb{R}$ is the Lagrange multiplier.

Assume that $(W, \hat{U})$ is a solution of the original optimization problem (71) – (72). Then, according to the above discussion, $W$ must satisfy (74) – (75). Right multiplying (74) by $P^T$, and using $PP^T = s_c I_{n_c}$, gives

$$s_c W P^T = \nu W P^T$$

So either $\nu = s_c$ or $WP^T = 0$. The latter is not possible since the choice $WP^T = 0$ leads to an objective value equal to zero in the original optimization problem (71) – (72). We must therefore have $\nu = s_c$, and equation (74) becomes

$$W = \frac{1}{s_c} W P^T P \tag{76}$$

which can obviously be written as

$$W = cFP$$

by setting $F := \frac{1}{c\,s_c} W P^T$. Since $W$ satisfies (75) we must have

$$c^2\, n_w = \|W\|_F^2 = c^2 \|FP\|_F^2 = c^2 \langle FP, FP \rangle_F = c^2 \langle FPP^T, F \rangle_F = c^2\, s_c \|F\|_F^2, \tag{77}$$

and so $\|F\|_F^2 = n_w/s_c = n_c$.

According to (73), $\hat{U}$ bust be a scalar multiple of the matrix

$$WP^T Z = (cFP)P^T Z = c\, s_c\, FZ$$

Using the fact that $ZZ^T = \frac{KL}{n_c} I$ and $\|F\|_F^2 = n_c$ we then obtain that

$$\|FZ\|_F^2 = \langle FZ, FZ \rangle_F = \langle FZZ^T, F \rangle_F = \frac{KL}{n_c} \|F\|_F^2 = KL \tag{78}$$

and so equation (73) becomes

$$\hat{U} = c\,\sqrt{n_w}\, \frac{WP^T Z}{\|WP^T Z\|_F} = c\,\sqrt{n_w}\, \frac{FZ}{\sqrt{KL}} \tag{79}$$

which concludes the proof. $\qquad\square$

### E.2 PROOF OF COLLAPSE

Recall that the regularized risk associated with the network $h_{W,U}$ is defined by

$$\mathcal{R}(W, U) = \mathcal{R}_0(W, U) + \frac{\lambda}{2} \left( \|W\|_F^2 + \|U\|_F^2 \right) \tag{80}$$

and recall that the set of weights in type-I collapse configuration is

$$\Omega_c^I = \Big\{ (W, U) : \text{There exist an equiangular matrix } \mathfrak{F} \text{ such that}$$

$$W = c\, \mathfrak{F}\, P \quad \text{and} \quad \hat{U} = c\, \sqrt{\frac{n_w}{KL}}\, \mathfrak{F}\, Z \Big\} \tag{81}$$

This subsection is devoted to the proof of the following proposition.

**Proposition A.** *We have the following:*

(i) *If the latent variables $\mathbf{z}_1, \ldots, \mathbf{z}_K$ are mutually distinct and satisfy assumption A, then there exists $c \in \mathbb{R}$ such that*

$$\Omega_c^I \subset \arg\min \mathcal{R}$$

(ii) *If the latent variables $\mathbf{z}_1, \ldots, \mathbf{z}_K$ are mutually distinct and satisfy assumptions A and B, then any $(W, U)$ that minimizes $\mathcal{R}$ must belong to $\Omega_c^I$ for some $c \in \mathbb{R}$.*

This proposition states that, under appropriate symmetry assumption, the weights of the network $h_{W,U}$ do collapse into a type-I configuration. This proposition however does not provide the value of the constant $c$ involved in the collapse. Determining this constant will be done in the subsection E.3.

We start with a simple lemma.

**Lemma G.** *Any global minimizer of (80) must belong to $\mathcal{N}$.*

*Proof.* Let $(W^\star, U^\star)$ be a global minimizer. Define $B = \frac{1}{K}\sum_{k=1}^{K} U_k^\star$ and

$$U_0 = [U_1^\star - B \quad U_2^\star - B \quad \cdots \quad U_K^\star - B]$$

From the definition of the unregularized risk we have $\mathcal{R}_0(W^\star; U_0) = \mathcal{R}_0(W^\star; U^\star)$ and therefore

$$
\frac{1}{K}\left(\mathcal{R}(W^\star; U_0) - \mathcal{R}(W^\star; U^\star)\right) = \frac{\lambda}{2}\frac{1}{K}\sum_{k=1}^{K}\left(\|U_k^\star - B\|_F^2 - \|U_k^\star\|_F^2\right)
$$

$$
= \frac{\lambda}{2}\frac{1}{K}\sum_{k=1}^{K}\left(\|B\|_F^2 - 2\langle B, U_k^\star\rangle_F\right)
$$

$$
= \frac{\lambda}{2}\left(\|B\|_F^2 - 2\left\langle B, \frac{1}{K}\sum_{k=1}^{K}U_k^\star\right\rangle_F\right)
$$

$$
= -\frac{\lambda}{2}\|B\|_F^2
$$

So $B$ must be equal to zero, otherwise we would have $\mathcal{R}(W^\star, U_0) < \mathcal{R}(W^\star, U^\star)$.  $\square$

The next lemma bring together the bilinear optimization problem from subsection E.1 and the sharp lower bound on the unregularized risk that we derived in section D.

**Lemma H.** *Assume the latent variables satisfy assumption A. Assume also that $(W^\star, U^\star)$ is a global minimizer of (80) and let $c \in \mathbb{R}$ be such that*

$$\frac{1}{2}\left(\|W^\star\|_F^2 + \|U^\star\|_F^2\right) = c^2\, n_w.$$

*Then the following hold:*

*(i) Any $(W, U)$ that belongs to $\mathcal{N} \cap \mathcal{E} \cap \mathcal{B}_c^I$ is also a global minimizer of (80).*

*(ii) If $\mathcal{N} \cap \mathcal{E} \cap \mathcal{B}_c^I \neq \emptyset$, then $(W^\star, U^\star)$ must belong to $\mathcal{N} \cap \mathcal{E} \cap \mathcal{B}_c^I$.*

*Proof.* Recall from theorem D that

$$\mathcal{R}_0(W, U) = g\left(-\left\langle \hat{U}, WQ^T Z\right\rangle_F\right) \qquad \text{for all } (W, U) \in \mathcal{N} \cap \mathcal{E} \tag{82}$$

$$\mathcal{R}_0(W, U) > g\left(-\left\langle \hat{U}, WQ^T Z\right\rangle_F\right) \qquad \text{for all } (W, U) \in \mathcal{N} \cap \mathcal{E}^c \tag{83}$$

We start by proving (i). If $(W, U) \in \mathcal{N} \cap \mathcal{E} \cap \mathcal{B}_c^I$, then we have

$$
\mathcal{R}_0(W^\star, U^\star) \geq g\left(-\left\langle \hat{U}^\star, W^\star Q^T Z\right\rangle_F\right) \qquad \text{[because } (W^\star, U^\star) \in \mathcal{N} \text{ due to lemma G ]}
$$

$$
\geq g\left(-\left\langle \hat{U}, WQ^T Z\right\rangle_F\right) \qquad \text{[because } (W, U) \in \mathcal{B}_c^I \text{ and } g \text{ is increasing]}
$$

$$
= \mathcal{R}_0(W, U) \qquad \text{[because } (W, U) \in \mathcal{N} \cap \mathcal{E} \text{ ]}
$$

Since $(W, U) \in \mathcal{B}_c^I$ we must have $\frac{1}{2}\left(\|W\|_F^2 + \|U\|_F^2\right) = c^2\, n_c = \frac{1}{2}\left(\|W^\star\|_F^2 + \|U^\star\|_F^2\right)$. Therefore $\mathcal{R}(W, U) \leq \mathcal{R}(W^\star, U^\star)$ and $(W, U)$ is a minimizer.

We now prove (ii) by contradiction. Suppose that $(W^\star, U^\star) \notin \mathcal{N} \cap \mathcal{E} \cap \mathcal{B}_c^I$. This must mean that

$$(W^\star, U^\star) \notin \mathcal{E} \cap \mathcal{B}_c^I$$

since it clearly belongs to $\mathcal{N}$. If $(W^\star, U^\star) \notin \mathcal{E}$ then the first inequality in the above computation is strict according to (83). If $(W^\star, U^\star) \notin \mathcal{B}_c^I$ then the second inequality is strict because $g$ is strictly increasing.  $\square$

The above lemma establishes connections between the set of minimizers of the risk and the set $\mathcal{E} \cap \mathcal{N} \cap \mathcal{B}_c^I$. The next two lemmas shows that the set $\mathcal{E} \cap \mathcal{N} \cap \mathcal{B}_c^I$ is closely related to the set of collapsed configurations $\Omega_c^I$. In other words we use the set $\mathcal{E} \cap \mathcal{N} \cap \mathcal{B}_c^I$ as a bridge between the set of minimizers and the set of type-I collapse configurations.

**Lemma I.** *If the latent variables satisfy the symmetry assumption A, then*

$$\Omega_c^I \subset \mathcal{E} \cap \mathcal{N} \cap \mathcal{B}_c^I$$

*Proof.* We already know from (67) that $\Omega_c^I \subset \mathcal{B}_c^I$. We now show that $\Omega_c^I \subset \mathcal{E}$. Suppose $(W, U) \in \Omega_c^I$. Then there exists an equiangular matrix $\mathfrak{F} \in \mathbb{R}^{d \times n_c}$ such that

$$W = c\, \mathfrak{F}\, P \qquad \text{and} \qquad \hat{U} = c'\, \mathfrak{F}\, Z$$

where $c' = c\sqrt{n_w/(KL)}$. Recall from (25) that

$$P\zeta(\mathbf{x}) = Z_k \qquad \text{for all } \mathbf{x} \in \mathcal{X}_k.$$

Consider two latent variables

$$\mathbf{z}_k = [\alpha_1, \dots, \alpha_L] \quad \text{and} \quad \mathbf{z}_j = [\alpha_1', \dots, \alpha_L']$$

and assume $\mathbf{x}$ is generated by $\mathbf{z}_k$, meaning that $\mathbf{x} \in \mathcal{X}_k$. We then have

$$\begin{aligned}
\mathfrak{M}_{W,U}(\mathbf{x}, j) &= \left\langle \hat{U}_k - \hat{U}_j, W\zeta(\mathbf{x}) \right\rangle_F \\
&= cc' \left\langle \mathfrak{F}\, Z_k - \mathfrak{F}\, Z_j, \mathfrak{F}\, P\zeta(\mathbf{x}) \right\rangle_F \\
&= cc' \left\langle \mathfrak{F}\, Z_k - \mathfrak{F}\, Z_j, \mathfrak{F}\, Z_k \right\rangle_F \\
&= cc' \sum_{\ell=1}^L \left\langle \mathfrak{f}_{\alpha_\ell} - \mathfrak{f}_{\alpha_\ell'}, \mathfrak{f}_{\alpha_\ell} \right\rangle_F \\
&= cc' \left( L - \sum_{\ell=1}^L \left\langle \mathfrak{f}_{\alpha_\ell'}, \mathfrak{f}_{\alpha_\ell} \right\rangle_F \right)
\end{aligned}$$

Since $\mathfrak{f}_1, \dots, \mathfrak{f}_{n_c}$ are equiangular, we have

$$\sum_{\ell=1}^L \left\langle \mathfrak{f}_{\alpha_\ell'}, \mathfrak{f}_{\alpha_\ell} \right\rangle_F = \left( L - \text{dist}(\mathbf{z}_j, \mathbf{z}_k) \right) - \frac{1}{n_c - 1}\text{dist}(\mathbf{z}_j, \mathbf{z}_k) = L - \frac{n_c}{n_c - 1}\text{dist}(\mathbf{z}_j, \mathbf{z}_k).$$

Therefore

$$\mathfrak{M}_{W,U}(\mathbf{x}, j) = cc'\frac{n_c}{n_c - 1}\text{dist}(\mathbf{z}_j, \mathbf{z}_k)$$

and it is clear that the margin only depends on $\text{dist}(\mathbf{z}_j, \mathbf{z}_k)$, and therefore $(W, U)$ satisfies the equimargin property.

Finally we show that $\Omega_c^I \subset \mathcal{N}$. Suppose $(W, U) \in \Omega_c^I$. From property (ii) of lemma B we have

$$\sum_{k=1}^K Z_k = \frac{K}{n_c}\mathbf{1}_{n_c}\mathbf{1}_L^T$$

Therefore,

$$\sum_{k=1}^K \hat{U}_k = c' \sum_{k=1}^K \mathfrak{F}\, Z_k = c'\, \frac{K}{n_c}\mathfrak{F}\, \mathbf{1}_{n_c}\mathbf{1}_L^T = 0$$

where we have used the fact that $\mathfrak{F}\, \mathbf{1}_{n_c} = 0$. $\qquad\square$

**Lemma J.** *If the latent variables satisfy assumptions A and B, then*

$$\Omega_c^I = \mathcal{E} \cap \mathcal{N} \cap \mathcal{B}_c^I$$

*Proof.* From the previous lemma we know that $\Omega_c^I \subset \mathcal{E} \cap \mathcal{N} \cap \mathcal{B}_c^I$ so we need to show that

$$\mathcal{E} \cap \mathcal{N} \cap \mathcal{B}_c^I \subset \Omega_c^I.$$

Let $(W, U) \in \mathcal{E} \cap \mathcal{N} \cap \mathcal{B}_c^I$. Since $(W, U)$ belongs to $\mathcal{B}_c^I$, there exists a matrix $F \in \mathbb{R}^{d \times n_c}$ with $\|F\|_F^2 = n_c$ such that

$$W = c \, F \, P \qquad \text{and} \qquad U = c' \, F \, Z \tag{84}$$

where $c' = c\sqrt{n_w/(KL)}$. Our goal is to show that $F$ is equiangular, meaning that it satisfies the two relations

$$F \, \mathbf{1}_{n_c} = 0 \qquad \text{and} \qquad F^T F = \frac{n_c}{n_c - 1} \, I_{n_c} - \frac{1}{n_c - 1} \, \mathbf{1}_{n_c} \mathbf{1}_{n_c}^T. \tag{85}$$

The first relation is easily obtained. Indeed, using the fact that $(W, U) \in \mathcal{N}$ together with the identity $\sum_{k=1}^K Z_k = \frac{K}{n_c} \mathbf{1}_{n_c} \mathbf{1}_L^T$ (which hold due to lemma B), we obtain

$$0 = \sum_{k=0}^K U_k = c' \sum_{k=0}^K F Z_k = c' \frac{K}{n_c} F \mathbf{1}_{n_c} \mathbf{1}_L^T.$$

We then note that the matrix $F \mathbf{1}_{n_c} \mathbf{1}_L^T$ is the zero matrix if and only if $F \mathbf{1}_{n_c} = 0$.

We now prove the second equality of (85). Assume that $\mathbf{x} \in \mathcal{X}_k$. Using the fact that $P\zeta(\mathbf{x}) = Z_k$ together with (84), we obtain

$$\begin{aligned}
\mathfrak{M}_{W,U}(\mathbf{x}, j) &= \left\langle \hat{U}_k - \hat{U}_j, W\zeta(\mathbf{x}) \right\rangle_F \\
&= c \, c' \, \langle F \, Z_k - F \, Z_j, F \, P\zeta(\mathbf{x}) \rangle_F \\
&= c \, c' \, \langle F \, Z_k - F \, Z_j, F \, Z_k \rangle_F \\
&= c \, c' \, \langle F^T F(Z_k - Z_j), Z_k \rangle_F \\
&= c \, c' \, \left\langle F^T F, \, \Gamma^{(k,j)} \right\rangle_F
\end{aligned} \tag{86}$$

We recall that the matrices

$$\Gamma^{(k,j)} = Z_k (Z_k - Z_j)^T \in \mathbb{R}^{n_c \times n_c}.$$

are precisely the ones involved in the statement of assumption B. Since $(W, U) \in \mathcal{E}$, the margins must only depend on the distance between the latent variables. Due to (86), we can be express this as

$$\left\langle F^T F, \, \Gamma^{(j,k)} \right\rangle_F = \left\langle F^T F, \, \Gamma^{(j',k')} \right\rangle_F \qquad \forall j, k, j', k' \in [K] \text{ s.t. } \mathrm{dist}(\mathbf{z}_j, \mathbf{z}_k) = \mathrm{dist}(\mathbf{z}_{j'}, \mathbf{z}_{k'})$$

Since the $F^T F$ is clearly positive semi-definite, we may then use assumption B to conclude that $F^T F \in \mathcal{A}$. Recalling definition (42) of the set $\mathcal{A}$, we therefore have

$$F^T F = a \, I_{n_c} + b \, \mathbf{1}_{n_c} \mathbf{1}_{n_c}^T \tag{87}$$

for some $a, b \in \mathbb{R}$. To conclude our proof, we need to show that

$$a = \frac{n_c}{n_c - 1} \qquad \text{and} \qquad b = -\frac{1}{n_c - 1}. \tag{88}$$

Combining (87) with the first equality of (85), we obtain

$$0 = F^T F \mathbf{1}_{n_c} = a \, \mathbf{1}_{n_c} + b \, \mathbf{1}_{n_c} \mathbf{1}_{n_c}^T \mathbf{1}_{n_c} = (a + b n_c) \mathbf{1}_{n_c} \tag{89}$$

Combining (87) with the fact that $\|F\|_F^2 = n_c$, we obtain

$$n_c = \|F\|_F^2 = \mathrm{Tr}(F^T F) = n_c(a + b) \tag{90}$$

The constants $a, b \in \mathbb{R}$, according to (89) and (90) must therefore solve the system

$$\begin{cases} a + b n_c &= 0 \\ a + b &= 1 \end{cases}$$

and one can easily check that the solution of this system is precisely given by (88). $\qquad\square$

We conlude this subsection by proving proposition A.

*Proof of Proposition A.* Let $(W^\star, U^\star)$ be a global minimizer of $\mathcal{R}$ and let $c \in \mathbb{R}$ be such that

$$\frac{1}{2}\left(\|W^\star\|_F^2 + \|U^\star\|_F^2\right) = c^2\, n_w$$

If the latent variables satisfies assumption A, we can use lemma I together with the first statement of lemma H to obtain

$$\Omega_c^I \subset \mathcal{E} \cap \mathcal{N} \cap \mathcal{B}_c^I \subset \arg\min \mathcal{R},$$

which is precisely statement (i) of the proposition.

We now prove statement (ii) of the proposition. If the latent variables satisfies assumption A and B then lemma J asserts that

$$\Omega_c^I = \mathcal{E} \cap \mathcal{N} \cap \mathcal{B}_c^I$$

The set $\Omega_c^I$ is clearly not empty (because the set of equiangular matrices is not empty), and we may therefore use the second statement of lemma H to obtain that

$$(W^\star, U^\star) \in \mathcal{E} \cap \mathcal{N} \cap \mathcal{B}_c^I = \Omega_c^I$$

$\square$

### E.3    DETERMINING THE CONSTANT $c$

The next lemma provides an explicit formula for the regularized risk of a network whose weights are in type-I collapse configuration with constant $c$.

**Lemma K.** *Assume the latent variables satisfy assumption A. If the pair of weights $(W, U)$ belongs to $\Omega_c^I$, then*

$$\mathcal{R}(W, U) = \log\left(1 - \frac{K}{n_c^L} + \frac{K}{n_c^L}\left(1 + (n_c - 1)e^{-\eta\, n_w c^2}\right)^L\right) + \lambda\, n_w c^2 \tag{91}$$

*where $\eta = \frac{n_c}{n_c - 1}\sqrt{\frac{1}{n_w K L}}$.*

From the above lemma it is clear that if the pair $(W, U) \in \Omega_c^I$ minimizes $\mathcal{R}$, then the constant $c$ must minimize the right hand side of (91). Therefore combining lemma K with proposition A concludes the proof of theorem E.

**Remark**    In the previous subsections, we only relied on relations (i), (ii) and (iii) of lemma B to prove collapse. Assumption A was never fully needed. In this section however, in order to determine the specific values of the constant involved in the collapse, we will need the actual combinatorial values provided by assumption A.

The remainder of this section is devoted to the proof of lemma K.

*Proof of lemma K.* Recall from (45) that the unregularized risk can be expressed as

$$\mathcal{R}_0(W, U) = \frac{1}{K}\sum_{k=1}^K \sum_{\mathbf{x} \in \mathcal{X}_k} \log\left(1 + \sum_{j \neq k} e^{-\mathfrak{M}_{W,U}(\mathbf{x}, j)}\right)\mathcal{D}_{\mathbf{z}_k}(\mathbf{x})$$

We also recall that the set $\Omega_c^I$ is given by

$$\Omega_c^I = \Big\{(W, U) : \text{There exist an equiangular matrix } \mathfrak{F} \text{ such that}$$

$$W = c\,\mathfrak{F}\,P \quad \text{and} \quad \hat{U} = c\,\sqrt{\frac{n_w}{KL}}\,\mathfrak{F}\,Z\Big\} \tag{92}$$

and that $P\zeta(\mathbf{x}) = Z_k$ for all $\mathbf{x} \in \mathcal{X}_k$ (see equation (25) from section B). Consider two latent variables

$$\mathbf{z}_k = [\alpha_1, \ldots, \alpha_L] \quad \text{and} \quad \mathbf{z}_j = [\alpha_1', \ldots, \alpha_L']$$

and assume $\mathbf{x}$ is generated by $\mathbf{z}_k$, meaning that $\mathbf{x} \in \mathcal{X}_k$.

$$\mathfrak{M}_{W,U}(\mathbf{x},j) = \left\langle \hat{U}_k - \hat{U}_j, W\zeta(\mathbf{x}) \right\rangle_F$$

$$= c^2 \sqrt{\frac{n_w}{KL}} \; \langle \mathfrak{F} \, Z_k - \mathfrak{F} \, Z_j, \mathfrak{F} \, P\zeta(\mathbf{x}) \rangle_F$$

$$= c^2 \sqrt{\frac{n_w}{KL}} \; \langle \mathfrak{F} \, Z_k - \mathfrak{F} \, Z_j, \mathfrak{F} \, Z_k \rangle_F$$

$$= c^2 \sqrt{\frac{n_w}{KL}} \; \sum_{\ell=1}^{L} \left\langle \mathfrak{f}_{\alpha_\ell} - \mathfrak{f}_{\alpha'_\ell}, \mathfrak{f}_{\alpha_\ell} \right\rangle_F$$

$$= c^2 \sqrt{\frac{n_w}{KL}} \; \left( L - \sum_{\ell=1}^{L} \left\langle \mathfrak{f}_{\alpha'_\ell}, \mathfrak{f}_{\alpha_\ell} \right\rangle_F \right)$$

Since $\mathfrak{f}_1, \ldots, \mathfrak{f}_{n_c}$ are equiangular, we have

$$\sum_{\ell=1}^{L} \left\langle \mathfrak{f}_{\alpha'_\ell}, \mathfrak{f}_{\alpha_\ell} \right\rangle_F = \left( L - \mathrm{dist}(\mathbf{z}_j, \mathbf{z}_k) \right) - \frac{1}{n_c - 1} \mathrm{dist}(\mathbf{z}_j, \mathbf{z}_k) = L - \frac{n_c}{n_c - 1} \mathrm{dist}(\mathbf{z}_j, \mathbf{z}_k).$$

Therefore

$$\mathfrak{M}_{W,U}(\mathbf{x},j) = c^2 \sqrt{\frac{n_w}{KL}} \frac{n_c}{n_c - 1} \mathrm{dist}(\mathbf{z}_j, \mathbf{z}_k)$$

Letting $\omega = \sqrt{\frac{n_w}{KL}} \frac{n_c}{n_c-1}$ we therefore obtain

$$\mathcal{R}_0(W,U) = \frac{1}{K} \sum_{k=1}^{K} \sum_{\mathbf{x} \in \mathcal{X}_k} \log \left( 1 + \sum_{j \neq k} e^{-\omega c^2 \mathrm{dist}(\mathbf{z}_j, \mathbf{z}_k)} \right) \mathcal{D}_{\mathbf{z}_k}(\mathbf{x})$$

$$= \frac{1}{K} \sum_{k=1}^{K} \log \left( 1 + \sum_{j \neq k} e^{-\omega c^2 \mathrm{dist}(\mathbf{z}_j, \mathbf{z}_k)} \right) \tag{93}$$

where we have used the quantity inside the log does not depends on $\mathbf{x}$. We proved in section C (see equation (38)) that if the latent variables satisfy assumption A, then

$$|S_r| = \frac{K}{n_c^L} \binom{L}{r} (n_c - 1)^r$$

Using this identity we obtain

$$\sum_{j \neq k} e^{-\omega c^2 \mathrm{dist}(\mathbf{z}_j, \mathbf{z}_k)} = \sum_{r=1}^{L} |\{j : \mathrm{dist}(\mathbf{z}_j, \mathbf{z}_k) = r\}| \; e^{-\omega c^2 r}$$

$$= \frac{K}{n_c^L} \sum_{r=1}^{L} \binom{L}{r} (n_c - 1)^r \, e^{-\omega c^2 r}$$

$$= -\frac{K}{n_c^L} + \frac{K}{n_c^L} \sum_{r=0}^{L} \binom{L}{r} (n_c - 1)^r \, e^{-\omega c^2 r}$$

$$= -\frac{K}{n_c^L} + \frac{K}{n_c^L} \left( 1 + (n_c - 1)e^{-\omega c^2} \right)^L$$

where we have used the binomial theorem to obtain the last equality. The above quantity does not depends on $k$, therefore (93) can be expressed as

$$\mathcal{R}_0(W,U) = \log \left( 1 - \frac{K}{n_c^L} + \frac{K}{n_c^L} \left( 1 + (n_c - 1)e^{-\omega c^2} \right)^L \right)$$

We then remark that the matrix $\mathfrak{F} \, P$ has $n_w$ columns, and that each of these columns has norm 1. Similarly, the $\mathfrak{F} \, Z$ has $KL$ columns of length 1. We therefore have

$$\frac{1}{2} \left( \|W\|_F^2 + \|\hat{U}\|_F^2 \right) = \frac{1}{2} \left( c^2 \|\mathfrak{F} \, P\|_F^2 + c^2 \frac{n_w}{KL} \|\mathfrak{F} \, Z\|_F^2 \right) = c^2 n_w.$$

To conclude the proof we simply remark that $\omega = n_w \eta$. $\qquad \square$

# F   PROOF OF THEOREM 3 AND ITS CONVERSE

In this section we prove theorem 3 under assumption A, and its converse under assumptions A and B. We start by recalling the definition of a type-II collapse configuration.

**Definition E** (Type-II Collapse). *The weights $(W, U)$ of the network $h_{W,U}^*$ form a type-II collapse configuration if and only if the conditions*

   i) $\varphi(\mathbf{w}_{(\alpha,\beta)}) = \sqrt{d}\,\mathfrak{f}_\alpha$   *for all* $(\alpha, \beta) \in \mathcal{V}$.

   ii) *There exists $c \geq 0$ so that $\mathbf{u}_{k,\ell} = c\,\mathfrak{f}_\alpha$   for all $(k, \ell)$ satisfying $z_{k,\ell} = \alpha$ and all $\alpha \in \mathcal{C}$.*

*hold for some collection $\mathfrak{f}_1, \ldots, \mathfrak{f}_{n_c} \in \mathbb{R}^d$ of mean-zero equiangular vectors.*

As in the previous section we will reformulate the above definition using matrix notations. Toward this aim we make the following definition:

**Definition F.** *(Mean-Zero Equiangular Matrices) A matrix $\mathfrak{F} \in \mathbb{R}^{d \times n_c}$ is said to be a mean-zero equiangular matrix if and only if the relations*

$$\mathbf{1}_d^T\,\mathfrak{F} = 0, \qquad \mathfrak{F}\,\mathbf{1}_{n_c} = 0 \qquad and \qquad \mathfrak{F}^T\mathfrak{F} = \frac{n_c}{n_c - 1}\,I_{n_c} - \frac{1}{n_c - 1}\,\mathbf{1}_{n_c}\mathbf{1}_{n_c}^T$$

*hold.*

Comparing the above definition with the definition of equiangular vectors provided in the main paper, we easily see that $\mathfrak{F}$ is a mean-zero equiangular matrix if and only if its columns are mean-zero equiangular vectors. Relations (i) and (ii) of definition F can be conveniently expressed as

$$\varphi(W) = \sqrt{d}\,\mathfrak{F}\,P \qquad and \qquad \hat{U} = c\,\mathfrak{F}\,Z$$

for some equiangular matrix $\mathfrak{F}$. We then set

$$\Omega_c^{II} = \Big\{(W, U) : \text{There exist a mean-zero equiangular matrix } \mathfrak{F} \text{ such that}$$

$$\varphi(W) = \sqrt{d}\,\mathfrak{F}\,P \quad \text{and} \quad \hat{U} = c\,\mathfrak{F}\,Z\Big\} \quad (94)$$

and note that $\Omega_c^{II}$ is simply the set of weights $(W, U)$ which are in a type-II collapse configuration. We now state the main theorem of this section.

**Theorem F.** *Assume the non-degenerate condition $\mu_\beta > 0$ holds. Let $\tau \geq 0$ denote the unique minimizer of the strictly convex function*

$$H^*(t) = \log\left(1 - \frac{K}{n_c^L} + \frac{K}{n_c^L}\left(1 + (n_c - 1)e^{-\eta^* t}\right)^L\right) + \frac{\lambda}{2}t^2 \qquad \text{where } \eta^* = \frac{n_c}{n_c - 1}\,\frac{1}{\sqrt{KL/d}}$$

*and let $c = \tau/\sqrt{KL}$. Then we have the following:*

   (i) *If the latent variables $\mathbf{z}_1, \ldots, \mathbf{z}_K$ are mutually distinct and satisfy assumption A, then*

$$\Omega_c^{II} \subset \arg\min \mathcal{R}^*$$

   (ii) *If the latent variables $\mathbf{z}_1, \ldots, \mathbf{z}_K$ are mutually distinct and satisfy assumptions A and B, then*

$$\Omega_c^{II} = \arg\min \mathcal{R}^*$$

Note that statement (i) corresponds to theorem 3 of the main paper, whereas statement (ii) can be viewed as its converse. To prove F we will follow the same steps than in the previous section. The main difference occurs in the study of the bilinear problem, as we will see in the next subsection. We will assume

$$\mu_\beta > 0$$

everywhere in this section — all lemmas and propositions are proven under this assumption, even when not explicitly stated.

Before to go deeper in our study let us state a very simple lemma that expresses the regularized risk $\mathcal{R}^*$ associated with network $h^*$ in term of the function $\mathcal{R}_0$ defined by equation (44).

**Lemma L.** *Given a pair of weights $(W, U)$, we have*

$$\mathcal{R}^*(W, U) = \mathcal{R}_0\Big(\varphi(W), U\Big) + \frac{\lambda}{2}\|U\|_F^2 \tag{95}$$

*Proof.* Recall from section B that

$$h_{W,U}(\mathbf{x}) = U \operatorname{Vec}\left[W\zeta(\mathbf{x})\right]$$
$$h^*_{W,U}(\mathbf{x}) = U \operatorname{Vec}\left[\varphi\Big(W\zeta(\mathbf{x})\Big)\right]$$

Note that since $\zeta(\alpha, \beta)$ is a one hot vector, we obviously have that $\varphi(W\zeta(\alpha, \beta)) = \varphi(W)\zeta(\alpha, \beta)$. Therefore the the network $h^*$ and $h$ are related as follow:

$$h^*_{W,U}(\mathbf{x}) = U \operatorname{Vec}\left[\varphi\Big(W\zeta(\mathbf{x})\Big)\right] = U \operatorname{Vec}\left[\varphi(W)\zeta(\mathbf{x})\right] = h_{\varphi(W), U}(\mathbf{x})$$

As a consequence, the regularized risk associated with the network $h^*_{W,U}$ can be expressed as

$$\mathcal{R}^*(W, U) = \frac{1}{K}\sum_{k=1}^K \mathbb{E}_{\mathbf{x}\sim\mathcal{D}_{\mathbf{z}_k}}\left[\ell(h^*_{W,U}(\mathbf{x}), k)\right] + \frac{\lambda}{2}\|U\|_F^2$$

$$= \frac{1}{K}\sum_{k=1}^K \mathbb{E}_{\mathbf{x}\sim\mathcal{D}_{\mathbf{z}_k}}\left[\ell(h_{\varphi(W),U}(\mathbf{x}), k)\right] + \frac{\lambda}{2}\|U\|_F^2$$

$$= \mathcal{R}_0(\varphi(W), U) + \frac{\lambda}{2}\|U\|_F^2$$

where $\mathcal{R}_0$ is the unregularized risk defined in (44). $\qquad\square$

### F.1 The bilinear optimization problem

Let

$$\operatorname{Range}(\varphi) = \{V \in \mathbb{R}^{d\times n_w} : \text{There exist } W \in \mathbb{R}^{d\times n_w} \text{ such that } V = \varphi(W)\}$$

and consider the optimization problem

$$\text{maximize} \quad \left\langle \hat{U}, VQ^TZ\right\rangle_F \tag{96}$$

$$\text{subject to} \quad V \in \operatorname{Range}(\varphi) \quad \text{and} \quad \|\hat{U}\|_F^2 = KLc^2 \tag{97}$$

where the optimization variables are the matrix $V \in \mathbb{R}^{d\times n_w}$ and the matrix $\hat{U} \in \mathbb{R}^{d\times KL}$.

**Lemma M.** *Assume the latent variables satisfy assumption A. Then $(V, U)$ is a solution of the optimization problem (96) – (97) if and only if it belongs to the set*

$$\mathcal{B}_c^{II} = \Big\{(V, U) : \text{There exist a matrix } F \in \mathcal{F} \text{ such that } V = \sqrt{d}\,FP \text{ and } \hat{U} = c\,FZ\Big\} \tag{98}$$

*where $\mathcal{F}$ denotes the set of matrices whose columns have unit length and mean zero, that is*

$$\mathcal{F} = \{F \in \mathbb{R}^{d\times n_c} : \mathbf{1}_d^T F = 0 \text{ and the columns of } F \text{ have unit length}\}.$$

The remainder of this subsection is devoted to the proof of the above lemma.

We start by showing that all $(V, U) \in \mathcal{B}_c^{II}$ have same objective values and satisfy the constraints.

**Claim D.** *If $(V, U) \in \mathcal{B}_c^{II}$, then*

$$V \in \operatorname{Range}(\varphi) \quad, \quad \|\hat{U}\|_F^2 = KLc^2, \quad \text{and} \quad \left\langle \hat{U}, VQ^TZ\right\rangle_F = c\sqrt{d}\,KL$$

*Proof.* Assume $(V, U) \in \mathcal{B}_c^{II}$. Since the columns of $P$ are one hot vectors in $\mathbb{R}^{n_c}$, the columns of $FP$ have unit length and mean zero. Therefore the columns of $V$ have norm equal to $\sqrt{d}$ and mean zero. Therefore $V \in \operatorname{Range}(\varphi)$.

Using $ZZ^T = \frac{KL}{n_c}I$ from lemma B, together with the fact that $\|F\|_F^2 = n_c$ since its columns have unit length, we obtain

$$\|FZ\|_F^2 = \langle FZ, FZ \rangle_F = \langle FZZ^T, F \rangle_F = \left( \frac{KL}{n_c} \right) \|F\|_F^2 = KL \tag{99}$$

As a consequence we have $\|\hat{U}\|_F^2 = c^2\,KL$. Finally, note that

$$PQ^T = I_{n_c}$$

as can clearly be seen from formulas (23) and (26). We therefore have

$$\left\langle \hat{U}, VQ^T Z \right\rangle_F = c\sqrt{d}\left\langle FZ, FPQ^T Z \right\rangle_F = c\sqrt{d}\left\langle FZ, FZ \right\rangle_F = c\sqrt{d}\,KL$$

$$\square$$

We then prove that

**Claim E.** *If $(V, \hat{U})$ is a solution of (96) – (97), then $(V, \hat{U}) \in \mathcal{B}_c^{II}$.*

Note that according to the first claim, all $(V, \hat{U}) \in \mathcal{B}_c^{II}$ have same objective value, and therefore, according to the above claim, they must all be maximizer. As a consequence, proving the above claim will conclude the proof of lemma M.

*Proof of the claim.* Maximizing (96) – (97) over $\hat{U}$ first gives

$$\hat{U} = c\sqrt{KL}\, \frac{VQ^T Z}{\|VQ^T Z\|_F} \tag{100}$$

and therefore the optimization problem reduces to

$$\text{maximize} \quad \|VQ^T Z\|_F^2 \tag{101}$$
$$\text{subject to} \quad V \in \text{Range}(\varphi) \tag{102}$$

Using the fact that $ZZ^T = \frac{KL}{n_c}I$ we then get

$$\|VQ^T Z\|_F^2 = \langle VQ^T Z, VQ^T Z \rangle_F = \langle VQ^T ZZ^T, VQ^T \rangle_F = \frac{KL}{n_c}\|VQ^T\|_F^2 \tag{103}$$

and so the problem further reduces to

$$\text{maximize} \quad \|VQ^T\|_F^2 \tag{104}$$
$$\text{subject to} \quad V \in \text{Range}(\varphi) \tag{105}$$

Let us define

$$\mathbf{v}_{(\alpha,\beta)} := V\zeta(\alpha, \beta)$$

In other words $\mathbf{v}_{(\alpha,\beta)}$ is the $j^{th}$ column of $V$, where $j = (\alpha - 1)s_c + \beta$. The KKT conditions for the optimization problem (104) – (105) then amount to solving the system

$$VQ^T Q = VD_\nu + \mathbf{1}_d\,\boldsymbol{\lambda}^T \tag{106}$$
$$\langle \mathbf{v}_{(\alpha,\beta)}, \mathbf{1}_d \rangle = 0 \qquad \text{for all } (\alpha, \beta) \in \mathcal{V} \tag{107}$$
$$\|\mathbf{v}_{(\alpha,\beta)}\|^2 = d \qquad \text{for all } (\alpha, \beta) \in \mathcal{V} \tag{108}$$

for $D_\nu$ some $n_w \times n_w$ diagonal matrix of Lagrange multipliers for the constraint (108) and $\boldsymbol{\lambda} \in \mathbb{R}^{n_w}$ a vector of Lagrange multipliers for the mean zero constraints. Left multiplying the first equation by $\mathbf{1}_d^T$ and using the second shows $\boldsymbol{\lambda} = \mathbf{0}_{n_w}$, and so it proves equivalent to find solutions of the reduced system

$$VQ^T Q = VD_\nu \tag{109}$$
$$\langle \mathbf{v}_{(\alpha,\beta)}, \mathbf{1}_d \rangle = 0 \qquad \text{for all } (\alpha, \beta) \in \mathcal{V} \tag{110}$$
$$\|\mathbf{v}_{(\alpha,\beta)}\|^2 = d \qquad \text{for all } (\alpha, \beta) \in \mathcal{V} \tag{111}$$

instead. Recalling the identity $Q\,\zeta(\alpha,\beta) = \mu_\beta \mathbf{e}_\alpha$ (see (27) in section B) we obtain

$$Q^T Q\,\zeta(\alpha,\beta) = \mu_\beta\,Q^T\,\mathbf{e}_\alpha$$

and so right multiplying (109) by $\zeta(\alpha,\beta)$ gives

$$V Q^T\,\mathbf{e}_\alpha = \frac{\nu(\alpha,\,\beta)}{\mu_\beta}\,\mathbf{v}_{(\alpha,\,\beta)} \qquad \text{for all } (\alpha,\beta) \in \mathcal{V}$$

where we have denoted by $\nu(\alpha,\beta)$ the Lagrange multiplier corresponding to the constraint (111). Define the support sets

$$\Xi_\alpha := \{\beta \in [s_c] : \nu(\alpha,\,\beta) \neq 0\} \qquad \text{and} \qquad \Xi := \{\alpha : \Xi_\alpha \neq \emptyset\}$$

of the Lagrange multipliers. If $\alpha \in \Xi$ then imposing the norm constraint (111) gives

$$\|V Q^T\,\mathbf{e}_\alpha\| = \frac{\nu(\alpha,\,\beta)}{\mu_\beta}\sqrt{d},$$

and so $\|V Q^T\,\mathbf{e}_\alpha\| > 0$ if $\alpha \in \Xi$ since $\nu(\alpha,\,\beta) > 0$ for some $\beta \in [s_c]$ by definition. This implies that the relation

$$\mathbf{v}_{(\alpha,\,\beta)} = \sqrt{d}\,\frac{V Q^T\,\mathbf{e}_\alpha}{\|V Q^T\,\mathbf{e}_\alpha\|} \qquad \text{for all} \qquad (\alpha,\,\beta) \in \Xi \times [s_c]$$

must hold. As a consequence there exist mean-zero, unit length vectors $\mathbf{f}_1, \ldots, \mathbf{f}_{n_c}$ (namely the normalized $V Q^T\,\mathbf{e}_\alpha$) so that

$$\mathbf{v}_{(\alpha,\,\beta)} = \sqrt{d}\,\mathbf{f}_\alpha$$

holds for all pairs $(\alpha,\beta)$ with $\alpha \in \Xi$. Taking a look at (26), we easily see that its $\alpha^{th}$ row of the matrix $Q$ can be written as $Q^T\,\mathbf{e}_\alpha = \sum_\beta \mu_\beta \zeta(\alpha,\beta)$, and therefore

$$V Q^T\,\mathbf{e}_\alpha = \sum_{\beta \in [s_c]} \mu_\beta V \zeta(\alpha,\beta) = \sum_{\beta \in [s_c]} \mu_\beta \mathbf{v}_{(\alpha,\,\beta)} = \sqrt{d}\,\mathbf{f}_\alpha\left(\sum_{\beta \in [s_c]} \mu_\beta\right) = \sqrt{d}\,\mathbf{f}_\alpha$$

holds as well. If $\alpha \notin \Xi$ then $V Q^T\,\mathbf{e}_\alpha = \mathbf{0}$ since the corresponding Lagrange multiplier vanishes. It therefore follows that

$$\|V Q^T\|_F^2 = \sum_{\alpha \in [n_c]} \|V Q^T \mathbf{e}_\alpha\|^2 = d\sum_{\alpha \in \Xi} \|\mathbf{f}_\alpha\|^2 = d\,|\Xi|$$

and so global maximizers of (104) – (105) must have full support. In other words, there exist mean-zero, unit-length vectors $\mathbf{f}_1, \ldots, \mathbf{f}_{n_c}$ so that

$$\mathbf{v}_{(\alpha,\,\beta)} = \sqrt{d}\,\mathbf{f}_\alpha \tag{112}$$

holds. Equivalently $V = \sqrt{d}\,FP$ for some $F \in \mathcal{F}$. We then recover $\hat{U}$ using (100).

$$\hat{U} = c\sqrt{KL}\,\frac{V Q^T Z}{\|V Q^T Z\|_F} = c\sqrt{KL}\,\frac{FPQ^T Z}{\|FPQ^T Z\|_F} = c\sqrt{KL}\,\frac{FZ}{\|FZ\|_F} \tag{113}$$

where we have used the fact that $PQ^T = I_{n_c}$. To conclude the proof, we use the fact $\|FZ\|_F = \sqrt{KL}$, as was shown in (99). $\qquad\square$

## F.2  PROOF OF COLLAPSE

Recall from lemma L that the regularized risk associated with the network $h^*_{W,U}$ can be expressed as

$$\mathcal{R}^*(W,U) = \mathcal{R}_0\Big(\varphi(W),\,U\Big) + \frac{\lambda}{2}\|U\|_F^2 \tag{114}$$

and recall that the set of weights in type-II collapse configuration is

$$\Omega_c^{II} = \Big\{(W,U) : \text{There exist a mean-zero equiangular matrix } \mathfrak{F} \text{ such that}$$

$$\varphi(W) = \sqrt{d}\,\mathfrak{F}\,P \quad \text{and} \quad \hat{U} = c\,\mathfrak{F}\,Z\Big\} \tag{115}$$

This subsection is devoted to the proof of the following proposition.

**Proposition B.** *We have the following:*

    *(i) If the latent variables $\mathbf{z}_1, \ldots, \mathbf{z}_K$ are mutually distinct and satisfy assumption A, then there exists $c \in \mathbb{R}$ such that*

$$\Omega_c^{II} \subset \arg\min \mathcal{R}^*$$

    *(ii) If the latent variables $\mathbf{z}_1, \ldots, \mathbf{z}_K$ are mutually distinct and satisfy assumptions A and B, then any $(W, U)$ that minimizes $\mathcal{R}^*$ must belong to $\Omega_c^{II}$ for some $c \in \mathbb{R}$.*

As in the previous section, we have the following lemma.

**Lemma N.** *Any global minimizer of (114) must belong to $\mathcal{N}$.*

The proof is identical to the proof of lemma G. The next lemma bring together the bilinear optimization problem from subsection F.1 and the sharp lower bound on the unregularized risk that we derived in section D.

**Lemma O.** *Assume the latent variables satisfy assumption A. Assume also that $(W^\star, U^\star)$ is a global minimizer of (114) and let $c \in \mathbb{R}$ be such that*

$$\|U^\star\|_F^2 = K L c^2$$

*The the following hold:*

    *(i) Any $(W, U)$ that satisfies*

$$(\varphi(W), U) \in \mathcal{N} \cap \mathcal{E} \cap \mathcal{B}_c^{II}$$

    *is also a global minimizer of $\mathcal{R}^*$.*

    *(ii) If $\mathcal{N} \cap \mathcal{E} \cap \mathcal{B}_c^{II} \neq \emptyset$, then*

$$(\varphi(W^\star), U^\star) \in \mathcal{N} \cap \mathcal{E} \cap \mathcal{B}_c^{II}$$

*Proof.* Recall from theorem D that

$$\mathcal{R}_0(V, U) = g\left( -\left\langle \hat{U}, V Q^T Z \right\rangle_F \right) \qquad \text{for all } (V, U) \in \mathcal{N} \cap \mathcal{E} \tag{116}$$

$$\mathcal{R}_0(V, U) > g\left( -\left\langle \hat{U}, V Q^T Z \right\rangle_F \right) \qquad \text{for all } (V, U) \in \mathcal{N} \cap \mathcal{E}^c \tag{117}$$

We start by proving (i). Define $V^\star = \varphi(W^\star)$, and assume that $U, V, W$ are such that $\varphi(W) = V$ and $(V, U) \in \mathcal{N} \cap \mathcal{E} \cap \mathcal{B}_c$. Then we have

$$
\begin{aligned}
\mathcal{R}_0(\varphi(W^\star), U^\star) &= \mathcal{R}_0(V^\star, U^\star) \\
&\geq g\left( -\langle U^\star, V^\star Q Z \rangle_F \right) && [\text{because } (V^\star, U^\star) \in \mathcal{N}\,] \\
&\geq g\left( -\langle U, V Q Z \rangle_F \right) && [\text{because } (V, U) \in \mathcal{B}_c^{II}\,] \\
&= \mathcal{R}_0(V, U) && [\text{because } (V, U) \in \mathcal{N} \cap \mathcal{E}\,] \\
&= \mathcal{R}_0(\varphi(W), U)
\end{aligned}
$$

Since $\|U\|_F^2 = K L c^2 = \|U^\star\|_F^2$, we have $\mathcal{R}^*(W, U) \leq \mathcal{R}^*(W^\star, U^\star)$ and therefore $(W, U)$ is a minimizer.

We now prove (ii) by contradiction. Suppose that $(\varphi(W^\star), U^\star) \notin \mathcal{N} \cap \mathcal{E} \cap \mathcal{B}_c^{II}$. This must mean that

$$(\varphi(W^\star), U^\star) \notin \mathcal{E} \cap \mathcal{B}_c^{II}$$

since it clearly belongs to $\mathcal{N}$. If $(\varphi(W^\star), U^\star) \notin \mathcal{E}$ then the first inequality in the above computation is strict according to (117). If $(\varphi(W^\star), U^\star) \notin \mathcal{B}_c^{II}$ then the second inequality is strict because $g$ is strictly increasing. $\qquad\square$

The next two lemmas shows that the set $\mathcal{E} \cap \mathcal{N} \cap \mathcal{B}_c^{II}$ is closely related to the set of collapsed configurations $\Omega_c^{II}$. In order to states these lemmas, the following definition will prove convenient

$$\overline{\Omega}_c^{II} = \Big\{ (V, U) : \text{There exist a mean-zero equiangular matrix } \mathfrak{F} \text{ such that}$$

$$V = \sqrt{d}\, \mathfrak{F}\, P \quad \text{and} \quad \hat{U} = c\, \mathfrak{F}\, Z \Big\} \tag{118}$$

Note that $(W, U) \in \Omega_c^{II}$ if and only if $(\varphi(W), U) \in \overline{\Omega}_c^{II}$. Also, in light of (98), the inclusion

$$\overline{\Omega}_c^{II} \subset \mathcal{B}_c^{II}$$

is obvious. We now prove the following lemma.

**Lemma P.** *If the latent variables satisfy the symmetry assumption A, then*

$$\overline{\Omega}_c^{II} \subset \mathcal{E} \cap \mathcal{N} \cap \mathcal{B}_c^{II}$$

*Proof.* The proof is almost identical to the one of lemma I. We repeat it for completeness. We already know that $\overline{\Omega}_c^{II} \subset \mathcal{B}_c^{II}$. We the show that $\overline{\Omega}_c^{II} \subset \mathcal{E}$. Suppose $(V, U) \in \overline{\Omega}_c^{II}$. Then there exists a mean-zero equiangular matrix $\mathfrak{F} \in \mathbb{R}^{d \times n_c}$ such that

$$V = \sqrt{d}\, \mathfrak{F}\, P \qquad \text{and} \qquad \hat{U} = c\, \mathfrak{F}\, Z$$

Recall from (25) that $P\zeta(\mathbf{x}) = Z_k$ for all $\mathbf{x} \in \mathcal{X}_k$. Consider two latent variables

$$\mathbf{z}_k = [\alpha_1, \ldots, \alpha_L] \quad \text{and} \quad \mathbf{z}_j = [\alpha_1', \ldots, \alpha_L']$$

and assume $\mathbf{x}$ is generated by $\mathbf{z}_k$, meaning that $\mathbf{x} \in \mathcal{X}_k$. We then have

$$
\begin{aligned}
\mathfrak{M}_{V,U}(\mathbf{x}, j) &= \left\langle \hat{U}_k - \hat{U}_j, V\zeta(\mathbf{x}) \right\rangle_F \\
&= c\sqrt{d}\, \left\langle \mathfrak{F}\, Z_k - \mathfrak{F}\, Z_j, \mathfrak{F}\, P\zeta(\mathbf{x}) \right\rangle_F \\
&= c\sqrt{d}\, \left\langle \mathfrak{F}\, Z_k - \mathfrak{F}\, Z_j, \mathfrak{F}\, Z_k \right\rangle_F \\
&= c\sqrt{d}\, \sum_{\ell=1}^{L} \left\langle \mathfrak{f}_{\alpha_\ell} - \mathfrak{f}_{\alpha_\ell'}, \mathfrak{f}_{\alpha_\ell} \right\rangle_F \\
&= c\sqrt{d}\, \operatorname{dist}(\mathbf{z}_j, \mathbf{z}_k)
\end{aligned}
$$

From the above computation it is clear that the margin only depends on $\operatorname{dist}(\mathbf{z}_j, \mathbf{z}_k)$, and therefore $(V, U)$ satisfies the equimargin property.

Finally we show that $\overline{\Omega}_c^{II} \subset \mathcal{N}$. Suppose $(V, U) \in \overline{\Omega}_c^{II}$. Using the identity $\sum_{k=1}^{K} Z_k = \frac{K}{n_c} \mathbf{1}_{n_c} \mathbf{1}_L^T$ we obtain

$$\sum_{k=1}^{K} \hat{U}_k = c \sum_{k=1}^{K} \mathfrak{F}\, Z_k = c\, \frac{K}{n_c} \mathfrak{F}\, \mathbf{1}_{n_c} \mathbf{1}_L^T = 0$$

where we have used the fact that $\mathfrak{F}\, \mathbf{1}_{n_c} = 0$. $\qquad\square$

Finally, we have the following lemma.

**Lemma Q.** *If the latent variables satisfy assumptions A and B, then*

$$\overline{\Omega}_c^{II} = \mathcal{E} \cap \mathcal{N} \cap \mathcal{B}_c^{II}$$

*Proof.* The proof, again, is very similar to the one of lemma J. From the previous lemma we know that $\overline{\Omega}_c^{II} \subset \mathcal{E} \cap \mathcal{N} \cap \mathcal{B}_c^{II}$ so we need to show that

$$\mathcal{E} \cap \mathcal{N} \cap \mathcal{B}_c^{II} \subset \overline{\Omega}_c^{II}.$$

Let $(V, U) \in \mathcal{E} \cap \mathcal{N} \cap \mathcal{B}_c^{II}$. Since $(V, U)$ belongs to $\mathcal{B}_c^{II}$, there exists a matrix $F \in \mathbb{R}^{d \times n_c}$ whose columns have unit length and mean $0$ such that

$$V = \sqrt{d}\, F\, P \qquad \text{and} \qquad U = c\, F\, Z$$

Our goal is to show that $F$ is a mean-zero equiangular matrix, meaning that it satisfies the three relations

$$\mathbf{1}_{n_c}^T F = 0, \qquad F\, \mathbf{1}_{n_c} = 0 \qquad \text{and} \qquad F^T F = \frac{n_c}{n_c - 1}\, I_{n_c} - \frac{1}{n_c - 1}\, \mathbf{1}_{n_c} \mathbf{1}_{n_c}^T. \qquad (119)$$

We already know that the first relation is satisfied since the columns of $F$ have mean $0$. The second relation is easily obtained. Indeed, using the fact that $(V, U) \in \mathcal{N}$ together with the identity $\sum_{k=1}^{K} Z_k = \frac{K}{n_c} \mathbf{1}_{n_c} \mathbf{1}_L^T$ (which hold due to lemma B), we obtain

$$0 = \sum_{k=0}^{K} U_k = c' \sum_{k=0}^{K} F Z_k = c \frac{K}{n_c} F \mathbf{1}_{n_c} \mathbf{1}_L^T.$$

which implies $F \mathbf{1}_{n_c} = 0$.

We now prove the third equality of (119). Assume that $\mathbf{x} \in \mathcal{X}_k$. Using the fact that $P\zeta(\mathbf{x}) = Z_k$ together with (84), we obtain

$$
\begin{aligned}
\mathfrak{M}_{V,U}(\mathbf{x}, j) &= \left\langle \hat{U}_k - \hat{U}_j, V\zeta(\mathbf{x}) \right\rangle_F \\
&= c\sqrt{d} \left\langle F Z_k - F Z_j, F P\zeta(\mathbf{x}) \right\rangle_F \\
&= c\sqrt{d} \left\langle F Z_k - F Z_j, F Z_k \right\rangle_F \\
&= c\sqrt{d} \left\langle F^T F(Z_k - Z_j), Z_k \right\rangle_F \\
&= c\sqrt{d} \left\langle F^T F, \Gamma^{(k,j)} \right\rangle_F
\end{aligned}
\tag{120}
$$

Since $(V, U) \in \mathcal{E}$, the margins must only depend on the distance between the latent variables. Due to (120), we can be express this as

$$\left\langle F^T F, \Gamma^{(j,k)} \right\rangle_F = \left\langle F^T F, \Gamma^{(j',k')} \right\rangle_F \qquad \forall j, k, j', k' \in [K] \text{ s.t. } \mathrm{dist}(\mathbf{z}_j, \mathbf{z}_k) = \mathrm{dist}(\mathbf{z}_{j'}, \mathbf{z}_{k'})$$

Since the $F^T F$ is clearly positive semi-definite, we may then use assumption B to conclude that $F^T F \in \mathcal{A}$. Recalling definition (42) of the set $\mathcal{A}$, we therefore have

$$F^T F = a \, I_{n_c} + b \, \mathbf{1}_{n_c} \mathbf{1}_{n_c}^T \tag{121}$$

for some $a, b \in \mathbb{R}$. To conclude our proof, we need to show that

$$a = \frac{n_c}{n_c - 1} \qquad \text{and} \qquad b = -\frac{1}{n_c - 1}. \tag{122}$$

Combining (121) with the first equality of (119), we obtain

$$0 = F^T F \mathbf{1}_{n_c} = a \, \mathbf{1}_{n_c} + b \, \mathbf{1}_{n_c} \mathbf{1}_{n_c}^T \mathbf{1}_{n_c} = (a + b n_c) \mathbf{1}_{n_c}$$

Since the columns of $F$ have unit length, the diagonal entries of $F^T F$ must all be equal to 1, and therefore (121) implies that $a + b = 1$. The constants $a, b \in \mathbb{R}$, according must therefore solve the system

$$\begin{cases} a + b n_c &= 0 \\ a + b &= 1 \end{cases}$$

and one can easily check that the solution of this system is precisely given by (122). $\qquad \square$

We conlude this subsection by proving proposition B.

*Proof of Proposition B.* Let $(W^\star, U^\star)$ be a global minimizer of $\mathcal{R}$ and let $c \in \mathbb{R}$ be such that

$$\|U^\star\|_F^2 = KLc^2$$

We first prove statement (i) of the proposition. If the latent variables satisfies assumption A then lemma P asserts that

$$\overline{\Omega}_c^{II} \subset \mathcal{E} \cap \mathcal{N} \cap \mathcal{B}_c^{II}$$

Assume $(W, U) \in \Omega_c^{II}$. This implies that $(\varphi(W), U) \in \overline{\Omega}_c^{II}$, and and therefore $(\varphi(W), U) \in \mathcal{E} \cap \mathcal{N} \cap \mathcal{B}_c^{II}$. We can then use lemma O to conclude that $(W, U)$ is a global minimizer of $\mathcal{R}^*$.

We now prove statement (ii) of the proposition. If the latent variables satisfies assumption A and B then lemma Q asserts that

$$\overline{\Omega}_c^{II} = \mathcal{E} \cap \mathcal{N} \cap \mathcal{B}_c^{II}$$

The set $\overline{\Omega}_c^{II}$ is clearly not empty (because the set of mean-zero equiangular matrices is not empty), and we may therefore use the second statement of lemma O to obtain that

$$(\varphi(W^\star), U^\star) \in \mathcal{E} \cap \mathcal{N} \cap \mathcal{B}_c^{II} = \overline{\Omega}_c^{II}$$

which in turn implies $(W^\star, U^\star) \in \Omega_c^{II}$. $\qquad\square$

### F.3 DETERMINING THE CONSTANT $c$

The next lemma provides an explicit formula for the regularized risk of a network $h_{W,U}^*$ whose weights are in type-II collapse configuration with constant $c$.

**Lemma R.** *Assume the latent variables satisfy assumption A. If the pair of weights $(W, U)$ belongs to $\Omega_c^{II}$, then*

$$\mathcal{R}^*(W, U) = \log\left(1 - \frac{K}{n_c^L} + \frac{K}{n_c^L}\left(1 + (n_c - 1)e^{-\eta^* \sqrt{KL}\,c}\right)^L\right) + \frac{\lambda}{2}\left(\sqrt{KL}\,c\right)^2 \qquad (123)$$

*where $\eta^* = \frac{n_c}{n_c - 1}\sqrt{\frac{d}{KL}}$.*

Combining lemma R with proposition B concludes the proof of theorem F.

*Proof of lemma R.* We recall that

$$\mathcal{R}_0(W, U) = \frac{1}{K}\sum_{k=1}^K \sum_{\mathbf{x} \in \mathcal{X}_k} \log\left(1 + \sum_{j \neq k} e^{-\mathfrak{M}_{W,U}(\mathbf{x}, j)}\right) \mathcal{D}_{\mathbf{z}_k}(\mathbf{x})$$

and

$$\Omega_c^{II} = \Big\{(W, U) : \text{There exist a mean-zero equiangular matrix } \mathfrak{F} \text{ such that}$$

$$\varphi(W) = \sqrt{d}\,\mathfrak{F}\,P \quad \text{and} \quad \hat{U} = c\,\mathfrak{F}\,Z\Big\} \qquad (124)$$

Consider two latent variables

$$\mathbf{z}_k = [\alpha_1, \ldots, \alpha_L] \quad \text{and} \quad \mathbf{z}_j = [\alpha_1', \ldots, \alpha_L']$$

and assume $\mathbf{x} \in \mathcal{X}_k$. Using the identity $P\zeta(\mathbf{x}) = Z_k$ we then obtain

$$\begin{aligned}
\mathfrak{M}_{\varphi(W),U}(\mathbf{x}, j) &= \left\langle \hat{U}_k - \hat{U}_j, \varphi(W)\zeta(\mathbf{x})\right\rangle_F \\
&= c\sqrt{d}\,\langle \mathfrak{F}\,Z_k - \mathfrak{F}\,Z_j, \mathfrak{F}\,P\zeta(\mathbf{x})\rangle_F \\
&= c\sqrt{d}\,\langle \mathfrak{F}\,Z_k - \mathfrak{F}\,Z_j, \mathfrak{F}\,Z_k\rangle_F \\
&= c\sqrt{d}\sum_{\ell=1}^L \left\langle \mathfrak{f}_{\alpha_\ell} - \mathfrak{f}_{\alpha_\ell'}, \mathfrak{f}_{\alpha_\ell}\right\rangle_F \\
&= c\sqrt{d}\left(L - \sum_{\ell=1}^L \left\langle \mathfrak{f}_{\alpha_\ell'}, \mathfrak{f}_{\alpha_\ell}\right\rangle_F\right) \\
&= c\sqrt{d}\frac{n_c}{n_c - 1}\mathrm{dist}(\mathbf{z}_j, \mathbf{z}_k)
\end{aligned}$$

Letting $\omega^* = \sqrt{d}\frac{n_c}{n_c-1}$ we therefore obtain

$$\begin{aligned}
\mathcal{R}_0(W, U) &= \frac{1}{K}\sum_{k=1}^K \sum_{\mathbf{x} \in \mathcal{X}_k} \log\left(1 + \sum_{j \neq k} e^{-c\,\omega^*\,\mathrm{dist}(\mathbf{z}_j, \mathbf{z}_k)}\right) \mathcal{D}_{\mathbf{z}_k}(\mathbf{x}) \\
&= \frac{1}{K}\sum_{k=1}^K \log\left(1 + \sum_{j \neq k} e^{-c\,\omega^*\,\mathrm{dist}(\mathbf{z}_j, \mathbf{z}_k)}\right) \qquad (125)
\end{aligned}$$

where we have used the quantity inside the log does not depends on $\mathbf{x}$. Using the identity $|S_r| = \frac{K}{n_c^L}\binom{L}{r}(n_c - 1)^r$ we then obtain obtain

$$
\begin{aligned}
\sum_{j \neq k} e^{-c\,\omega\,\text{dist}(\mathbf{z}_j, \mathbf{z}_k)} &= \sum_{r=1}^{L} |\{j : \text{dist}(\mathbf{z}_j, \mathbf{z}_k) = r\}| \; e^{-c\,\omega^*\,r} \\
&= \frac{K}{n_c^L} \sum_{r=1}^{L} \binom{L}{r}(n_c - 1)^r \; e^{-c\,\omega^*\,r} \\
&= -\frac{K}{n_c^L} + \frac{K}{n_c^L} \sum_{r=0}^{L} \binom{L}{r}(n_c - 1)^r \; e^{-c\,\omega^*\,r} \\
&= -\frac{K}{n_c^L} + \frac{K}{n_c^L}\Big(1 + (n_c - 1)e^{-c\,\omega^*}\Big)^L
\end{aligned}
$$

where we have used the binomial theorem to obtain the last equality. The above quantity does not depends on $k$, therefore (125) can be expressed as

$$
\mathcal{R}_0(W, U) = \log\left(1 - \frac{K}{n_c^L} + \frac{K}{n_c^L}\Big(1 + (n_c - 1)e^{-c\,\omega^*}\Big)^L\right)
$$

We then remark that the matrix $\mathfrak{F} Z$ has $KL$ columns, and that each of these columns has norm 1. We therefore have

$$
\|\hat{U}\|_F^2 = \|c\,\mathfrak{F}\,Z\|_F^2 = c^2 KL \qquad \text{for all } (W, U) \in \Omega_c^{II}
$$

To conclude the proof we simply note that $\omega^* = \sqrt{KL}\,\eta^*$. $\qquad\qquad\square$

## G   Proof of theorem 2

This section is devoted to the proof of theorem 2 from the main paper, which we recall below for convenience.

**Theorem 2** (Directional Collapse of $h$). *Assume $K = n_c^L$ and $\{\mathbf{z}_1, \ldots, \mathbf{z}_K\} = \mathcal{Z}$. Assume also that the regularization parameter $\lambda$ satisfies*

$$
\lambda^2 < \frac{L}{n_c^{L+1}} \sum_{\beta=1}^{s_c} \mu_\beta^2 \tag{126}
$$

*Finally, assume that $(W, U)$ is in a type-III collapse configuration for some constants $c, r_1, \ldots, r_{s_c} \geq 0$. Then $(W, U)$ is a critical point of $\mathcal{R}$ if and only if $(c, r_1, \ldots, r_{s_c})$ solve the system*

$$
\frac{\lambda}{L}\,\frac{r_\beta}{c}\left(n_c - 1 + \exp\left(\frac{n_c}{n_c - 1}c\,r_\beta\right)\right) = \mu_\beta \qquad \text{for all } 1 \leq \beta \leq s_c \tag{127}
$$

$$
\sum_{\beta=1}^{s_c} \left(\frac{r_\beta}{c}\right)^2 = L n_c^{L-1}. \tag{128}
$$

At the end of this section, we also show that if (149) holds, then the system (150) – (151) has a unique solution (see proposition D in subsection G.2).

The strategy to prove theorem 2 is straightforward: we simply need to evaluate the gradient of the risk on weights $(W, U)$ which are in a type-III collapse configuration. Setting this gradient to zero will then lead to a system for the constants $c, r_1, \ldots, r_{s_c}$ defining the configuration. While conceptually simple, the gradient computation is quite lengthy.

We start by deriving formulas for the partial derivatives of $\mathcal{R}_0$ with respect to the linear weights $\mathbf{u}_{k,\ell}$ and the word embeddings $\mathbf{w}_{(\alpha,\beta)}$. As we will see, $\partial\mathcal{R}_0/\partial\mathbf{u}_{k,\ell}$ and $\partial\mathcal{R}_0/\partial\mathbf{w}_{(\alpha,\beta)}$ plays symmetric roles. In order to observe this symmetry, the following notation will prove convenient:

$$
\Phi_{(\alpha,\beta),(k,\ell)}(W, U) := \frac{1}{K}\sum_{j=1}^{K}\sum_{\mathbf{x}\in\mathcal{X}_j} \mathbf{1}_{\{x_\ell=(\alpha,\beta)\}}\Big(\mathbf{1}_{\{j=k\}} - q_{k,W,U}(\mathbf{x})\Big)\mathcal{D}_{\mathbf{z}_j}(\mathbf{x}) \tag{129}
$$

where

$$q_{k,W,U}(\mathbf{x}) := \frac{e^{\langle \hat{U}_k, W\zeta(\mathbf{x})\rangle_F}}{\sum_{k'=1}^{K} e^{\langle U_{k'}, W\zeta(\mathbf{x})\rangle_F}}$$

We may now state the first lemma of this section:

**Lemma S.** *The partial derivatives of $\mathcal{R}_0$ with respect to $\mathbf{u}_{k,\ell}$ and $\mathbf{w}_{(\alpha,\beta)}$ are given by*

$$-\frac{\partial \mathcal{R}_0}{\partial \mathbf{u}_{k,\ell}}(W,U) = \sum_{\alpha=1}^{n_c} \sum_{\beta=1}^{s_c} \Phi_{(\alpha,\beta),(k,\ell)}(W,U) \, \mathbf{w}_{(\alpha,\beta)}$$

$$-\frac{\partial \mathcal{R}_0}{\partial \mathbf{w}_{(\alpha,\beta)}}(W,U) = \sum_{k=1}^{K} \sum_{\ell=1}^{L} \Phi_{(\alpha,\beta),(k,\ell)}(W,U) \, \mathbf{u}_{k,\ell}$$

*Proof.* Given $K$ matrices $V_1, \dots, V_K \in \mathbb{R}^{n_w \times KL}$, we define

$$f(V_1, \dots, V_K) := \frac{1}{K} \sum_{k=1}^{K} \sum_{\mathbf{x} \in \mathcal{X}_k} \ell\Big(\langle V_1, \zeta(\mathbf{x})\rangle_F, \dots, \langle V_K, \zeta(\mathbf{x})\rangle_F \, ; k\Big) \, \mathcal{D}_{\mathbf{z}_k}(\mathbf{x})$$

where $\ell(y_1, \dots, y_K; k)$ is the cross entropy loss

$$\ell(y_1, \dots, y_K; k) = -\log\left(\frac{\exp(y_k)}{\sum_{k'=1}^{K} \exp(y_{k'})}\right)$$

The partial derivative of $f$ with respect to the matrix $V_j$ can easily be found to be

$$-\frac{\partial f}{\partial V_j}(V_1, \dots, V_K) = \frac{1}{K} \sum_{k=1}^{K} \sum_{\mathbf{x} \in \mathcal{X}_k} \left(\mathbf{1}_{\{j=k\}} - \frac{e^{\langle V_j, \zeta(\mathbf{x})\rangle_F}}{\sum_{k'=1}^{K} e^{\langle V_{k'}, \zeta(\mathbf{x})\rangle_F}}\right) \zeta(\mathbf{x}) \, \mathcal{D}_{\mathbf{z}_k}(\mathbf{x}) \quad (130)$$

We then recall from (19) that the $k^{th}$ entry of the vector $\mathbf{y} = h_{W,U}(\mathbf{x})$ is

$$y_k = \Big\langle \hat{U}_k \,, \, W\zeta(\mathbf{x}) \Big\rangle_F = \Big\langle W^T \hat{U}_k \,, \, \zeta(\mathbf{x}) \Big\rangle_F$$

and so the unregularized risk can be expressed in term of the function $f$:

$$\mathcal{R}_0(W,U) = \frac{1}{K} \sum_{k=1}^{K} \sum_{\mathbf{x} \in \mathcal{X}_k} \ell\Big(\langle W^T \hat{U}_1, \zeta(\mathbf{x})\rangle_F, \dots, \langle W^T \hat{U}_K, \zeta(\mathbf{x})\rangle_F \, ; k\Big) \, \mathcal{D}_{\mathbf{z}_k}(\mathbf{x})$$

$$= f(W^T \hat{U}_1, \dots, W^T \hat{U}_K)$$

The chain rule then gives

$$\frac{\partial \mathcal{R}_0}{\partial W}(W,U) = \sum_{j=1}^{K} \hat{U}_j \left[\frac{\partial f}{\partial V_j}(W^T \hat{U}_1, \dots, W^T \hat{U}_K)\right]^T \quad (131)$$

$$\frac{\partial \mathcal{R}_0}{\partial \hat{U}_j}(W,U) = W \left[\frac{\partial f}{\partial V_j}(W^T \hat{U}_1, \dots, W^T \hat{U}_K)\right] \quad (132)$$

Using formula (130) for $\partial f/\partial V_j$ and the notation

$$q_{j,W,U}(\mathbf{x}) := \frac{e^{\langle W^T \hat{U}_j, \zeta(\mathbf{x})\rangle_F}}{\sum_{k'=1}^{K} e^{\langle W^T U_{k'}, \zeta(\mathbf{x})\rangle_F}}$$

we can express (131) and (132) as follow

$$-\frac{\partial \mathcal{R}_0}{\partial W}(W,U) = \sum_{j=1}^{K} \hat{U}_j \left[\frac{1}{K} \sum_{k=1}^{K} \sum_{\mathbf{x} \in \mathcal{X}_k} \Big(\mathbf{1}_{\{j=k\}} - q_{j,W,U}(\mathbf{x})\Big) \zeta(\mathbf{x}) \, \mathcal{D}_{\mathbf{z}_k}(\mathbf{x})\right]^T$$

$$-\frac{\partial \mathcal{R}_0}{\partial \hat{U}_j}(W,U) = W \left[\frac{1}{K} \sum_{k=1}^{K} \sum_{\mathbf{x} \in \mathcal{X}_k} \Big(\mathbf{1}_{\{j=k\}} - q_{j,W,U}(\mathbf{x})\Big) \zeta(\mathbf{x}) \, \mathcal{D}_{\mathbf{z}_k}(\mathbf{x})\right]$$

We now compute the partial derivative of $\mathcal{R}_0$ with respect to $\mathbf{u}_{j,\ell}$. Let $\mathbf{e}_\ell \in \mathbb{R}^L$ be the $\ell^{th}$ basis vector. We then have

$$-\frac{\partial \mathcal{R}_0}{\partial \mathbf{u}_{j,\ell}}(W, U) = -\left[\frac{\partial \mathcal{R}_0}{\partial \hat{U}_j}(W, U)\right] \mathbf{e}_\ell$$

$$= \frac{1}{K} \sum_{k=1}^{K} \sum_{\mathbf{x} \in \mathcal{X}_k} \left(\mathbf{1}_{\{j=k\}} - q_{j,W,U}(\mathbf{x})\right) \ (W\zeta(\mathbf{x})\,\mathbf{e}_\ell) \ \mathcal{D}_{\mathbf{z}_k}(\mathbf{x})$$

Recall from (13) that $W\zeta(\mathbf{x})$ is the matrix that contains the $d$-dimensional embeddings of the words that constitute the sentence $\mathbf{x} \in \mathcal{X}$. So $W\zeta(\mathbf{x})\,\mathbf{e}_\ell$ is simply the embedding of the $\ell^{th}$ word of the sentence $\mathbf{x}$, and we can write it as

$$W\zeta(\mathbf{x})\,\mathbf{e}_\ell = \sum_{\alpha=1}^{n_c} \sum_{\beta=1}^{s_c} \mathbf{1}_{\{x_\ell=(\alpha,\beta)\}} \mathbf{w}_{(\alpha,\beta)}$$

We therefore have

$$-\frac{\partial \mathcal{R}_0}{\partial \mathbf{u}_{j,\ell}}(W, U) = \frac{1}{K} \sum_{k=1}^{K} \sum_{\mathbf{x} \in \mathcal{X}_k} \left(\mathbf{1}_{\{j=k\}} - q_{j,W,U}(\mathbf{x})\right) \left(\sum_{\alpha=1}^{n_c} \sum_{\beta=1}^{s_c} \mathbf{1}_{\{x_\ell=(\alpha,\beta)\}} \mathbf{w}_{(\alpha,\beta)}\right) \mathcal{D}_{\mathbf{z}_k}(\mathbf{x})$$

$$= \sum_{\alpha=1}^{n_c} \sum_{\beta=1}^{s_c} \left(\frac{1}{K} \sum_{k=1}^{K} \sum_{\mathbf{x} \in \mathcal{X}_k} \left(\mathbf{1}_{\{j=k\}} - q_{j,W,U}(\mathbf{x})\right) \mathbf{1}_{\{x_\ell=(\alpha,\beta)\}} \mathcal{D}_{\mathbf{z}_k}(\mathbf{x})\right) \mathbf{w}_{(\alpha,\beta)}$$

$$= \sum_{\alpha=1}^{n_c} \sum_{\beta=1}^{s_c} \Phi_{(\alpha,\beta),(j,\ell)}(W, U) \, \mathbf{w}_{(\alpha,\beta)}$$

which is the desired formula.

We now compute the gradient with respect $\mathbf{w}_{(\alpha,\beta)}$. Recalling that $\zeta(\alpha, \beta)$ is the one hot vector associate with word $(\alpha, \beta)$, we have

$$-\frac{\partial \mathcal{R}_0}{\partial \mathbf{w}_{(\alpha,\beta)}}(W, U) = -\left[\frac{\partial \mathcal{R}_0}{\partial W}(W, U)\right] \ \zeta(\alpha, \beta)$$

$$= \frac{1}{K} \sum_{j=1}^{K} \sum_{k=1}^{K} \sum_{\mathbf{x} \in \mathcal{X}_k} \left(\mathbf{1}_{\{j=k\}} - q_{j,W,U}(\mathbf{x})\right) \ \left(\hat{U}_j\, \zeta(\mathbf{x})^T \zeta(\alpha, \beta)\right) \mathcal{D}_{\mathbf{z}_k}(\mathbf{x})$$

Recall that the $\ell^{th}$ column of $\zeta(\mathbf{x})$ is the one-hot encoding of the $\ell^{th}$ word in the sentence $\mathbf{x}$. Therefore, the $\ell^{th}$ entry of the vector $\zeta(\mathbf{x})^T \zeta(\alpha, \beta) \in \mathbb{R}^L$ is given by the formula

$$\left[\zeta(\mathbf{x})^T \zeta(\alpha, \beta)\right]_\ell = \begin{cases} 1 & \text{if } x_\ell = (\alpha, \beta) \\ 0 & \text{otherwise} \end{cases}$$

As a consequence

$$\hat{U}_j\, \zeta(\mathbf{x})^T \zeta(\alpha, \beta) = \sum_{\ell=1}^{L} \mathbf{1}_{\{x_\ell=(\alpha,\beta)\}} \mathbf{u}_{j,\ell}$$

which leads to

$$-\frac{\partial \mathcal{R}_0}{\partial \mathbf{w}_{(\alpha,\beta)}}(W, U) = \frac{1}{K} \sum_{j=1}^{K} \sum_{k=1}^{K} \sum_{\mathbf{x} \in \mathcal{X}_k} \left(\mathbf{1}_{\{j=k\}} - q_{j,W,U}(\mathbf{x})\right) \ \left(\sum_{\ell=1}^{L} \mathbf{1}_{\{x_\ell=(\alpha,\beta)\}} \mathbf{u}_{j,\ell}\right) \mathcal{D}_{\mathbf{z}_k}(\mathbf{x})$$

$$= \sum_{\ell=1}^{L} \sum_{j=1}^{K} \left(\frac{1}{K} \sum_{k=1}^{K} \sum_{\mathbf{x} \in \mathcal{X}_k} \left(\mathbf{1}_{\{j=k\}} - q_{j,W,U}(\mathbf{x})\right) \ \mathbf{1}_{\{x_\ell=(\alpha,\beta)\}}\right) \mathcal{D}_{\mathbf{z}_k}(\mathbf{x})\, \mathbf{u}_{j,\ell}$$

$$= \sum_{\ell=1}^{L} \sum_{j=1}^{K} \Phi_{(\alpha,\beta),(j,\ell)}(W, U)\, \mathbf{u}_{j,\ell}$$

which is the desired formula. $\qquad\square$

## G.1 Gradient of the Risk for Weights in Type-III Collapse Configuration

In lemma S we computed the gradient of the risk for any possible weights $(W, U)$ and for any possible latent variables $\mathbf{z}_1, \ldots, \mathbf{z}_K$. In this section we will derive a formula for the gradient when the weights are in type-III collapse configuration and when the latent variables satisfy $\{\mathbf{z}_1, \ldots, \mathbf{z}_K\} = \mathcal{Z}$. We start by recalling the definition of a type-III collapse configuration.

**Definition G** (Type-III Collapse). *The weights $(W, U)$ of the network $h_{W,U}$ form a type-III collapse configuration if and only if*

  i) *There exists positive scalars $r_\beta \geq 0$ so that $\mathbf{w}_{(\alpha, \beta)} = r_\beta \, \mathfrak{f}_\alpha$ for all $(\alpha, \beta) \in \mathcal{V}$.*

  ii) *There exists $c \geq 0$ so that $\mathbf{u}_{k,\ell} = c \, \mathfrak{f}_\alpha$ for all $(k, \ell)$ satisfying $z_{k,\ell} = \alpha$ and all $\alpha \in \mathcal{C}$.*

*hold for some collection $\mathfrak{f}_1, \ldots, \mathfrak{f}_{n_c} \in \mathbb{R}^d$ of equiangular vectors.*

We also define the constant $\gamma \in \mathbb{R}$ and the sigmoid $\sigma : \mathbb{R} \to \mathbb{R}$ as follow:

$$\gamma := \frac{1}{n_c - 1} \qquad \text{and} \qquad \sigma(x) := \frac{1}{1 + \gamma e^{(1+\gamma)x}} \tag{133}$$

The goal of this subsection is to prove the following proposition.

**Proposition C.** *Suppose $K = n_c^L$ and $\{\mathbf{z}_1, \ldots, \mathbf{z}_K\} = \mathcal{Z}$. If the weights $(W, U)$ are in a type-III collapse configuration with constants $c, r_1, \ldots, r_{s_c} \geq 0$, then*

$$-\frac{\partial \mathcal{R}_0}{\partial \mathbf{u}_{k,\ell}}(W, U) = \frac{1}{c} \frac{1 + \gamma}{n_c^L} \left( \sum_{\beta=1}^{s_c} \mu_\beta \, \sigma(c \, r_\beta) \, r_\beta \right) \mathbf{u}_{k,\ell}$$

$$-\frac{\partial \mathcal{R}_0}{\partial \mathbf{w}_{(\alpha,\beta)}}(W, U) = c \, \frac{L(1 + \gamma)}{n_c} \, \frac{\mu_\beta \, \sigma(c \, r_\beta)}{r_\beta} \, \mathbf{w}_{(\alpha,\beta)}$$

Importantly, note that the above proposition states that $\partial \mathcal{R}_0 / \partial \mathbf{u}_{k,\ell}$ and $\mathbf{u}_{k,\ell}$ are aligned, and that $\partial \mathcal{R}_0 / \partial \mathbf{w}_{(\alpha,\beta)}$ and $\mathbf{w}_{(\alpha,\beta)}$ are aligned.

We start by introducing some notations which will make these gradient computations easier. The latent variables $\mathbf{z}_1, \ldots, \mathbf{z}_K$ will be written as

$$\mathbf{z}_k = [\, z_{k,1} \, , \, z_{k,2} \, , \, \ldots \, , \, z_{k,L} \,] \in \mathcal{Z}$$

where $1 \leq z_{k,\ell} \leq n_c$. We remark that any sentence $\mathbf{x}$ generated by the latent variable $\mathbf{z}_k$ must be of the form

$$\mathbf{x} = [(z_{k,1}, \beta_1), \ldots, (z_{k,L}, \beta_L)]$$

for some $(\beta_1, \ldots, \beta_L) \in [n_c]^L$, and that this sentence has a probability $\mu_{\beta_1} \mu_{\beta_2} \cdots \mu_{\beta_L}$ of being sampled. In light of this, we make the following definitions. For every $\boldsymbol{\beta} = (\beta_1, \ldots, \beta_L) \in [n_c^L]$ we let

$$\mathbf{x}_{k,\boldsymbol{\beta}} := [(z_{k,1}, \beta_1), \ldots, (z_{k,L}, \beta_L)] \in \mathcal{X} \tag{134}$$

$$\mu[\boldsymbol{\beta}] := \mu[\beta_1] \, \mu[\beta_2] \, \cdots \, \mu[\beta_L] \in [0, 1] \tag{135}$$

where we have used $\mu[\beta_\ell]$ instead of $\mu_{\beta_\ell}$ in order to avoid the double subscript. With these definitions at hand we have that

$$\mathcal{D}_{\mathbf{z}_j}(\mathbf{x}_{k,\boldsymbol{\beta}}) = \begin{cases} \mu[\boldsymbol{\beta}] & \text{if } k = j \\ 0 & \text{otherwise} \end{cases}$$

We are now ready to prove proposition C. We break the computation into four lemmas. The first one simply uses the notations that we just introduced in order to express $\Phi_{(\alpha,\beta),(k,\ell)}$ in a more convenient format.

**Lemma T.** *The quantity $\Phi_{(\alpha^\star, \beta^\star),(k,\ell)}(W, U)$ can be expressed as*

$$\Phi_{(\alpha^\star, \beta^\star),(k,\ell)}(W, U) = \frac{1}{K} \sum_{\boldsymbol{\beta} \in [n_c^L]} \mathbf{1}_{\{\beta_\ell = \beta^\star\}} \left( \mathbf{1}_{\{z_{k,\ell} = \alpha^\star\}} - \sum_{j=1}^K \mathbf{1}_{\{z_{j,\ell} = \alpha^\star\}} \, q_{k,W,U}(\mathbf{x}_{j,\boldsymbol{\beta}}) \right) \mu[\boldsymbol{\beta}].$$

*Proof.* Using the above notations, we rewrite $\Phi_{(\alpha,\beta),(k,\ell)}(W,U)$ as follow:

$$\Phi_{(\alpha^\star,\beta^\star),(k,\ell)}(W,U) = \frac{1}{K}\sum_{j=1}^{K}\sum_{\mathbf{x}\in\mathcal{X}_j}\mathbf{1}_{\{x_\ell=(\alpha^\star,\beta^\star)\}}\Big(\mathbf{1}_{\{j=k\}} - q_{k,W,U}(\mathbf{x})\Big)\mathcal{D}_{\mathbf{z}_j}(\mathbf{x})$$

$$= \frac{1}{K}\sum_{j=1}^{K}\sum_{\boldsymbol{\beta}\in[n_c^L]}\mathbf{1}_{\{(z_{j,\ell},\beta_\ell)=(\alpha^\star,\beta^\star)\}}\Big(\mathbf{1}_{\{j=k\}} - q_{k,W,U}(\mathbf{x}_{j,\boldsymbol{\beta}})\Big)\mathcal{D}_{\mathbf{z}_j}(\mathbf{x}_{j,\boldsymbol{\beta}})$$

$$= \frac{1}{K}\sum_{j=1}^{K}\sum_{\boldsymbol{\beta}\in[n_c^L]}\mathbf{1}_{\{z_{j,\ell}=\alpha^\star\}}\mathbf{1}_{\{\beta_\ell=\beta^\star\}}\Big(\mathbf{1}_{\{j=k\}} - q_{k,W,U}(\mathbf{x}_{j,\boldsymbol{\beta}})\Big)\mu[\boldsymbol{\beta}]$$

$$= \frac{1}{K}\sum_{\boldsymbol{\beta}\in[n_c^L]}\mathbf{1}_{\{\beta_\ell=\beta^\star\}}\left(\sum_{j=1}^{K}\mathbf{1}_{\{z_{j,\ell}=\alpha^\star\}}\Big(\mathbf{1}_{\{j=k\}} - q_{k,W,U}(\mathbf{x}_{j,\boldsymbol{\beta}})\Big)\right)\mu[\boldsymbol{\beta}]$$

To conclude the proof we simply remark that $\sum_j\mathbf{1}_{\{z_{j,\ell}=\alpha^\star\}}\mathbf{1}_{\{j=k\}} = \mathbf{1}_{\{z_{k,\ell}=\alpha^\star\}}$. $\square$

The following notation will be needed in our next lemma:

$$\delta(\alpha,\alpha') = \begin{cases} 1 & \text{if } \alpha=\alpha' \\ -\gamma & \text{if } \alpha\neq\alpha' \end{cases} \qquad \text{for all } \alpha,\alpha'\in[n_c] \tag{136}$$

where we recall that $\gamma = 1/(n_c-1)$. We think of $\delta(\alpha,\alpha')$ as a 'biased Kroecker delta' on the concepts. Importantly, note that if $\mathfrak{f}_1,\ldots,\mathfrak{f}_{n_c}$ are equiangular, then

$$\langle\mathfrak{f}_\alpha,\mathfrak{f}_{\alpha'}\rangle = \delta(\alpha,\alpha')$$

which is the motivation behind this definition. We may now state our second lemma.

**Lemma U.** *Assume $K = n_c^L$ and $\{\mathbf{z}_1,\ldots,\mathbf{z}_K\} = \mathcal{Z}$. Assume also that the weights $(W,U)$ are in a type-III collapse configuration with constants $c, r_1,\ldots,r_{s_c}\geq 0$. Then*

$$q_{k,W,U}(\mathbf{x}_{j,\boldsymbol{\beta}}) = \frac{\prod_{\ell=1}^{L}\exp\Big(c\,r_{\beta_\ell}\,\delta(z_{j,\ell},z_{k,\ell})\Big)}{\prod_{\ell=1}^{L}\psi(c\,r_{\beta_\ell})} \qquad \text{where} \quad \psi(x) = e^x + \frac{1}{\gamma}e^{-\gamma x}.$$

*for all $j,k\in[K]$ and all $\boldsymbol{\beta} = (\beta_1,\ldots,\beta_L)\in[n_c]^L$.*

*Proof.* Recalling that $\mathbf{x}_{j,\boldsymbol{\beta}} := [(z_{j,1},\beta_1),\ldots,(z_{j,L},\beta_L)]$, we obtain

$$\left\langle\hat{U}_k,W\zeta(\mathbf{x}_{j,\boldsymbol{\beta}})\right\rangle_F = \sum_{\ell=1}^{L}\langle\mathbf{u}_{k,\ell},\mathbf{w}_{(z_{j,\ell},\beta_\ell)}\rangle = \sum_{\ell=1}^{L}\langle\,c\,\mathfrak{f}_{z_{k,\ell}}\,,\,r_{\beta_\ell}\,\mathfrak{f}_{z_{j,\ell}}\,\rangle = c\sum_{\ell=1}^{L}r_{\beta_\ell}\delta(z_{k,\ell}\,,z_{j,\ell})$$

We then have

$$q_{k,W,U}(\mathbf{x}_{j,\boldsymbol{\beta}}) = \frac{e^{\langle\hat{U}_k,W\zeta(\mathbf{x}_{j,\boldsymbol{\beta}})\rangle_F}}{\sum_{k'=1}^{K}e^{\langle\hat{U}_{k'},W\zeta(\mathbf{x}_{j,\boldsymbol{\beta}})\rangle_F}} = \frac{\exp\Big(c\sum_{\ell=1}^{L}r_{\beta_\ell}\delta(z_{k,\ell}\,,z_{j,\ell})\Big)}{\sum_{k'=1}^{K}\exp\Big(c\sum_{\ell=1}^{L}r_{\beta_\ell}\delta(z_{k',\ell}\,,z_{j,\ell})\Big)}$$

$$= \frac{\prod_{\ell=1}^{L}\exp\Big(c\,r_{\beta_\ell}\,\delta(z_{k,\ell}\,,z_{j,\ell})\Big)}{\sum_{k'=1}^{K}\prod_{\ell=1}^{L}\exp\Big(c\,r_{\beta_\ell}\,\delta(z_{k',\ell}\,,z_{j,\ell})\Big)}$$

Since $\{\mathbf{z}_1,\ldots,\mathbf{z}_K\} = \mathcal{Z}$, the latent variables $\mathbf{z}_{k'} = [z_{k',1},\ldots,z_{k',L}]$ achieve all possible tuples $[\alpha'_1,\cdots,\alpha'_L]\in[n_c]^L$. The bottom term can therefore be expressed as

$$\sum_{k'=1}^{K}\prod_{\ell=1}^{L}\exp\Big(c\,r_{\beta_\ell}\,\delta(z_{k',\ell}\,,z_{j,\ell})\Big)$$

$$= \sum_{\alpha'_1=1}^{n_c}\sum_{\alpha'_2=1}^{n_c}\cdots\sum_{\alpha'_L=1}^{n_c}\exp\Big(c\,r_{\beta_1}\delta(\alpha'_1,z_{j,1})\Big)\exp\Big(c\,r_{\beta_2}\delta(\alpha'_2,z_{j,2})\Big)\cdots\exp\Big(c\,r_{\beta_L}\delta(\alpha'_L,z_{j,L})\Big)$$

$$= \prod_{\ell=1}^{L}\left(\sum_{\alpha'_\ell=1}^{n_c}\exp\Big(c\,r_{\beta_\ell}\delta(\alpha'_\ell,z_{k,\ell})\Big)\right)$$

Recalling the definition of $\delta(\alpha, \alpha')$, we find that

$$
\sum_{\alpha'_\ell=1}^{n_c} \exp\left(c\,r_{\beta_\ell} \delta(\alpha'_\ell, z_{k,\ell})\right) = \exp(c\,r_{\beta_\ell}) + \sum_{\alpha'_\ell \neq z_{k,\ell}} \exp\left(-\frac{c\,r_{\beta_\ell}}{n_c-1}\right)
$$

$$
= \exp(c\,r_{\beta_\ell}) + (n_c-1)\exp\left(-\frac{c\,r_{\beta_\ell}}{n_c-1}\right) = \psi(c\,r_{\beta_\ell}) \quad (137)
$$

$\square$

We now find a convenient expression for the term appearing between parenthesis in the statement of lemma T.

**Lemma V.** *Assume $K = n_c^L$ and $\{\mathbf{z}_1, \dots, \mathbf{z}_K\} = \mathcal{Z}$. Assume also that the weights $(W, U)$ are in a type-III collapse configuration with constants $c, r_1, \dots, r_{s_c} \geq 0$. Then*

$$
\mathbf{1}_{\{z_{k,\ell}=\alpha^\star\}} - \sum_{j=1}^{K} \mathbf{1}_{\{z_{j,\ell}=\alpha^\star\}}\, q_{k,W,U}(\mathbf{x}_{j,\boldsymbol{\beta}}) = \delta(z_{k,\ell}, \alpha^\star)\, \sigma(c\,r_{\beta_\ell}) \quad (138)
$$

*for all $k \in [K], \ell \in [L], \alpha^\star \in [n_c]$ and all $\boldsymbol{\beta} = (\beta_1, \dots, \beta_L) \in [n_c]^L$.*

*Proof.* For simplicity we are going to prove equation (138) in the case $\ell = 1$. Using the previous lemma we obtain

$$
\sum_{j=1}^{K} \mathbf{1}_{\{z_{j,1}=\alpha^\star\}} q_{k,W,U}(\mathbf{x}_{j,\boldsymbol{\beta}}) = \sum_{j=1}^{K} \mathbf{1}_{\{z_{j,1}=\alpha^\star\}} \frac{\prod_{\ell=1}^{L} \exp\left(c\,r_{\beta_\ell} \delta(z_{j,\ell}, z_{k,\ell})\right)}{\prod_{\ell=1}^{L} \psi(c\,r_{\beta_\ell})}
$$

Since the latent variables $\mathbf{z}_j = [z_{j,1}, \dots, z_{j,L}]$ achieve all possible tuples $[\alpha_1, \cdots, \alpha_L] \in [n_c]^L$, we can rewrite the above as

$$
\sum_{\alpha_1=1}^{n_c} \sum_{\alpha_2=1}^{n_c} \cdots \sum_{\alpha_L=1}^{n_c} \mathbf{1}_{\{\alpha_1=\alpha^\star\}} \frac{\prod_{\ell=1}^{L} \exp\left(c\,r_{\beta_\ell} \delta(\alpha_\ell, z_{k,\ell})\right)}{\prod_{\ell=1}^{L} \psi(c\,r_{\beta_\ell})}
$$

$$
= \sum_{\alpha_2=1}^{n_c} \cdots \sum_{\alpha_L=1}^{n_c} \frac{\exp\left(c\,r_{\beta_1} \delta(\alpha^\star, z_{k,1})\right) \prod_{\ell=2}^{L} \exp\left(c\,r_{\beta_\ell} \delta(\alpha_\ell, z_{k,\ell})\right)}{\prod_{\ell=1}^{L} \psi(c\,r_{\beta_\ell})}
$$

$$
= \frac{\exp\left(c\,r_{\beta_1} \delta(\alpha^\star, z_{k,1})\right)}{\prod_{\ell=1}^{L} \psi(c\,r_{\beta_\ell})} \sum_{\alpha_2=1}^{n_c} \cdots \sum_{\alpha_L=1}^{n_c} \prod_{\ell=2}^{L} \exp\left(c\,r_{\beta_\ell} \delta(\alpha_\ell, z_{k,\ell})\right) \quad (139)
$$

We then note that

$$
\sum_{\alpha_2=1}^{n_c} \cdots \sum_{\alpha_L=1}^{n_c} \prod_{\ell=2}^{L} \exp\left(c\,r_{\beta_\ell} \delta(\alpha_\ell, z_{k,\ell})\right) = \prod_{\ell=2}^{L} \left(\sum_{\alpha'_\ell=1}^{n_c} \exp\left(c\,r_{\beta_\ell} \delta(\alpha_\ell, z_{k,\ell})\right)\right)
$$

and, repeating computation (137), we find that

$$
\sum_{\alpha_\ell=1}^{n_c} \exp\left(c\,r_{\beta_\ell} \delta(\alpha_\ell, z_{k,\ell})\right) = \psi(c\,r_{\beta_\ell})
$$

Going back to (139) we therefore have

$$
\sum_{j=1}^{K} \mathbf{1}_{\{z_{j,1}=\alpha^\star\}} q_{k,W,U}(\mathbf{x}_{j,\boldsymbol{\beta}}) = \frac{\exp\left(c\,r_{\beta_1} \delta(\alpha^\star, z_{k,1})\right)}{\prod_{\ell=1}^{L} \psi(c\,r_{\beta_\ell})} \prod_{\ell=2}^{L} \psi(c\,r_{\beta_\ell}) = \frac{\exp\left(c\,r_{\beta_1} \delta(\alpha^\star, z_{k,1})\right)}{\psi(c\,r_{\beta_1})}
$$

and so

$$
\mathbf{1}_{\{z_{k,1}=\alpha^\star\}} - \sum_{j=1}^{K} \mathbf{1}_{\{z_{j,1}=\alpha^\star\}}\, q_{k,W,U}(\mathbf{x}_{j,\boldsymbol{\beta}}) = \begin{cases} 1 - \frac{\exp(c\,r_{\beta_1})}{\psi(c\,r_{\beta_1})} & \text{if } z_{k,1} = \alpha^\star \\ -\frac{\exp(-\gamma\,c\,r_{\beta_1})}{\psi(c\,r_{\beta_1})} & \text{if } z_{k,1} \neq \alpha^\star \end{cases} \quad (140)
$$

We now manipulate the above formula. Recalling that $\gamma = 1/(n_c - 1)$, and recalling the definition of $\psi(x)$, we get

$$1 - \frac{e^x}{\psi(x)} = 1 - \frac{e^x}{e^x + \frac{1}{\gamma}e^{-\gamma x}} = \frac{1}{1 + \gamma e^{(1+\gamma)x}} = \sigma(x) \tag{141}$$

and

$$-\frac{e^{-\gamma x}}{\psi(x)} = -\frac{e^{-\gamma x}}{e^x + \frac{1}{\gamma}e^{-\gamma x}} = -\gamma \left( \frac{1}{1 + \gamma e^{(1+\gamma)x}} \right) = -\gamma \sigma(x)$$

which concludes the proof. $\qquad \square$

Our last lemma provides a formula for the quantity $\Phi_{(\alpha^\star, \beta^\star),(k,\ell)}(W, U)$ when the weights are in a type-III collapse configuration.

**Lemma W.** *Assume $K = n_c^L$ and $\{\mathbf{z}_1, \ldots, \mathbf{z}_K\} = \mathcal{Z}$. Assume also that the weights $(W, U)$ are in a type-III collapse configuration with constants $c, r_1, \ldots, r_{s_c} \geq 0$. Then*

$$\Phi_{(\alpha,\beta),(k,\ell)}(W, U) = \frac{\mu_\beta}{n_c^L} \, \sigma(c\, r_\beta) \, \delta(z_{k,\ell}, \alpha) \tag{142}$$

*for all $k \in [K], \ell \in [L], \alpha \in [n_c]$ and $\beta \in [s_c]$.*

*Proof.* Combining lemmas T and V, and recalling that $K = n_c^L$, we obtain

$$\Phi_{(\alpha^\star, \beta^\star),(k,\ell)}(W, U) = \frac{1}{n_c^L} \sum_{\boldsymbol{\beta} \in [n_c^L]} \mathbf{1}_{\{\beta_\ell = \beta^\star\}} \left( \mathbf{1}_{\{z_{k,\ell} = \alpha^\star\}} - \sum_{j=1}^K \mathbf{1}_{\{z_{j,\ell} = \alpha^\star\}} \, q_{k,W,U}(\mathbf{x}_{j,\boldsymbol{\beta}}) \right) \mu[\boldsymbol{\beta}]$$

$$= \frac{1}{n_c^L} \sum_{\boldsymbol{\beta} \in [n_c^L]} \mathbf{1}_{\{\beta_\ell = \beta^\star\}} \Big( \delta(z_{k,\ell}, \alpha^\star) \, \sigma(c\, r_{\beta_\ell}) \Big) \mu[\boldsymbol{\beta}]$$

$$= \frac{\delta(z_{k,\ell}, \alpha^\star)}{n_c^L} \sum_{\boldsymbol{\beta} \in [n_c^L]} \mathbf{1}_{\{\beta_\ell = \beta^\star\}} \, \sigma(c\, r_{\beta_\ell}) \, \mu[\boldsymbol{\beta}]$$

Choosing $\ell = 1$ for simplicity we get

$$\sum_{\boldsymbol{\beta} \in [n_c^L]} \mathbf{1}_{\{\beta_1 = \beta^\star\}} \, \sigma(c\, r_{\beta_1}) \, \mu[\boldsymbol{\beta}] = \sum_{\beta_1=1}^{s_c} \sum_{\beta_2=1}^{s_c} \cdots \sum_{\beta_L=1}^{s_c} \mathbf{1}_{\{\beta_1 = \beta^\star\}} \, \sigma(c\, r_{\beta_1}) \, \mu[\beta_1]\mu[\beta_2]\cdots\mu[\beta_L]$$

$$= \sum_{\beta_2=1}^{s_c} \cdots \sum_{\beta_L=1}^{s_c} \sigma(c\, r_{\beta^\star}) \, \mu[\beta^\star]\mu[\beta_2]\cdots\mu[\beta_L]$$

$$= \mu[\beta^\star] \, \sigma(c\, r_{\beta^\star})$$

which concludes the proof. $\qquad \square$

We now prove the proposition.

*Proof of proposition C.* Combining lemmas S and W, and using the fact that $\mathbf{w}_{(\alpha,\beta)} = r_\beta \mathfrak{f}_\alpha$, we obtain

$$-\frac{\partial \mathcal{R}_0}{\partial \mathbf{u}_{k,\ell}}(W, U) = \sum_{\alpha=1}^{n_c} \sum_{\beta=1}^{s_c} \Phi_{(\alpha,\beta),(k,\ell)}(W, U) \, \mathbf{w}_{(\alpha,\beta)}$$

$$= \sum_{\alpha=1}^{n_c} \sum_{\beta=1}^{s_c} \left( \frac{\mu_\beta}{n_c^L} \, \sigma(c\, r_\beta) \, \delta(z_{k,\ell}, \alpha) \right) r_\beta \mathfrak{f}_\alpha$$

$$= \frac{1}{n_c^L} \left( \sum_{\beta=1}^{s_c} \mu_\beta \, \sigma(c\, r_\beta) \, r_\beta \right) \left( \sum_{\alpha=1}^{n_c} \delta(z_{k,\ell}, \alpha) \, \mathfrak{f}_\alpha \right)$$

Using the fact that $\sum_{\alpha=1}^{n_c} \mathfrak{f}_\alpha = 0$ we get

$$\sum_{\alpha=1}^{n_c} \delta(z_{k,\ell}, \alpha) \, \mathfrak{f}_\alpha = \mathfrak{f}_{z_{k,\ell}} - \gamma \sum_{\alpha \neq z_{k,\ell}} \mathfrak{f}_\alpha = \mathfrak{f}_{z_{k,\ell}} + \gamma \, \mathfrak{f}_{z_{k,\ell}} - \gamma \sum_{\alpha=1}^{n_c} \mathfrak{f}_\alpha = (1+\gamma) \, \mathfrak{f}_{z_{k,\ell}} \qquad (143)$$

Using the fact that $\mathbf{u}_{k,\ell} = c \, \mathfrak{f}_{z_{k,\ell}}$ we then get

$$-\frac{\partial \mathcal{R}_0}{\partial \mathbf{u}_{k,\ell}}(W, U) = \frac{1}{n_c^L} \left( \sum_{\beta=1}^{s_c} \mu_\beta \, \sigma(c\, r_\beta) \, r_\beta \right) (1+\gamma) \, \mathfrak{f}_{z_{k,\ell}}$$

$$= \frac{1+\gamma}{n_c^L} \left( \sum_{\beta=1}^{s_c} \mu_\beta \, \sigma(c\, r_\beta) \, r_\beta \right) \frac{\mathbf{u}_{k,\ell}}{c}$$

which is the desired formula.

Moving to the other gradient we get

$$-\frac{\partial \mathcal{R}_0}{\partial \mathbf{w}_{(\alpha,\beta)}}(W, U) = \sum_{k=1}^{K} \sum_{\ell=1}^{L} \Phi_{(\alpha,\beta),(k,\ell)}(W, U) \, \mathbf{u}_{k,\ell}$$

$$= \sum_{k=1}^{K} \sum_{\ell=1}^{L} \left( \frac{\mu_\beta}{n_c^L} \, \sigma(c\, r_\beta) \, \delta(z_{k,\ell}, \alpha) \right) c \, \mathfrak{f}_{z_{k,\ell}}$$

$$= \frac{\mu_\beta}{n_c^L} \sigma(c\, r_\beta) \, c \sum_{\ell=1}^{L} \left( \sum_{k=1}^{K} \delta(z_{k,\ell}, \alpha) \, \mathfrak{f}_{z_{k,\ell}} \right)$$

Since the latent variables $\mathbf{z}_k = [z_{k,1}, \ldots, z_{k,L}]$ achieve all possible tuples $[\alpha_1', \cdots, \alpha_L'] \in [n_c]^L$, we have, fixing $\ell = 1$ for simplicity,

$$\sum_{k=1}^{K} \delta(z_{k,1}, \alpha) \, \mathfrak{f}_{z_{k,1}} = \sum_{\alpha_1'=1}^{n_c} \sum_{\alpha_2'=1}^{n_c} \cdots \sum_{\alpha_L'=1}^{L} \delta(\alpha_1', \alpha) \, \mathfrak{f}_{\alpha_1'} = n_c^{L-1} \sum_{\alpha_1'=1}^{L} \delta(\alpha_1', \alpha) \, \mathfrak{f}_{\alpha_1'} \qquad (144)$$

Repeating computation (143) shows that the above is equal to $n_c^{L-1}(1+\gamma) \, \mathfrak{f}_\alpha$. We then use the fact that $\mathbf{w}_{(\alpha,\beta)} = r_\beta \mathfrak{f}_\alpha$ to obtain

$$-\frac{\partial \mathcal{R}_0}{\partial \mathbf{w}_{(\alpha,\beta)}}(W, U) = \frac{\mu_\beta}{n_c^L} \sigma(c\, r_\beta) \, cL\left( n_c^{L-1}(1+\gamma) \, \mathfrak{f}_\alpha \right)$$

$$= \frac{\mu_\beta}{n_c^L} \sigma(c\, r_\beta) \, cL\left( n_c^{L-1}(1+\gamma) \, \frac{\mathbf{w}_{(\alpha,\beta)}}{r_\beta} \right)$$

$$= \frac{\mu_\beta}{n_c} \sigma(c\, r_\beta) \, cL\left( (1+\gamma) \, \frac{\mathbf{w}_{(\alpha,\beta)}}{r_\beta} \right)$$

which is the desired formula. $\square$

### G.2 PROOF OF THE THEOREM AND STUDY OF THE NON-LINEAR SYSTEM

In this subsection we start by proving theorem 2, and then we show that the system (150) – (151) has a unique solution if the regularization parameter $\lambda$ is small enough.

*Proof of theorem 2.* Recall that the regularized risk associated with the network $h_{W,U}$ is defined by

$$\mathcal{R}(W, U) = \mathcal{R}_0(W, U) + \frac{\lambda}{2} \left( \|W\|_F^2 + \|U\|_F^2 \right) \qquad (145)$$

$$= \mathcal{R}_0(W, U) + \frac{\lambda}{2} \left( \sum_{\alpha=1}^{n_c} \sum_{\beta=1}^{s_c} \|\mathbf{w}_{(\alpha,\beta)}\|^2 + \sum_{k=1}^{K} \sum_{\ell=1}^{L} \|\mathbf{u}_{k,\ell}\|^2 \right) \qquad (146)$$

and therefore $(W, U)$ is a critical points if and only if

$$-\frac{\partial \mathcal{R}_0}{\partial \mathbf{u}_{k,\ell}}(W, U) = \lambda \, \mathbf{u}_{k,\ell} \qquad \text{and} \qquad -\frac{\partial \mathcal{R}_0}{\partial \mathbf{w}_{(\alpha,\beta)}}(W, U) = \lambda \, \mathbf{w}_{(\alpha,\beta)}$$

According to proposition C, if (W,U) is in a type-III collapse configuration, then the above equations becomes

$$\frac{1}{c}\frac{1+\gamma}{n_c^L}\left(\sum_{\beta=1}^{s_c} \mu_\beta \, \sigma(c\,r_\beta)\, r_\beta\right)\mathbf{u}_{k,\ell} = \lambda \, \mathbf{u}_{k,\ell} \quad \text{and} \quad c\,\frac{L(1+\gamma)}{n_c}\,\frac{\mu_\beta \, \sigma(c\,r_\beta)}{r_\beta}\mathbf{w}_{(\alpha,\beta)} = \lambda \, \mathbf{w}_{(\alpha,\beta)}$$

So $(W, U)$ is critical if and only if the constants $r_1, \ldots, r_{s_c}$ and $c$ satisfy the $s_c + 1$ equations

$$\frac{1}{c}\frac{1+\gamma}{n_c^L}\sum_{\beta=1}^{s_c} \mu_\beta \, \sigma(c\,r_\beta) r_\beta = \lambda \tag{147}$$

$$c\,\frac{L(1+\gamma)}{n_c}\frac{\mu_\beta \, \sigma(c\,r_\beta)}{r_\beta} = \lambda \qquad \text{for all } \beta \in [s_c] \tag{148}$$

From the second equation we have that

$$(1+\gamma)\,\mu_\beta \, \sigma(c\,r_\beta)\, r_\beta = \frac{n_c\,\lambda\,r_\beta^2}{L\,c}$$

Using this we can rewrite the first equation as

$$\frac{1}{c}\frac{1}{n_c^L}\sum_{\beta=1}^{s_c}\frac{n_c\,\lambda\,r_\beta^2}{L\,c} = \lambda \qquad \text{which simplifies to} \qquad \sum_{\beta=1}^{s_c}\left(\frac{r_\beta}{c}\right)^2 = Ln_c^{L-1}.$$

which is the desired equation (see (151)).

We now rewrite the second equation as

$$\frac{\lambda}{L}\frac{r_\beta}{c}\frac{n_c}{(1+\gamma)\,\sigma(c\,r_\beta)} = \mu_\beta$$

We then recall that $\sigma(x) := \frac{1}{1+\gamma e^{(1+\gamma)x}}$ and therefore

$$\frac{n_c}{(1+\gamma)\,\sigma(c r_\beta)} = \frac{n_c}{1+\gamma}(1+\gamma e^{(1+\gamma)c r_\beta}) = n_c - 1 + \exp\left(\frac{n_c}{n_c-1}c\,r_\beta\right)$$

and therefore the second equation can be written as

$$\frac{\lambda}{L}\frac{r_\beta}{c}\left(n_c - 1 + \exp\left(\frac{n_c}{n_c-1}c\,r_\beta\right)\right) = \mu_\beta.$$

$\square$

We now prove that if the regularization parameter $\lambda$ is small enough then the system has a unique solution.

**Proposition D.** *Assume $\mu_1 \geq \mu_2 \geq \ldots \geq \mu_{s_c} > 0$ and*

$$\lambda^2 < \frac{L}{n_c^{L+1}}\sum_{\beta=1}^{s_c}\mu_\beta^2. \tag{149}$$

*Then the system $s_c + 1$ equations*

$$\frac{\lambda}{L}\frac{r_\beta}{c}\left(n_c - 1 + \exp\left(\frac{n_c}{n_c-1}c\,r_\beta\right)\right) = \mu_\beta \qquad \text{for all } 1 \leq \beta \leq s_c \tag{150}$$

$$\sum_{\beta=1}^{s_c}\left(\frac{r_\beta}{c}\right)^2 = Ln_c^{L-1} \tag{151}$$

*has a unique solution $(c, r_1, \ldots, r_{s_c}) \in \mathbb{R}_+^{s_c+1}$. Moreover this solution satisfies $r_1 \geq r_2 \geq \ldots \geq r_{s_c} > 0$.*

*Proof.* Letting $\rho_\beta := r_\beta/c$, the system is equivalent to

$$g(c, \rho_\beta) = \frac{L}{\lambda n_c} \mu_\beta \qquad \text{for all } \beta \in [s_c] \tag{152}$$

$$\sum_{\beta=1}^{s_c} \rho_\beta^2 = Ln_c^{L-1} \tag{153}$$

for the unknowns $(c, \rho_1, \rho_2, \ldots, \rho_{s_c})$ where

$$g(c, x) = x \left(1 + \gamma e^{(1+\gamma)c^2 x}\right) / (1 + \gamma) \qquad \text{and} \qquad \gamma = 1/(n_c - 1)$$

Note that

$$\frac{\partial g}{\partial x}(c, x) \geq \left(1 + \gamma e^{(1+\gamma)c^2 x}\right) / (1 + \gamma) \geq 1 \qquad \forall (c, x) \in \mathbb{R} \times [0, +\infty)$$

and therefore $x \mapsto g(c, x)$ is strictly increasing on $[0, +\infty)$. Also note that we have

$$g(c, 0) = 0, \qquad \lim_{x \to +\infty} g(c, x) = +\infty$$

So $x \mapsto g(c, x)$ is a bijection from $[0, +\infty)$ to $[0, +\infty)$ as well as a bijection from $(0, +\infty)$ to $(0, +\infty)$. Recall that $\mu_\beta \in (0, +\infty)$ for all $\beta \in [s_c]$. Therefore given $c \in \mathbb{R}$ and $\beta \in [s_c]$, the equation

$$g(c, x) = \frac{L}{\lambda n_c} \mu_\beta$$

has a unique solution in $(0, +\infty)$ that we denote by $\phi_\beta(c)$. In other words, the function $\phi_\beta(c)$ is implicitly defined by

$$g(c, \phi_\beta(c)) = \frac{L}{\lambda n_c} \mu_\beta. \tag{154}$$

Also, since $g(0, x) = x$, we have

$$\phi_\beta(0) = \frac{L}{\lambda n_c} \mu_\beta$$

**Claim F.** *The function $\phi_\beta : [0, +\infty) \to (0, +\infty)$ is continuous, strictly decreasing, and satisfies* $\lim_{c \to +\infty} \phi_\beta(c) = 0$.

*Proof.* We first show that $c \mapsto \phi_\beta(c)$ is continuous. Since $\frac{\partial g}{\partial x}(c, x) \geq 1$ for all $x \geq 0$, we have

$$g(c, x_2) - g(c, x_1) = \int_{x_1}^{x_2} \frac{\partial g}{\partial x}(c, x)dx \geq \int_{x_1}^{x_2} 1dx = x_2 - x_1 \qquad \text{for all } c \text{ and all } x_2 \geq x_1 \geq 0.$$

As a consequence, for all $c_1, c_2$, we have

$$|\phi_\beta(c_2) - \phi_\beta(c_1)| \leq |g(c_1, \phi_\beta(c_2)) - g(c_1, \phi_\beta(c_1))| = |g(c_1, \phi_\beta(c_2)) - g(c_2, \phi_\beta(c_2))| \tag{155}$$

where we have used the fact that $g(c_1, \phi_\beta(c_1)) = \frac{L}{\lambda n_c} \mu_\beta = g(c_2, \phi_\beta(c_2))$. From (155) it is clear that the continuity of $c \mapsto g(c, x)$ implies the continuity of $c \mapsto \phi_\beta(c)$.

We now prove that $\phi_\beta$ is strictly decreasing on $[0, +\infty)$. Let $0 \leq c_1 < c_2$. Note that for any $x > 0$, the function $c \mapsto g(c, x)$ is strictly increasing on $[0, +\infty)$. Since $\phi_\beta(c) > 0$ we therefore have

$$g(c_2, \phi_\beta(c_2)) = \frac{L}{\lambda n_c} \mu_\beta = g(c_1, \phi_\beta(c_1)) < g(c_2, \phi_\beta(c_1))$$

Since $x \mapsto g(c, x)$ is strictly increasing for all $c$, the above implies that $\phi_\beta(c_2) < \phi_\beta(c_1)$.

Finally we show that $\lim_{c \to +\infty} \phi_\beta(c) = 0$. Since $\phi_\beta$ is decreasing and non-negative on $[0, +\infty)$, the $\lim_{c \to +\infty} \phi_\beta(c) = A$ is well defined. We obviously have $\phi_\beta(c) \geq A$ for all $c \geq 0$. Since $x \mapsto g(c, x)$ is increasing we have

$$\frac{L}{\lambda n_c} \mu_\beta = g(c, \phi_\beta(c)) \geq g(c, A)$$

But the function $g(c, A)$ is unbounded for all $A > 0$. Therefore we must have $A = 0$. □

System (152)–(153) is equivalent to

$$\rho_\beta = \phi_\beta(c) \qquad \text{for all } \beta \in [s_c] \tag{156}$$

$$\sum_{\beta=1}^{s_c} (\phi_\beta(c))^2 = L n_c^{L-1} \tag{157}$$

Define the function

$$\Phi(c) := \sum_{\beta=1}^{s_c} (\phi_\beta(c))^2$$

Then $\Phi$ clearly inherits the properties of the $\phi_\beta$'s: it is continuous, strictly decreasing, and satisfies

$$\Phi(0) = \sum_{\beta=1}^{s_c} \left( \frac{L}{\lambda n_c} \mu_\beta \right)^2 \qquad \text{and} \qquad \lim_{c \to +\infty} \Phi(c) = 0$$

Therefore, if

$$L n_c^{L-1} \leq \sum_{\beta=1}^{L} \left( \frac{L}{\lambda n_c} \mu_\beta \right)^2$$

then there is a unique $c \geq 0$ satisfying (157). Since $x \mapsto g(c, x)$ is increasing, equation (152) implies that the corresponding $\rho_\beta$'s satisfy $\rho_1 \geq \rho_2 \geq \ldots \geq \rho_{s_c} > 0$.

$\square$

# H  NO SPURIOUS LOCAL MINIMIZER FOR $\mathcal{R}(W, U)$.

In this section we prove that if $d > \min(n_w, KL)$, then $\mathcal{R}(W, U)$ does not have spurious local minimizers; all local minimizers are global. To do this, we introduce the function

$$f : \mathbb{R}^{d \times KL} \to \mathbb{R}$$

define as follow. Any matrix $V \in \mathbb{R}^{d \times KL}$ can be partition into $K$ submatrices $V_k \in \mathbb{R}^{d \times L}$ according

$$V = [V_1 \quad V_2 \quad \cdots \quad V_K] \qquad \text{where } V_k \in \mathbb{R}^{d \times L} \tag{158}$$

The function $f$ is then defined by the formula

$$f(V) := \frac{1}{K} \sum_{k=1}^{K} \sum_{\mathbf{x} \in \mathcal{X}_k} \ell\left( \left\langle V_1, \zeta(\mathbf{x}) \right\rangle_F, \ldots, \left\langle V_K, \zeta(\mathbf{x}) \right\rangle_F ; k \right) \; \mathcal{D}_{\mathbf{z}_k}(\mathbf{x})$$

where $\ell(y_1, \ldots, y_K; k)$ denotes the cross entropy loss

$$\ell(y_1, \ldots, y_K; k) = -\log\left( \frac{\exp(y_k)}{\sum_{k'=1}^{K} \exp(y_{k'})} \right)$$

We remark that $f$ is clearly convex and differentiable. We then recall from (19) that the $k^{th}$ entry of the vector $\mathbf{y} = h_{W,U}(\mathbf{x})$ is

$$y_k = \left\langle \hat{U}_k \,, \, W \zeta(\mathbf{x}) \right\rangle_F = \left\langle W^T \hat{U}_k \,, \, \zeta(\mathbf{x}) \right\rangle_F$$

Recalling that $\hat{U} = [\hat{U}_1 \quad \cdots \quad \hat{U}_K]$, we then see that the risk can be expressed as

$$\mathcal{R}(W, U) = f(W^T \hat{U}) + \frac{\lambda}{2} \left( \|W\|_F^2 + \|\hat{U}\|_F^2 \right) \tag{159}$$

The fact that $\mathcal{R}(W, U)$ does not have spurious local minimizers come from the following general theorem.

**Theorem G.** *Let $g : \mathbb{R}^{m \times n} \to \mathbb{R}$ be a convex and differentiable function. Define*

$$\varphi(A, B) := g(A^T B) + \frac{\lambda}{2} \left( \|A\|_F^2 + \|B\|_F^2 \right) \qquad where \quad A \in \mathbb{R}^{d \times m} \text{ and } B \in \mathbb{R}^{d \times n}$$

*and assume $\lambda > 0$ and $d > \min(m, n)$. Then any local minimizer $(A, B)$ of the function $\varphi : \mathbb{R}^{d \times m} \times \mathbb{R}^{d \times n} \to \mathbb{R}$ is also a global minimizer.*

The above theorem directly apply to (159) and shows that the risk $\mathcal{R}(W, U)$ does not have spurious local minimizers when $\lambda > 0$ and $d > \min(n_w, KL)$.

The remainder of the section is devoted to the proof of theorem G. We will follow the exact same steps as in Zhu et al. (2021), and provide the proof mostly for completeness (and also to show how the techniques from Zhu et al. (2021) apply to our case). Finally, we refer to Laurent & Brecht (2018) for a proof of theorem G in the case $\lambda = 0$.

*Proof of theorem G.* To prove the theorem it suffices to assume that $d > m$ without loss of generality. To see this, note that the function $\tilde{g}(D) = g(D^T)$ is also convex and differentiable and note that $(A, B)$ is a local minimum of

$$g(A^T B) + \frac{\lambda}{2} \left( \|A\|_F^2 + \|B\|_F^2 \right)$$

if and only if it is a local minimum of

$$\tilde{g}(B^T A) + \frac{\lambda}{2} \left( \|A\|_F^2 + \|B\|_F^2 \right)$$

So the theorem for the case $d > n$ follows by appealing to the case $d > m$ with the function $\tilde{g}$.

So we may assume $d > m$. Following Zhu et al. (2021), we define the function $\psi : \mathbb{R}^{m \times n} \to \mathbb{R}$ by

$$\psi(D) := g(D) + \|D\|_*$$

where $\|D\|_*$ denote the nuclear norm of $D$. We then have:

**Claim G.** *For all $A \in \mathbb{R}^{d \times m}$ and $B \in \mathbb{R}^{d \times n}$, we have that $\psi(A^T B) \le \varphi(A, B)$.*

*Proof.* This is a direct consequence of the inequality

$$\|A^T B\|_* \le \frac{1}{2} \left( \|A\|_F^2 + \|B\|_F^2 \right)$$

that we reprove here for completeness. Let $A^T B = U \Sigma V^T$ be the compact SVD of $A^T B$. That is $\Sigma \in \mathbb{R}^{r \times r}$, $U \in \mathbb{R}^{m \times r}$, $V \in \mathbb{R}^{n \times r}$, and $r$ is the rank of $A^T B$. We then have

$$\|A^T B\|_* = \text{Tr}(\Sigma) = \text{Tr}(U^T A^T B V) = \langle AU, BV \rangle_F \le \frac{1}{2} \left( \|AU\|_F^2 + \|BV\|_F^2 \right) \le \frac{1}{2} \left( \|A\|_F^2 + \|B\|_F^2 \right)$$

$\square$

Computing the derivatives of $\varphi$ gives

$$\frac{\partial \varphi}{\partial A}(A, B) = B \left[ \nabla g(A^T B) \right]^T + \lambda A \qquad \text{and} \qquad \frac{\partial \varphi}{\partial B}(A, B) = A \nabla g(A^T B) + \lambda B \qquad (160)$$

So $(A, B)$ is a critical point of $\varphi$ if and only if

$$\lambda A = -B \left[ \nabla g(A^T B) \right]^T \qquad (161)$$

$$\lambda B = -A \nabla g(A^T B) \qquad (162)$$

Importantly, from the above we get

$$AA^T = BB^T \in \mathbb{R}^{d \times d} \qquad (163)$$

which implies that $A$ and $B$ have same singular values and same left singular vectors. Let $U \in \mathbb{R}^{d \times d}$ be the orthonormal matrix containing the eigenvectors of $AA^T = BB^T$. From this matrix we can construct an SVD for both $A$ and $B$:

$$A = U\Sigma_A V_A^T \qquad \text{and} \qquad B = U\Sigma_B V_B^T$$

where $\Sigma_A \in \mathbb{R}^{d \times m}$ and $\Sigma_B \in \mathbb{R}^{d \times n}$ have the same singular values. From this we get the SVD of $A^T B$,

$$A^T B = V_A \Sigma_A^T \Sigma_B V_B^T \tag{164}$$

and it is transparent that,

$$\|A^T B\|_* = \|A\|_F^2 = \|B\|_F^2. \tag{165}$$

In particular this implies that if $(A, B)$ is a critical point of $\varphi$, then we must have $\varphi(A, B) = \psi(A^T B)$. This also implies that

$$\left\langle \nabla g(A^T B), A^T B \right\rangle_F = \left\langle A \nabla g(A^T B), B \right\rangle_F = -\lambda \|B\|_F^2 = -\lambda \|A^T B\|_* \tag{166}$$

Using this together with the fact that the nuclear norm is the dual of the operator norm, that is $\|C\|_* = \sup_{\|G\|_{op} \leq 1} \langle G, C \rangle_F$, we easily obtain:

**Claim H.** *Suppose $(A, B)$ is a critical point of $\varphi$ which satisfies $\left\| \nabla g(A^T B) \right\|_{op} \leq \lambda$, then $(A, B)$ is a global minimizer of $\varphi$.*

*Proof.* For any matrix $C \in \mathbb{R}^{m \times n}$ we have

$$\|A^T B\|_* + \left\langle -\frac{1}{\lambda} \nabla g(A^T B), C - A^T B \right\rangle_F = \left\langle -\frac{1}{\lambda} \nabla g(A^T B), C \right\rangle_F \leq \sup_{\|G\|_{op} \leq 1} \langle G, C \rangle_F = \|C\|_*$$

and therefore $-\frac{1}{\lambda} \nabla g(A^T B) \in \partial \|A^T B\|_*$. This implies that $A^T B$ is a global min of $\psi$. The fact that $\varphi(A, B) = \psi(A^T B)$ (because $(A, B)$ is a critical point $\varphi$) together with Claim G, then implies that $(A, B)$ is a global minimizer of $\varphi$. □

We now show that all local min $(A, B)$ of $\varphi$ with $\ker(A^T) \neq \emptyset$ must satisfy $\left\| \nabla g(A^T B) \right\|_{op} \leq \lambda$.

**Claim I.** *Suppose $(A, B)$ is a critical point of $\varphi$ which satisfies*

(i) $\ker(A^T) \neq \emptyset$

(ii) $\left\| \nabla g(A^T B) \right\|_{op} > \lambda$

*Then $(A, B)$ is not local min.*

*Proof.* We follow the computation from Zhu et al. (2021). Let $(A, B)$ be a critical point of $\varphi$. Since $AA^T = BB^T$, we must have that $\ker(A^T) = \ker(AA^T) = \ker(BB^T) = \ker(B^T)$. According to (ii) these kernels are non trivial and we may choose a unit vector $\mathbf{z} \in \mathbb{R}^d$ that belongs to them. We then consider the perturbations

$$dA = \mathbf{z}\mathbf{a}^T \quad dB = \mathbf{z}\mathbf{b}^T$$

where $\mathbf{a} \in \mathbb{R}^m$ and $\mathbf{b} \in \mathbb{R}^n$ are unit vectors to be chosen later. Note that since $\mathbf{z}, \mathbf{a}$ and $\mathbf{b}$ are unit vectors we have $\|dA\|_F^2 = \|dB\|_F^2 = 1$. Moreover, the columns of $dA$ and $dB$ are clearly in the kernel of $A^T$ and $B^T$, therefore $A^T dA = A^T dB = B^T dA = B^T dB = 0$. This implies that all the 'cross terms' disappear when expanding the expression:

$$(A + \varepsilon dA)^T (B + \varepsilon dB) = A^T B + \varepsilon^2 dA^T dB = A^T B + \varepsilon^2 \mathbf{a}\mathbf{b}^T$$

We also have

$$\|A + \varepsilon dA\|_F^2 = \|A\|_F^2 + \|\varepsilon dA\|_F^2 = \|A\|_F^2 + \varepsilon^2$$

and similarly, $\|B + \varepsilon dB\|_F^2 = \|B\|_F^2 + \varepsilon^2$. We then get

$$
\begin{aligned}
\varphi(A + \varepsilon dA, B + \varepsilon dB) &= g\Big((A + \varepsilon dA)^T(B + \varepsilon dB)\Big) + \frac{\lambda}{2}\left(\|A + \varepsilon dA\|_F^2 + \|B + \varepsilon dB\|_F^2\right) \\
&= g(A^T B + \varepsilon^2 \mathbf{a}\mathbf{b}^T) + \frac{\lambda}{2}\left(\|A\|_F^2 + \|B\|_F^2\right) + \lambda\varepsilon^2 \\
&= \left[g(A^T B) + \left\langle \nabla f(A^T B), \varepsilon^2 \mathbf{a}\mathbf{b}^T\right\rangle_F + O(\varepsilon^4)\right] + \frac{\lambda}{2}\left(\|A\|_F^2 + \|B\|_F^2\right) + \lambda\varepsilon^2 \\
&= \varphi(A, B) + \varepsilon^2\Big(\left\langle \nabla g(A^T B), \mathbf{a}\mathbf{b}^T\right\rangle_F + \lambda\Big) + O(\varepsilon^4)
\end{aligned}
$$

Let $G = \nabla f(A^T B) \in \mathbb{R}^{m \times n}$. We want to choose the unit vectors $\mathbf{a}$ and $\mathbf{b}$ that makes $\left\langle G, \mathbf{a}\mathbf{b}^T\right\rangle_F$ as negative as possible. The best choice is to choose $-\mathbf{a}$ and $\mathbf{b}$ to be the first left and right singular vectors of $G$ since this give the negative of the best rank-one approximation of $G$. So we choose $\mathbf{a} \in \mathbb{R}^m$ and $\mathbf{b} \in \mathbb{R}^n$ such that $G\mathbf{b} = -\sigma_1 \mathbf{a}$, and therefore

$$
\left\langle \mathbf{a}\mathbf{b}^T, G\right\rangle_F = \text{Tr}(\mathbf{b}\mathbf{a}^T G) = \text{Tr}(\mathbf{a}^T G\mathbf{b}) = -\sigma_1
$$

which gives

$$
\varphi(A + \varepsilon dA, B + \varepsilon dB) = \varphi(A, B) + \varepsilon^2\Big(-\left\|\nabla g(A^T B)\right\|_{op} + \lambda\Big) + O(\varepsilon^4)
$$

and (ii) implies that $(A, B)$ is not a local min. $\qquad \square$

Combining the previous two claims we can easily prove the theorem. Indeed, if $d > m$, then the kernel of $A^T$ is nontrivial and (i) is always satisfied. As a consequence, if $(A, B)$ is a local min of $\varphi$, then $\left\|\nabla g(A^T B)\right\|_{op} \leq \lambda$, and therefore $(A, B)$ must be a global min of $\varphi$.

$\qquad \square$

