# OpenReview forum: "Feature Collapse"
_ICLR.cc/2024/Conference — ICLR 2024 poster_

### Official Review · Reviewer_xgCm · 2023-10-31

**Soundness:** 3 good
**Presentation:** 2 fair
**Contribution:** 2 fair
**Rating:** 6
**Confidence:** 3

**Summary:**

This paper defines the notion of "feature collapse" which relates to the intuition that entities that play a similar role should receive similar representations. This notion is made precise for a toy NLP task. Here, labels are invariant to substitutions of input tokens within certain equivalence classes. The paper then studies, both theoretically and empirically, under what conditions tokens within such an equivalence class are assigned similar representations, i.e. "feature collapse" has occurred. The paper shows that for this toy task, feature collapse corresponds to stronger generalization in the presence of tail tokens. The paper also empirically shows that feature collapse only occurs when the network includes layer norm. Otherwise, the norm of representations depends on the frequency of the tokens in the training data. A major contribution is a theoretical proof that explains why the magnitudes of the learned representations depend on their frequencies.

**Strengths:**

* The proof that token embeddings depend on token frequencies in the proposed setting may provide a useful theoretical insight for future work to build on. The proofs in the paper are non-trivial.
* The paper may help explain a reason why layer normalization can improve generalization.

**Weaknesses:**

* While the key contributions of the paper are theoretical, the motivation for the theoretical results would have been clearer if a stronger connection could be made to more realistic settings. For example, the connection between feature collapse and generalization outside the context of the toy task does not seem to be well established. Additionally, the degree to which feature collapse sufficiently occurs or does not occur in the context of more realistic models and tasks is not clear. While perhaps not strictly necessary for such a theoretical paper, addressing such questions would have improved the motivation for this work.
* The role of the L2 regularization term in equation 2 could be clarified. The authors state "The regularization terms play no essential role apart from making proofs easier; the empirical picture remains the same without weight decay". Could the authors include the experimental results that support this claim? This also seems to diminish (although not completely) the key theoretical contribution. It is not surprising that in the presence of L2 regularization, feature magnitudes would depend on their frequency in the training data. It would be more interesting if we had theory that elucidated why the learned embeddings for rarer tokens have lower magnitude in the absence of explicit regularization.

Minor typos:

* Intro paragraph 1 - "gives same features"
* Intro contribution bullet 1 - "how a network learn representations"

**Questions:**

* Is there any setting where feature collapse may *not* be desirable? For example, consider a task with a significant degree of label noise and with limited data samples to learn from. Intuitively, features that are observed more frequently during training may be more reliable predictors of the label. Therefore, to the extent that the magnitude of word embeddings relates to their salience in making predictions, it seems that feature collapse may not be strictly desirable in such a setting.

---

> ### Author Response · Authors · 2023-11-17
>
> Thank you for taking the time to provide a thoughtful and careful review. We answer your questions and address your concerns below.
>
> &nbsp;
>
> **Weakness1**:
> >  The motivation for the theoretical results would have been clearer if a stronger connection could be made to more realistic settings... The degree to which feature collapse sufficiently occurs or does not occur in the context of more realistic models and tasks is not clear.
>
> We have studied the feature collapse phenomenon in a controlled setting where proofs are analytically tractable. We agree that, to better motivate this analytical study, we could have provided empirical evidence that this phenomenon occurs in more complex settings.
>
> In the updated version of the paper we:
>
> 1) Empirically investigate feature collapse in the setting of a Context Free Grammar (CFG), and where the network is a deep neural network with ReLU nonlinearities and LayerNorm module at each layer. Context Free Grammars [1] are synthetic data models that generate sentences with probabilistic trees and are widely used to understand natural language models (e.g. [2,3,4]). A CFG can be viewed as a `deep' version of our data model, in that there is a hierarchy of latent variables (words, concepts, meta-concepts, meta-meta-concepts, etc), and the latent variables from one level generate latent variables in the level below. The empirical results shows that, if the architecture of the neural network matches that of the CFG, then the representations at each layer of the networks collapse according to the corresponding latent variables.
>
> 2) Train a 2-layer transformer on the classification task depicted on Figure 1 of our paper. Feature collapse does take place in this setting as well.
>
> 3) Train a 2-layer transformer (with GPT-2 architecture) to predict the next token on sentences generated by our data model (i.e. a language modeling task). Feature collapse also occurs in this situation.
>
> These three experiments are briefly described in the conclusion of the updated version of our paper (page 9, in blue), with precise details and visuals in the first section of the appendix (page 12-15, in blue). They demonstrate that feature collapse still occurs with a more complex data model (CFG), more complex networks (such as transformers and deep networks with ReLU), and a different task (language modeling).
>
>
> [1] Chomsky. Three models for the description of language
>
> [2] Kim, Dyer, Rush. Compound probabilistic context-free grammars for grammar induction
>
> [3] Allen-Zhu, Li. Physics of Language Models: Part 1, Context-Free Grammar
>
> [4] Liu, Xie, Li, Ma. Same Pre-training Loss, Better Downstream: Implicit Bias Matters for Language Models
>
> &nbsp;
>
> **Weakness2**:
> >  The role of the L2 regularization term in equation 2 could be clarified
>
> Indeed, we could have been clearer on this point, and we have updated the paper accordingly (see paragraph in blue on page 4). Without weight decay, the objective (2) does not have a global minimum, which makes the theoretical study much more challenging. In such a situation, gradient descent leads to solutions where the magnitude of the weights grows logarithmically to infinity and becomes more and more aligned with the gradient, c.f. [1,2]. In Appendix A1 of the new version of the paper, we reproduce all our experiments but without weight decay. To train the networks, we performed 5 million iterations of stochastic gradient descent with a batch size of $100$ and a learning rate of $0.1$. As can be seen in Figure 5 and 6 in the appendix, the results are qualitatively very similar to those obtained with weight decay.
>
> [1] Ji, Telgarsky. Directional convergence and alignment in deep learning
>
> [2] Soudry at al. The Implicit Bias of Gradient Descent on Separable Data
>
>
> &nbsp;
>
> **Question**:
> > Is there any setting where feature collapse may not be desirable?  For example...
>
> We fully agree that, in some settings, feature collapse might not be desirable. To put it simply, in our setting the frequency of a word is irrelevant to the classifications task. Using a network architecture that 'removes' this irrelevant information is therefore beneficial. In a setting where the word frequency is meaningful, such as the one described by the referee, a network that ignores it would perform poorly.
>
> Generally speaking, we believe that the range of applications for deep learning is so varied that no one mechanism could possibly be right for all of them. Rather, the right network architecture is heavily task dependent. Our contribution, we hope, is to shed some light on how different architecture choices leads to different types of representations.

---

> > ### Comment · Reviewer_xgCm · 2023-11-21
> >
> > Thank you for your reply. I have increased my score based on the response.

---

> > > ### Author Response · Authors · 2023-11-21
> > >
> > > Thank you once more for your careful review and constructive criticism. We sincerely appreciate it.

---

### Official Review · Reviewer_ZbKw · 2023-10-31

**Soundness:** 3 good
**Presentation:** 3 good
**Contribution:** 3 good
**Rating:** 8
**Confidence:** 3

**Summary:**

The manuscript introduces the concept of "Feature Collapse," which states that input features with analogous semantic meanings would receive identical representations. This phenomenon is examined under two distinct scenarios: the first being a 2-layer network, and the second incorporating LayerNorm on the 2-layer network. Empirical evaluations reveal that in a uniform setting, both network configurations attain feature collapse in terms of representations and weights. However, in a non-uniform context, the integration of LayerNorm becomes essential for the complete feature collapse. The study concludes with theoretical findings that prove feature collapse.

**Strengths:**

1. The work is well presented and all notions included in the paper are discussed clearly.
2. The concept of feature collapse is interesting and intuitive.
3. The paper potentially offers an explanation regarding the impact of LayerNorm.

**Weaknesses:**

1. While the reviewer appreciates the feature collapse concept and believes this phenomenon is not limited to the simple network settings used in the manuscript, the absence of empirical validation within more complex networks or datasets constrains the soundness and significance of the findings.
2. The authors differentiate their work from the Neural Collapse (NC) concept by emphasizing that, unlike NC, Feature Collapse inherently suggests enhanced generalization and transfer learning capabilities. But once again, this claim lacks strong empirical evidence, the 2-layer network used in the paper is not comprehensive enough to demonstrate this point.

**Questions:**

1. The work devotes to study word representations, would similar phenomenon happens in vision as well (e.g., VisionTransformer uses image patches, does image patch with similar sematic also receive identical representations)?
2. The work primarily studies that features of words with similar semantics would collapse, how about words that sharing similar syntactic in a sentence? Would something similar happen as well?

(The reviewer is not asking for empirical evidence, just curious to hear the authors' thoughts.)

---

> ### Author Response · Authors · 2023-11-17
>
> Thanks for the time you spent carefully reviewing our work. We first answer your concerns:
>
> **Weakness1**:
> > The absence of empirical validation within more complex networks or datasets constrains the soundness and significance of the findings.
>
> We agree. It is important to show (empirically) that the feature collapse phenomenon is not limited to the simple controlled setting where proofs are analytically tractable. In the updated version of the paper, we conduct such an empirical investigation with more complex data models and more complex networks. Specifically, we ---
>
> 1) Empirically investigate feature collapse in the setting of a Context Free Grammar (CFG), and where the network is a deep neural network with ReLU nonlinearities and LayerNorm module at each layer. Context Free Grammars [1] are synthetic data models that generate sentences with probabilistic trees and are widely used to understand natural language models (e.g. [2,3,4]). A CFG can be viewed as a `deep' version of our data model, in that there is a hierarchy of latent variables (words, concepts, meta-concepts, meta-meta-concepts, etc), and the latent variables from one level generate latent variables in the level below. The empirical results shows that, if the architecture of the neural network matches that of the CFG, then the representations at each layer of the networks collapse according to the corresponding latent variables.
>
> 2) Train a 2-layer transformer on the classification task depicted on Figure 1 of our paper. Feature collapse does take place in this setting as well.
>
> 3) Train a 2-layer transformer (with GPT-2 architecture) to predict the next token on sentences generated by our data model (i.e. a language modeling task). Feature collapse also occurs in this situation.
>
> These three experiments are briefly described in the conclusion of the updated version of our paper (page 9, in blue), with precise details and visuals in the first section of the appendix (page 12-15, in blue). They demonstrate that feature collapse still occurs with a more complex data model (CFG), more complex networks (such as transformers and deep networks with ReLU), and a different task (language modeling).
>
>
> [1] Chomsky. Three models for the description of language
>
> [2] Kim, Dyer, Rush. Compound probabilistic context-free grammars for grammar induction
>
> [3] Allen-Zhu, Li. Physics of Language Models: Part 1, Context-Free Grammar
>
> [4] Liu, Xie, Li, Ma. Same Pre-training Loss, Better Downstream: Implicit Bias Matters for Language Models
>
> &nbsp;
>
> **Weakness2**:
> > The authors differentiate their work from the Neural Collapse (NC) concept by emphasizing that, unlike NC, Feature Collapse inherently suggests enhanced generalization and transfer learning capabilities. But once again, this claim lacks strong empirical evidence.
>
> We agree that our claim lack strong empirical evidence. We had wanted to emphasize the fact that feature collapse and neural collapse are truly distinct phenomena. In doing so we did not intend to over-claim regarding feature collapse and generalization/transfer learning. The offending sentences
>
> `Thus neural collapse has no consequences for generalization or transfer learning. By contrast, feature collapse  describes the emergence of ‘good’ local features, and this phenomenon goes hand in hand with generalization and transfer learning.'
>
> have been removed from the paper.
>
> Let us clarify the point we inartfully tried to make. In the context of our data model, neural collapse means that the network maps all sentences of category 1 to a single point in its last layer, all sentences of category 2 to a single point in its last layer, etc., and that these points are maximally opposed. In other words, neural collapse states that the network has perfectly learned the labels explicitly given during training. Also, neural collapse occurs for any network that is expressive enough -- it doesn't depend on the architecture. It doesn't provide any information on whether or not the network has learned useful lower-level local features, and so in this limited sense, does not speak directly to transfer learning.
>
> For feature collapse we study how different architectural choices lead to different types of local representations in the first layer (and intermediate layers in the case of the CFG) of the network. The same intermediate representation is simultaneously leveraged by many categories during classification. For example, the representation for the word 'cheese' is used by the upper layer to classify sentences from many different categories. As the representations are not directly tied to the categories of the classification task, they could conceivably be used in a transfer learning scenario.

---

> ### Author Response · Authors · 2023-11-17
> **Official Comment by Authors (continued)**
>
> We now answer your questions:
>
> **Question1**:
> >  Would similar phenomenon happens in vision as well (e.g., VisionTransformer)?
>
> This is an interesting question. Our intuition is that yes, patches which are semantically similar should receive similar representations. Our new experiments, in appendix A.3 and A.4, show that feature collapse does occur with transformer trained on our simple data model. Studying empirically the phenomenon on a real image dataset would be very interesting.
>
> &nbsp;
>
> **Question2**:
> >  The work primarily studies that features of words with similar semantics would collapse, how about words that sharing similar syntactic in a sentence?
>
> Our current data model is quite simplistic in the sense that tokens are either 'perfectly similar', or 'perfectly dissimilar' -- there is no middle ground. In future works, we would like to investigate more sophisticated models in which tokens might have different attributes (say color/size/shape, or semantic/syntactic). Some tokens might have some attributes in common, but might differ with respect to some others. The question is then: how is this reflected in their representations? What architectures promote useful representations for such a data model? This is a challenging question, and we currently have no answer. Our hope is that one would see some type of partial collapse along some dimensions coding for the attribute.

---

> ### Comment · Reviewer_ZbKw · 2023-11-20
>
> Thanks to the authors for the detailed and insightful response. The reviewer believes that the newly added empirical results help to enhance the connection between the theoretical framework and empirical observations. I'm raising my score 6-8 (want to give 7, but there is not, so 8...).
>
> Thanks again for the responses.

---

> > ### Author Response · Authors · 2023-11-21
> >
> > Thank you once again for your time and constructive criticisms; they have significantly helped us improve the paper. Thanks!

---

### Official Review · Reviewer_nQiF · 2023-11-01

**Soundness:** 3 good
**Presentation:** 4 excellent
**Contribution:** 3 good
**Rating:** 6
**Confidence:** 3

**Summary:**

The paper investigates feature collapse and "good" features learned at the first few layers. The good features should be invariant for entities serving the same concept for tasks at hand. The authors construct a synthesized task to investigate how feature collapse works. The paper reveals three kinds of feature collapse through synthetic experiments and theories and can be roughly summarized as follows: (1) Type-I feature collapse for models w/o LayerNorm, which collapses in both magnitudes and directions and will happen with uniform distribution. (2) Type-II feature collapse for models w/ LayerNorm will still collapse in both directions in magnitudes under long-tail distribution due to the help of LayerNorm. (3) Type-III feature collapse for models w/o LayerNorm, which collapses in directions but not magnitudes and will happen with long-tail distribution.

**Strengths:**

1. The paper proposes a view to understand the feature collapse with good definitions of "good" features. The understanding of LayerNorm functionality in the feature collapse, which is a key component in the modern Transformer architecture, would help the community further understand language models.
2. The paper's presentation is extremely clear and easy to follow. The synthetic experiments in Section 2 are sound, plausible, and offer reasonable intuitions.

**Weaknesses:**

My major concern is if the Type-II feature collapse still holds when LayerNorm has trainable parameters. The reason is that LayerNorm w/ trainable parameters will not force the word features to have a fixed magnitude. A simple experiment would suffice.

While I tend to vote for acceptance, I am not an expert in this field, so I will keep my score modest.

**Questions:**

Please see the weakness part.

---

> ### Author Response · Authors · 2023-11-17
>
> Thanks for the time you spent carefully reviewing our work. We answer your main concern below.
>
> > My major concern is if the Type-II feature collapse still holds when LayerNorm has trainable parameters. The reason is that LayerNorm w/ trainable parameters will not force the word features to have a fixed magnitude. A simple experiment would suffice.
>
> Thank you for suggesting the experiment. Following your recommendation, we carried out the experiment and  report the results in Section A.2 of the appendix (page 13 of the updated version of the paper). As can be seen on Figure 7, Type II collapse still holds in this situation. We currently do not have a proof for this, but let us offer a heuristic argument. If the representations are collapsed *before* applying the diagonal scaling and bias, then they should still be collapsed after (because it is a simple affine transformation). Moreover, we would not expect the bias and diagonal scaling to fundamentally change the loss landscape.
> We have not done it yet but we think a full proof is accessible with the techniques of the paper.
>
> &nbsp;
>
> We would also like to draw your attention to a new set of experiments that are now reported in the appendix. In the same spirit as your question, these experiments are designed to show that the feature collapse phenomenon is not limited to the simple controlled setting in which we were able to prove it, but also extends to more complex data models and more complex networks. Specifically we:
>
> 1) Empirically investigate feature collapse in the setting of a Context Free Grammar (CFG), and where the network is a deep neural network with ReLU nonlinearities and LayerNorm module at each layer. Context Free Grammars [1] are synthetic data models that generate sentences with probabilistic trees and are widely used to understand natural language models (e.g. [2,3,4]). A CFG can be viewed as a `deep' version of our data model, in that there is a hierarchy of latent variables (words, concepts, meta-concepts, meta-meta-concepts, etc), and the latent variables from one level generate latent variables in the level below. The empirical results shows that, if the architecture of the neural network matches that of the CFG, then the representations at each layer of the networks collapse according to the corresponding latent variables.
>
> 2) Train a 2-layer transformer on the classification task depicted on Figure 1 of our paper. Feature collapse does take place in this setting as well.
>
> 3) Train a 2-layer transformer (with GPT-2 architecture) to predict the next token on sentences generated by our data model (i.e. a language modeling task). Feature collapse also occurs in this situation.
>
> These three experiments are briefly described in the conclusion of the updated version of our paper (page 9, in blue), with precise details and visuals in the first section of the appendix (page 12-15, in blue).  They demonstrate that feature collapse still occurs with a more complex data model (CFG), more complex networks (such as transformers and deep networks with ReLU), and a different task (language modeling).
>
>
> [1] Chomsky. Three models for the description of language
>
> [2] Kim, Dyer, Rush. Compound probabilistic context-free grammars for grammar induction
>
> [3] Allen-Zhu, Li. Physics of Language Models: Part 1, Context-Free Grammar
>
> [4] Liu, Xie, Li, Ma. Same Pre-training Loss, Better Downstream: Implicit Bias Matters for Language Models

---

> > ### Comment · Reviewer_nQiF · 2023-11-20
> >
> > Thanks to the authors' additional experiments and hard work! I've read the rebuttal and will keep my tend for acceptance.

---

> > > ### Author Response · Authors · 2023-11-21
> > >
> > > Thank you once more for your feedback, suggestions, and the time you spent carefully reviewing our work. We sincerely appreciate it.

---

### Official Review · Reviewer_pmX6 · 2023-11-01

**Soundness:** 3 good
**Presentation:** 3 good
**Contribution:** 3 good
**Rating:** 5
**Confidence:** 2

**Summary:**

The paper delves into the concept of "feature collapse," where entities with similar roles in a learning task are given analogous representations. To understand this, the authors use a synthetic task where a learner classifies 'sentences' made of L tokens. Their experiments reveal a direct relationship between feature collapse and generalization. They demonstrate that, with sufficient data, distinct tokens with the same task roles have identical feature representations in a network's first layer. The research conclusively proves that neural networks, when trained on this task and equipped with a LayerNorm module, develop interpretable and meaningful representations in their initial layer. The paper's key contributions include defining 'good features' mathematically and deriving analytical formulas for a two-layer network's weights.

**Strengths:**

Strength:
1. Paper is well organized
2. The settings of experiments and assumptions of theoretical analysis are carefully designed.

**Weaknesses:**

Weakness:

In general, I am not sure how the results could be useful.
* For the theory part, it would be helpful to highlight core technical challenges the authors met and addressed in the proofs, or any new proof techniques the authors developed.
* For the experiment part, I am not sure how it could help design or explain applications. I totally agree that semantically similar instances would lead to similar representations. But is it benign or not for the network to assign exactly the same representation to equivalent instances? If it is benign, how to ensure it happens in practice? If not, then how to avoid it? It would be helpful for the authors to give some convincing scenarios that people care about this phenomenon.

In addition, I suppose there should be a section for conclusions and limits.

**Questions:**

Questions:
1. Maybe a simpler and more intuitive experiment to demonstrate this “feature collapse” behavior, is to assign two embeddings (to be learned) to one token, but during training this token is randomly mapped to these two embeddings, and see if these two embeddings become similar after training.
2. Does the condition assumed on λ (in Thm. 2) align with the choice of λ in experiments?
3. Can we empirically verify the feature collapse in deeper layers?
4. I have concerns about the difficulty of learning equivalence from the data by the model itself without special treatments, including data augmentations and introducing explicit invariance into models. I think the root is: if the optimization objective is not the equivalence, the network could ignore the equivalence and just memorize the data. Another related and broader phenomenon is the failure to learn “A = B” equals to “B = A” in language models [1]. In computer vision, semantically equivalent inputs (pixels, small image patches, whole images, etc.) may not lead to the same feature, for both early layers and deep layers/final outputs. This is easy to understand, because even for contrastive learning where the network is forced to learn invariant features under augmentations, different views (augmented images) cannot guaranteed to lead to the same feature after training.

[1] 'The Reversal Curse: LLMs trained on "A is B" fail to learn "B is A"' 2023.

---

> ### Author Response · Authors · 2023-11-17
>
> Thanks for the time you spent carefully reviewing our work. We first answer your concerns:
>
>
> > For the theory part, it would be helpful to highlight core technical challenges the authors met and addressed in the proofs, or any new proof techniques the authors developed.
>
> Thank you for the question. One of the key difficulties was to realize that the latent structure of the problem exhibits a type of 'symmetry' when we consider a large enough number of latent variables. This symmetry can in turn be leveraged to obtain fully tractable analytical solutions. Specifically, the symmetry on the latent variables allows us to write the loss in a type of 'spherical coordinates', where a `sphere' is defined as the latent variables which are at a Hamming distance $r$ from a reference latent variable. This spherical decomposition for the loss then allowed us to explicitly characterize global minima. To the best of our knowledge, these techniques have not been used before in this context.
>
> Another surprise was the fact that, when the words are not uniformly distributed, we could still derive analytical formula for the critical points of the loss (both with and without layer norm). We did not expect such solutions to have tractable analytical formula.
>
> We have added such a discussion in the new version of the paper (see blue paragraph on p9).
>
>
> &nbsp;
>
> >  I totally agree that semantically similar instances would lead to similar representations. But is it benign or not for the network to assign exactly the same representation to equivalent instances?
>
>
> In general the desirability of full collapse depends on the task at hand. If collapse destroys information relevant to a particular task then it is clearly not benign. A simple but concrete example of this might be the overall frequency of a word in our data model. In our setting this frequency is irrelevant to our classification task; so, full collapse (i.e., words with distinct frequencies but the same meaning receive the same representation) is beneficial and empirically improves generalization. Had we changed the task to one where learning the frequency is needed to solve it, full collapse is not benign since the frequency information is destroyed.
>
>
> &nbsp;
>
> > If it is benign, how to ensure it happens in practice? If not, then how to avoid it?
>
> One of the outcomes of our study, which we did not expect going in but makes sense in retrospect, is that proper normalization helps similar tokens receive similar embeddings.
>
> &nbsp;
>
> > It would be helpful for the authors to give some convincing scenarios that people care about this phenomenon.
>
> Personally, we care about the phenomenon to the extent that it sheds light on the mechanisms by which semantically similar instances receive similar representations. Ideally, we would want a theoretical framework that explains the `if, when and how' of semantically similar instances receiving similar representations. This is a challenging question, and in particular, it requires a careful and precise mathematical definition of what 'semantically similar' means in the form of a metric and a proof that smallness in this metric implies small distance between representations.
>
> In the current work, we study an extreme version of the above problem. Specifically, we consider a setting in which tokens are either semantically identical or semantically unrelated (i.e., the semantic distance between tokens is binary). In this simple setting, we can prove that representations are either identical or maximally opposed.
>
> We are very interested in extending our analysis to settings in which the semantic distance between tokens in not binary. In future works, we would like to investigate more sophisticated models in which tokens might have different attributes (say color/size/shape). Some tokens might have some attributes in common, but might differ with respect to some others. The question is then: how is this reflected in their representations? What architectures promote useful representations for such a data model? This is a challenging question, and we currently have no answer. Our hope is that one would see some type of partial collapse along some dimensions coding for the attribute.
>
> &nbsp;
>
> > In addition, I suppose there should be a section for conclusions and limits.
>
> We have added such sections at the end of the new version of the paper (blue paragraph on page 9).

---

> ### Author Response · Authors · 2023-11-17
> **Official Comment by Authors (continued)**
>
> We now answer your questions:
>
> **Q1:**
> > Maybe a simpler and more intuitive experiment to demonstrate this “feature collapse” behavior, is to assign two embeddings (to be learned) to one token, but during training this token is randomly mapped to these two embeddings, and see if these two embeddings become similar after training.
>
> Note that this experiment is essentially the same than the one we conduct in our work. Indeed, in our setting, each `concept token' is randomly mapped to one of out of 400 embeddings (the 400 embeddings corresponding to the 400 words contains in the concept).
>
> &nbsp;
>
> **Q2:**
> > Does the condition assumed on λ (in Thm. 2) align with the choice of λ in experiments?
>
> Yes it does: we have now clarified this in the paper.
>
> &nbsp;
>
> **Q3:**
> > Can we empirically verify the feature collapse in deeper layers?
>
> Thanks for the question. We conducted a careful experiment to verify this. The data model presented in the paper extends to one with a deeper hierarchy of latent structures. Recall that words are partitioned into concepts and that the latent variables are sequences of concepts. We can further partitioned the latent variables into 'meta-concepts' and create 'deeper latent variables' that are sequences of 'meta-concepts'. We can iterate this process to obtain a hierarchy of any depth. Such data model is a particular instance of a Context Free Grammar [1], which generate sentences with probabilistic trees and are widely used to understand natural language models (e.g. [2,3,4]). In Figure 9 in the appendix, we provide an illustration of a simple depth 3 context free grammar.
>
> We ran experiments with a context free grammar of depth 4 (meaning that we have words, concepts, meta-concepts and meta-meta-concepts). We used a deep neural network with ReLU nonlinearities and LayerNorm module at each layer. The architecture of the neural network was chosen to match that of the context free grammar; see Figure 10 in the appendix. In Figure 11 we plot the activations after each of the three hidden layers and readily observe the expected feature collapse phenomenon. All segments of the input sentence that correspond to same concept, meta-concept, or meta-meta-concept, receive the same representations in the appropriate layer of the network (layer 1 for concepts, layer 2 for meta-concepts, and layer 3 for meta-meta-concepts). This shows that the feature collapse phenomenon also occurs in deeper layers.
>
> This experiment is briefly described in the conclusion of the updated version of our paper (p9 in blue). The precise details and visuals can be found in appendix A5 (page 13).
>
> [1] Chomsky. Three models for the description of language
>
> [2] Kim, Dyer, Rush. Compound probabilistic context-free grammars for grammar induction
>
> [3] Allen-Zhu, Li. Physics of Language Models: Part 1, Context-Free Grammar
>
> [4] Liu, Xie, Li, Ma. Same Pre-training Loss, Better Downstream: Implicit Bias Matters for Language Models
>
> &nbsp;
>
> **Q4:**
> > I have concerns about the difficulty of learning equivalence from the data by the model itself without special treatments, including data augmentations and introducing explicit invariance into models. I think the root is: if the optimization objective is not the equivalence, the network could ignore the equivalence and just memorize the data. Another related and broader phenomenon is the failure to learn “A = B” equals to “B = A” in language models [1]. In computer vision, semantically equivalent inputs (pixels, small image patches, whole images, etc.) may not lead to the same feature, for both early layers and deep layers/final outputs. This is easy to understand, because even for contrastive learning where the network is forced to learn invariant features under augmentations, different views (augmented images) cannot guaranteed to lead to the same feature after training.
>
> The referee is correct that, despite many attempts, even the most modern network architectures need not produce ideal representations, and that this can lead to sub-optimal performance (like a language model not recognizing that “A = B” is the same than “B = A”). Understanding the interplay between architecture, objective, and the presence or absence of `good' representations is exactly what motivated our study. Note that, in our case, good features emerge even if the loss is not the equivalence itself. Rather, the equivalence emerges as the best mechanism for solving the task. This does not hold in general, and we hope that further elaborations on the present work elucidate the circumstances where a desired equivalence is not learned. Identifying these roadblocks would then improve network design.
>
> &nbsp;
>
> **New Experiments with Transformers**:
> To conclude, we would like to draw your attention to a new set of experiments reported in  appendix A3 and A4, in which we  investigate the feature collapse phenomenon in transformer architectures in both a classification setup and the usual next-word-prediction setup.

---

### Meta-Review · Area_Chair_PCCK · 2023-12-15

**Metareview:**

The manuscript presents the idea of "Feature Collapse," which posits that input features sharing similar semantic meanings will be encoded with identical representations. This phenomenon is investigated in two different scenarios: the first scenario involves a 2-layer network, while the second scenario includes the incorporation of LayerNorm into the 2-layer network. Empirical assessments demonstrate that in a consistent environment, both network configurations exhibit feature collapse in relation to representations and network parameters.

Most reviewers find the work is well presented, the concept of feature collapse is interesting and intuitive, and the work can potentially explan the impact of LayerNorm in transformers and LLMs. Hence, we recommend for acceptance.

In the camera-ready version, please include all the comments the reviewers have raised.

**Justification For Why Not Higher Score:**

n/a

**Justification For Why Not Lower Score:**

Most reviewers agree this is a good paper.

---

### Decision · Program_Chairs · 2024-01-16

Accept (poster)